# Single cell transcriptional perturbome in pluripotent stem cell models

Elisa Balmas [1,3]✉, Maria L Ratto [1,3], Kirsten E Snijders [1,3], Silvia Becca [1], Carla Liaci [1], Irene Ricca [1], Giorgio R Merlo[1], Raffaele A Calogero[1], Luca Alessandrì[1], Sasha Mendjan [2] & Alessandro Bertero [1]✉

## Abstract

Functional genomics screens in human induced pluripotent stem cells (hiPSCs) remain challenging despite their transformative potential. We developed iPS2-seq: an inducible, clone-aware screening platform that enables phenotype-agnostic, single-cell resolved dissection of loss-of-function effects in hiPSC derivatives, including complex multicellular models such as organoids. iPS2-seq distinguishes true perturbation effects from genetic and epigenetic variability. It supports pooled and arrayed formats, integrates with microfluidic or split-pool single-cell RNA sequencing, and extends to multi-omic profiling of chromatin and proteins. A dedicated pipeline, *catcheR*, streamlines design and analysis. The platform enables stage-specific follow-up dissection of screen hits. We demonstrate this by targeting congenital heart disease-associated genes in monolayer cardiomyocytes and organoids. This reveals that epigenetic neuroectodermal priming interferes with germ layer differentiation in specific clones. Accounting for this bias, we show that *SMAD2* controls cardiac progenitor specification, with knockdown redirecting cells toward fibroblast and epicardial fates. iPS2-seq unlocks rigorous functional genomics in hiPSC-based models.

**Keywords** Functional Genomics; Human Pluripotent Stem Cells; Loss of Function; Pooled Screens; Single-cell RNA-seq
**Subject Categories** Chromatin, Transcription & Genomics; Stem Cells & Regenerative Medicine

## Introduction

The genomics revolution has led to the discovery of a growing number of putative disease-associated variants (Claussnitzer et al, 2020). This emerging power prompted the equally critical challenge of determining pathogenic causalities and mechanisms. Large-scale functional annotation of human genes remains arduous, particularly for developmentally transient or rare cell types (Shendure et al, 2019). As just one example, there is limited understanding of genotype–phenotype correlations in congenital heart disease (CHD; Morton et al (2022)), the most common form of life-threatening developmental defects (Tsao et al, 2023).

Dropout loss-of-function (LoF) screens have been instrumental, but their design requires a priori knowledge of disease mechanism, which is often lacking. Phenotype-agnostic screens relying on single-cell RNA sequencing (scRNA-seq) readouts have recently emerged as an attractive alternative (Camp et al, 2019). Various methods based on short hairpin RNA (shRNA; Aarts et al, (2017)) or CRISPR-Cas9 technologies (Dixit et al, 2016; Adamson et al, 2016; Jaitin et al, 2016) have been implemented in primary or immortalized cells that can be efficiently transfected or transduced. Yet, these models cannot cover the full spectrum of human cell types, nor do they faithfully recapitulate human pathophysiology.

Human pluripotent stem cells (hPSCs) offer the potential to study gene function in virtually any clinically relevant cell type. Patient-derived human induced pluripotent stem cells (hiPSCs) allow the study of disease modifiers to identify potential therapeutic targets in the context of personalized medicine (Vandana et al, 2023). Emerging approaches to obtain organoids (Corsini and Knoblich 2022), gastruloids (Steventon et al, 2021), blastoids (David et al, 2023), and integrated embryo models (Bao et al, 2022), provide advanced multi-scale platforms to understand cellular crosstalks and advance tissue maturation. Realizing this collective potential requires efficient methods to perturb gene function in hPSC-derived models. However, hPSCs pose specific challenges that complicate the implementation of pooled scRNA-seq screens (Balmas et al, 2023). These include high sensitivity to genotoxic nucleases, genetic and epigenetic clonal variability, asynchronous and heterogeneous differentiation, unstable transgene expression, and poor transfection/transduction efficiency post-differentiation. scRNA-seq screens in emerging PSC-based three-dimensional models add further complexity due to increased cell-type heterogeneity and structural constraints. While promising advances have been made (Li et al, 2023b), particularly in neural models (Kampmann, 2020), we still lack approaches applicable to systems more affected by these limitations, such as the heart, which so far has been examined largely by descriptive scRNA-seq studies (Miranda et al, 2023).

To bridge this technological gap, we developed iPS2-seq: iPS-optimized inducible Post-transcriptional Silencing in pool

---

[1]Molecular Biotechnology Center "Guido Tarone", University of Turin, Turin, Italy. [2]Institute of Molecular Biotechnology, Vienna, Austria. [3]These authors contributed equally: Elisa Balmas, Maria L Ratto, Kirsten E Snijders. ✉E-mail: elisa.balmas@unito.it; alessandro.bertero@unito.it

deconvoluted by single-cell sequencing. This method allows phenotype-agnostic screens in hPSCs and their derivatives through mRNA-depleting, clonally controlled, single-cell aware, isogenically engineered, stage-specific, and reversible LoF perturbations. We optimized molecular biology, genome editing, and single-cell genomics protocols to implement iPS2-seq with either a cost-effective homebrew sequential split-pool method (single-cell combinatorial indexing RNA sequencing, sci-RNA-seq; Cao et al (2017)) or a widely used commercial microfluidics partitioning platform (10X Genomics), which also enables multi-omic applications. We report a dedicated bioinformatics pipeline, *catcheR* (clonality and treatment controlled shRNA effect findeR), which supports screen design, annotation of perturbations to single-cell transcriptomes, and downstream analyses. We demonstrate the approach by studying CHD-associated genes in both monolayer hiPSC-derived cardiomyocytes (hiPSC-CMs) and self-assembling multilineage cardiac organoids (Hofbauer et al, 2021). In all, iPS2-seq provides a technological solution to meet the challenges of modern human functional genomics.

# Results

## Design principles underlying iPS2-seq

We set out to develop a pooled LoF functional genomics platform optimized for hPSC models, defined by the following key features. First, we sought unbiased phenotyping to enable the study of gene function when choosing a selective phenotypic marker is either not possible (e.g., unclear expectations) or not desirable (e.g., multiple relevant phenotypes to assess simultaneously). We decided to rely on scRNA-seq, which is now broadly accessible, scalable, reasonably sensitive, and well-supported analytically. We further avoided dependence on proprietary platforms. Thus, we developed a method that can be implemented with any scRNA-seq approach relying on polyadenosine (pA)-primed reverse transcription (RT) to capture 3' mRNA ends, a strategy used by most current methods.

Second, we ensured isogenic single-copy expression to avoid heterogeneity in perturbation copy number, positional effects, and insertional mutagenesis arising from alternatives like transfection or transduction. We reasoned that expressing pooled perturbations via genome editing at a defined locus would minimize measurement noise, enabling an equivalent signal with fewer cells, thereby reducing scRNA-seq costs and increasing screening scalability. We chose the human *AAVS1* locus, a genomic safe harbor that supports robust expression of shRNA and CRISPR-Cas9 perturbations in over a dozen hPSC-derived cell types (Bertero et al, 2016).

Third, we prioritized off-the-shelf applicability to facilitate widespread adoption across any hPSC line. We avoided strategies requiring multiple genome editing steps, a significant drain on time and resources, and a bottleneck to data generation. We also did not pursue allele-specific gene editing, as it is not foolproof and is influenced by genetic variation. Instead, we opted for an established gene trap-based strategy (Hockemeyer et al, 2009) that achieves >95% on-target, nuclease-facilitated homologous recombination and is thus compatible with pooled genome editing (Bertero et al, 2016). Notably, the lower efficiency of genome editing compared to other delivery methods does not represent the main bottleneck for

scRNA-seq screens, which are cost-limited at the final stage and typically involve fewer than ~100 genes (Li et al, 2023b).

Fourth, we implemented inducibility and reversibility, key features for dissecting stage-specific mechanisms during hPSC differentiation. Inducibility is also necessary for studying essential genes or those regulating the pluripotent state. We selected a tetracycline-inducible (tet-ON) system that we had previously optimized, which tightly regulates an RNA polymerase III (Pol III) promoter via a codon-optimized tet repressor (OPTtetR) driven by the strong CAG promoter (Bertero et al, 2016). The system is delivered as a single cassette, shows minimal leakiness without tet, and, critically, remains active in differentiated derivatives of all three germ layers. This contrasts with a widely used tet-ON system based on reverse tetracycline transactivator (rtTA), which activates the tetracycline responsive element (TRE), an RNA polymerase II (Pol II) promoter: we and others showed that while effective in hPSCs, some ectodermal derivatives, TRE-driven expression is often heterogeneous or inefficient in many hPSC-derived meso-derm- and endoderm-derived lineages (Bertero et al, 2016; Mandegar et al, 2016; Ordóvás et al, 2015; Guichardaz et al, 2024).

Fifth, we employed post-transcriptional silencing to avoid interference with nuclear gene regulation. We chose shRNAs, which are effective even when expressed as single copies (Bertero et al, 2016), are available in validated libraries for all human genes (Moffat et al, 2006), allow rapid and potent silencing of specific mRNA isoforms, and have well-understood, manageable off-target effects. Tet-ON-controlled shRNAs also allow tunable and reversible LoF. Partial LoF more faithfully models human diseases caused by functional haploinsufficiency than full gene loss. Since shRNAs expressed by Pol III lack a pA, required for identifying perturbations via 3' scRNA-seq, we paired each shRNA with a unique barcode embedded in the 3' untranslated region (UTR) of a polyadenylated transcript, supplied by the OPTetR gene in our all-in-one tet-ON system. We considered the strategy of Aarts and colleagues, which used Pol II-driven artificial microRNAs (Aarts et al, 2017), but this conflicted with our goal of creating an inducible system that remains active post-differentiation. We also rejected CRISPR-Cas9 knockouts due to their irreversibility and susceptibility to genetic compensation (Rossi et al, 2015; El-Brolosy et al, 2019), mRNA misregulation (Tuladhar et al, 2019), and p53-led genotoxic effects, particularly problematic in hPSCs (Merkle et al, 2017; Ihry et al, 2018; Haapaniemi et al, 2018; Enache et al, 2020). Lastly, we did not pursue CRISPR interference (CRISPRi), which requires multiple single guide RNAs (sgRNAs) for maximal efficiency (Replogle et al, 2020), and relies on expressing large fusion proteins with catalytically inactive Cas9 that can be silenced during hPSC differentiation (Karbassi et al, 2024).

Sixth, we introduced clonal awareness to rigorously control for genetic and epigenetic variability arising from hPSC culture and/or genome editing. Clonal variability is a well-known issue in the hPSC field, and we hypothesized that it would increase the noise in pooled screens, a possibility that, to our knowledge, has not been systematically tested. While scRNA-seq phenotyping allows grouping of cells by clonal origin via barcodes, identifying outlier clones remains a challenge without a robust baseline for comparison. The design choices described above help overcome this limitation and, to our knowledge, enable the first reliable separation of clonal effects from those of LoF perturbations within the same cell

population. Each shRNA barcode, inserted in the 3' UTR of the OPTtetR mRNA, is modified with a short stretch of random bases to serve as a "unique clonal identifier" (UCI), enabling discrimination of hPSC clones carrying the same shRNA but arising from distinct genome editing events, drawn from a large, diverse sequence pool. Crucially, since OPTtetR is constitutively expressed, the UCI can be read in tet-untreated cells to assess the clonal effects on transcriptomes, while tet-treated samples allow specific measurement of individual LoF perturbations.

## Method overview and setup of proof-of-principle experiments

An iPS2-seq experiment has four key stages: pooled shRNA cloning, pooled hPSC genome editing and, optionally, differentiation, scRNA-seq, and bioinformatics analysis (Fig. 1A). Through extensive experimental optimization (Appendix Text), we developed a scalable pipeline for molecular cloning and genome editing to obtain pools of hPSCs each carrying an individual inducible shRNA (Appendix Protocol 1). The identity of each shRNA can be deduced from a barcode (BC) detected through scRNA-seq in parallel to the single-cell transcriptome. Additionally, each hPSC clone can be tracked through a UCI, so as to distinguish perturbation effects from background noise arising from clonal variability. Following scRNA-seq analyses using either sci-RNA-seq (Appendix Protocol 2) or 10X Genomics (Appendix Protocol 3), data analysis is empowered by *catcheR* (Appendix Protocol 4). We demonstrate the method through a series of proof-of-principle experiments aimed at dissecting the cell-autonomous roles of five genes with an established genetic association to CHD (Appendix Table S1): *GATA4* (Pehlivan et al, 1999; Garg et al, 2003), *NKX2-5* (Schott et al, 1998), *SMAD2* (Zaidi et al, 2013), *KMT2D* (Ng et al, 2010), and *CHD7* (Vissers et al, 2004).

To express individual inducible barcoded shRNAs against these targets and controls in clonally-tagged hPSCs, we followed the procedure described in detail in Appendix Protocol 1. First, we performed a two-step pooled cloning of 23 barcoded shRNAs in a puromycin resistance-carrying, *AAVS1*-targeting plasmid to generate an all-in-one tet inducible cassette (Fig. EV1A). After the first cloning step of the shRNA plus associated BC and UCI, a single transformation yielded >2300 bacterial colonies (>100-fold shRNA coverage), with ~98% cloning efficiency (Fig. EV1B) and ~74% Sanger-confirmed accuracy (Appendix Table S2). shRNA representation was statistically in line with a normal distribution with the expected median (Appendix Table S2). shRNA-BC swapping occurred in ~11% of plasmids, a lower rate than in optimization experiments (Appendix Table S2), and was corrected through sequencing-based reassignment (Fig. 1B).

In the second cloning step, the OPTtetR cassette was inserted between the shRNA and UCI-BC to complete the all-in-one inducible construct. This yielded >230 colonies (>tenfold coverage), with ~96% efficiency (Fig. EV1C). Sequencing identified 210 distinct plasmids with balanced shRNA representation, aside from a few over-/under-represented constructs likely reflecting synthesis biases (Fig. 1C). In all, we obtained a sufficiently diverse and representative plasmid pool with known associations between shRNAs and unique barcodes.

Final plasmids were co-transfected into WTC-11 hiPSCs (Kreitzer et al, 2013) along with zinc finger nucleases (ZFN)

targeting *AAVS1* and a filler vector with a neomycin resistance cassette to enrich for biallelic integration. Cells were co-selected with puromycin and neomycin, followed by fialuridine (FIAU) to eliminate random integrants expressing thymidine kinase (TK, encoded outside the targeting cassette; Fig. 1A). From a first transfection, we recovered 32 clones. Genotyping and FIAU sensitivity revealed that ~45% of clones with suspected random integrations were eliminated (Fig. EV1D), suggesting that in the remaining ~55%, TK and, likely, the shRNA were silenced. Overall, up to ~81% of surviving clones were estimated to carry a single, functional shRNA. A second transfection yielded 48 additional clones, which were pooled and FIAU-selected to obtain ~40 more survivors. Together, the two rounds generated ~67 hPSC clones, with up to 54 predicted to express single inducible shRNAs, providing >twofold average coverage of the shRNAs. This moderate coverage represents a good real-world test of iPS2-seq.

## Phenotype agnostic screening with combinatorial indexing scRNA-seq

To exemplify iPS2-seq, we first focused on sci-RNA-seq, a sequential split-pool cell barcoding approach (Cao et al, 2017; Cao et al, 2019). Compared to commercial implementations, sci-RNA-seq does not require specialized instruments, costs as little as $0.03–0.2 per cell including NGS (depending on the experiment scale; Cao et al, (2019)), and is easily adapted to custom reagents. Conversely, sci-RNA-seq yields a lower amount of transcript counts per cell compared to commercial alternatives. Thus, we reasoned that if we could efficiently implement iPS2-seq in this context, it would not only provide a useful, cost-effective solution but also evidence that the method should be easily applicable in more sophisticated and sensitive commercial solutions.

After a series of optimization experiments (Appendix Text), we developed "iPS2-sci-seq", a custom variation of the sci-RNA-seq protocol which enables the efficient detection of shRNA-associated UCI-BCs in parallel to single-cell transcriptomes (Fig. 1D and Appendix Protocol 2). To boost UCI-BC detection, these are enriched during both RT and PCR using target-specific primers sharing cell barcodes with transcriptomic reads, yielding matched perturbation and expression data (Fig. EV1E,F).

We implemented this protocol in cardiomyocytes differentiated in a monolayer using a biphasic Wnt modulation protocol (Bertero et al, 2019a), inducing shRNA expression from day 0 to day 23 with tet. Two independent differentiations yielded >75% cardiac troponin T positive (cTnT +) hiPSC-derived cardiomyocytes (hiPSC-CMs; Fig. EV1G). We analyzed PFA-fixed nuclei by sci-RNA-seq with 96 RT and 384 PCR barcodes, profiling ~9600 nuclei. From ~300 million NGS reads, 6874 high-quality transcriptomes (over 500 unique molecular identifiers, UMIs) were recovered after demultiplexing and filtering (Fig. EV1H). Thus, iPS2-sci-seq libraries were comparable in quality to standard sci-RNA-seq (Cao et al, 2017).

Using *catcheR*, we generated a gene expression matrix indexing each nucleus by its shRNA and clonal ID, applying stringent UMI-based filtering for confident UCI-BC assignment (Appendix Protocol 4; Fig. 1E). This yielded 3465 nuclei confidently expressing individual shRNAs (1929 and 1536 nuclei for two biological replicates; Appendix Table S3), an acceptable ~50% of quality-filtered nuclei. Nuclei expressing a single shRNA were 87% of those

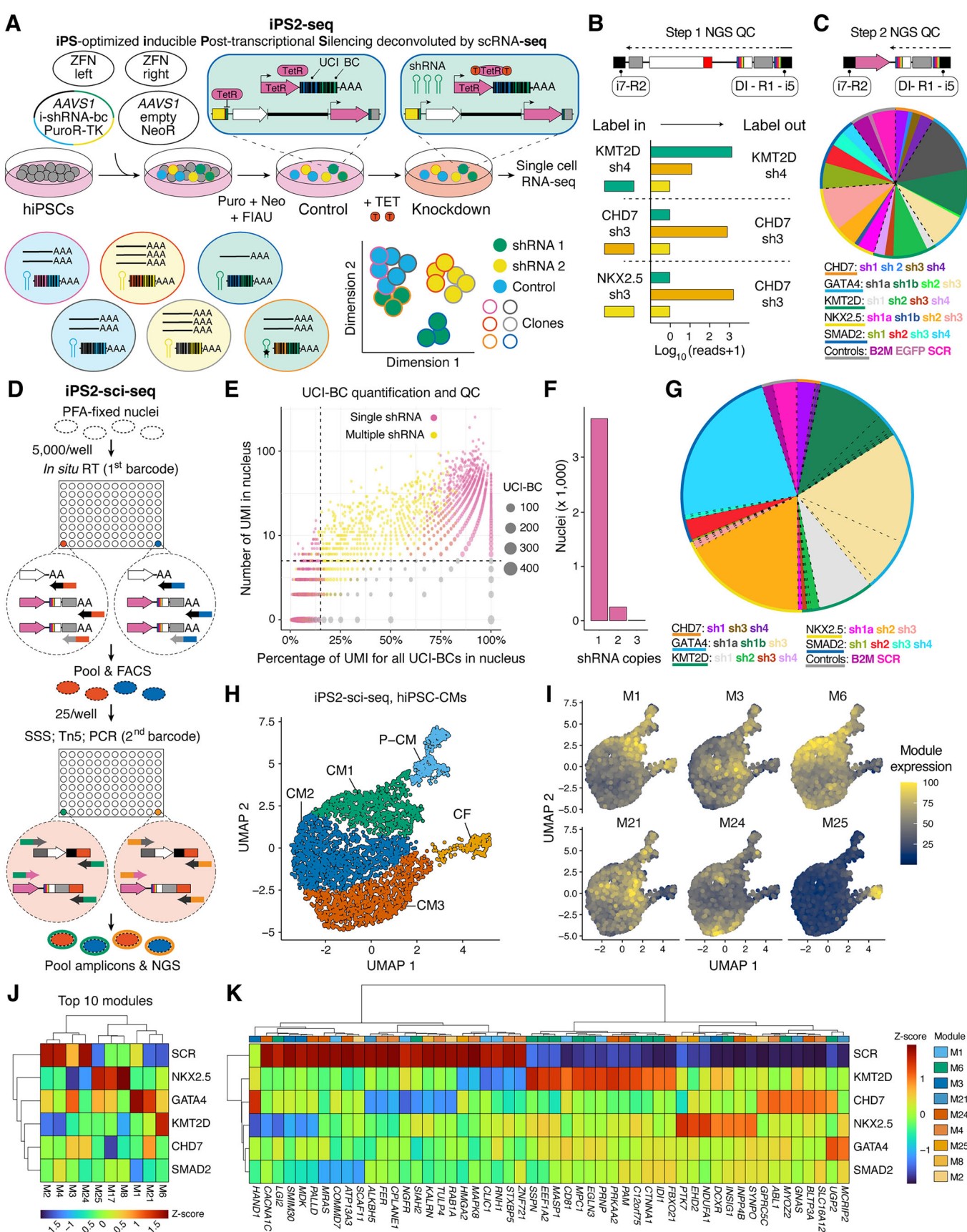

◄ **Figure 1.  Decoding the transcriptional perturbome with iPS2-sci-seq.**

(A) Schematic showing pooled generation of six hiPSC clones, each carrying one of three inducible shRNAs. Expression is distinguishable in single-cell transcriptomes via associated barcodes (BCs), while clonal origin is tracked by unique clonal identifiers (UCIs); this enables identification of outliers, such as cells containing a mutated shRNA (star). Targeting plasmids are generated through two sequential pooled cloning steps (Fig. EV1A). (B) Strategy and representative results of NGS-based quality control (QC) of shRNA and UCI-BC inserts in intermediate plasmids from step 1. Rare barcode swaps can be reassigned using UCIs. DI: diversity index. (C) As in (B), but showing QC of UCI-BCs in the final plasmids after step 2. Bacterial clones for each shRNA are quantified to evaluate distribution; Kolmogorov–Smirnov (KS) normality test $P = 0.0769$. (D) Customized sci-RNA-seq enriches for shRNA-associated UCI-BCs (3' of pink mRNA, OPTtetR) during both reverse transcription (RT) and PCR, assigning the same sets of barcodes (red and blue: RT; orange and green: PCR) as for endogenous transcripts (white mRNA). For simplicity, only the tetR-primed OPTtetR cDNA is shown at the second step, but the same PCR amplifies pT-primed OPTtetR (Fig. EV1E). SSS second-strand synthesis. (E) Filtering of UCI-BC counts (UMIs) from an iPS2-sci-seq experiment in hiPSC-CMs shown in (F–K). (F) Quantification of shRNA expression per nucleus. Only nuclei carrying a single shRNA are retained. (G) Distribution of hiPSC clones based on associated shRNAs (compare with plasmid distribution in (C)). (H) Dimensionality reduction and clustering. CM1-3 three subsets of cardiomyocytes, P-CM proliferating cardiomyocytes, CF cardiac fibroblasts (Fig. EV1J,K). (I) Aggregated expression scores for selected gene modules (M), visually distinct from top 10 in (J). (J) Hierarchical clustering of 10 out of 40 most variant gene modules by Z-score differences across perturbations. (K) Top 20% most variable genes by Z-score differences across perturbation, among those significantly affected according to scMAGeCK (Yang et al, 2020), drawn from the modules shown in (J). Source data are available online for this figure.

that could be assigned to at least one perturbation (Fig. 1F), consistent with genotyping estimates (Fig. EV1D). We recovered 45 clones representing 18 out of 23 shRNAs, with some clones over-represented (Fig. 1G), highlighting selective clonal expansion/depletion during editing or differentiation, a key consideration addressed in subsequent experiments. Overall, perturbation assignment was robust and cost-effective.

This proof-of-principle iPS2-sci-seq experiment was not designed nor powered to leverage the full clone- and treatment-aware design of iPS2-seq, but served as a test bed for conventional scRNA-seq analyses. Dimensionality reduction and clustering showcased an expected majority of CMs across replicates (Figs. 1H and EV1I), subdivided into three subclusters: CM1 (expressing *NPPB*, a marker of developing CMs), CM2 (high *MYH6/MYH7* ratio, characteristic of immature CMs; Karbassi et al, (2020)), and CM3 (high *TTN*, *RYR2*, and *CACNA1C*, suggestive of greater maturity). A minority of cells were cardiac fibroblasts (CFs; *COL3A1 +*, *COL1A1 +*, *FBN1 +*) or proliferating CMs (P-CMs; *MKI67 +*, *TOP2A +*, *CENPF +*; Fig. EV1J,K).

To broadly assess transcriptional differences, we used Monocle 3 to analyze gene module expression across clusters (Fig. 1I). For perturbations with >100 nuclei, we identified the top ten most variable modules (Fig. 1J). Scrambled (SCR) controls clustered distinctly from all perturbations, which showed downregulation of maturation-associated modules (e.g., M24) and upregulation of modules linked to immature CMs (e.g., M1 for *GATA4*; M21 for *GATA4* and *CHD7*; M6 for *KMT2D*) or CFs (e.g., M25, notably for *NKX2-5*). Intriguingly, *SMAD2* silencing decreased one of the developing CM modules (M1). To pinpoint gene-level effects within variable modules, we applied *scMAGeCK* (Yang et al, 2020), leveraging its linear regression framework in a non-clone-aware mode. Analyzing genes from the top 10 variable modules, we filtered the top 20% most significantly perturbed genes (Fig. 1K). Notable findings included *HAND1*, a cardiac progenitor transcription factor, and *CACNA1C*, the cardiac L-type calcium channel, both downregulated by *NKX2-5* LoF (Fig. EV1L), in line with previous studies (Tanaka et al, 1999; Anderson et al, 2018; Hofbauer et al, 2021), while *HAND1* was upregulated by *CHD7* LoF. Other hits included regulators of sarcomere structure, signaling, and chromatin dynamics. While detailed follow-up was beyond the scope of this proof-of-concept screen, these results highlight how iPS2-sci-seq can integrate with existing pipelines to cost-effectively generate mechanistic hypotheses.

## Phenotype agnostic screening with microfluidic partitioning

Encouraged by iPS2-sci-seq results, we tested iPS2-seq with the widely adopted 10X Genomics microfluidics platform, aiming for broader impact and easier translation. Optimization experiments in hiPSCs and day 23 hiPSC-CMs (Appendix Text; Fig. EV2A,B) led to the development of "iPS2-10X-seq", enabling parallel sequencing of UCI-BCs and transcriptomes (Appendix Protocol 3). Compared to iPS2-sci-seq, this strategy yielded more reliable assignment of cells to unique shRNA perturbations (Fig. EV2C), a higher fraction of usable cells (i.e., ~59% and ~62% for hiPSCs and CMs, respectively; Fig. EV2D,E; Appendix Table S3), more cells per shRNA (Appendix Table S4), and improved detection of gene silencing (Appendix Table S5). Noting that ~7–9% of cells in this dataset contained a cassette lacking the shRNA (Appendix Text; Fig. EV2F), we also extracted such cells as useful internal controls.

Our optimization experiment indicated clear changes in clone representation during cardiac differentiation (Fig. EV2E), echoing the iPS2-sci-seq findings (Fig. 1G). To investigate this further, we performed a larger iPS2-10X-seq experiment sampling tet-treated pools at various time points: day 0 (primed hiPSCs), day 2 (mesoderm), day 6 (cardiac progenitors), and day 12 (early cardiomyocytes; Fig. 2A). Taking advantage of genetic barcodes to identify doublets, we overloaded the microfluidic chip with 33,000 cells per time point. This reduced the fraction of cells assigned to unique shRNAs (42–63%) but increased the absolute yield (6,352–8,033 cells; Appendix Table S3). Batch correction with the two earlier samples allowed us to reconstruct the full differentiation trajectory, including late CMs at day 23. The resulting dataset comprised 54 hiPSC clones, 43 of which were represented by at least 20 cells at one or more time points, covering 19 of the 23 shRNAs, including 3 clones carrying control shRNAs (1 SCR and 2 *B2M*; Appendix Table S1). Dimensionality reduction and clustering clearly separated hiPSC and mesoderm samples from later stages, allowing identification of expected cell types (Fig. 2B).

Unexpectedly, hiPSC samples formed two major clusters, each dominated by distinct sets of clones (Fig. 2C). Clones from one of these clusters, including a control shRNA clone targeting *B2M*, were generally depleted by day 23 of differentiation (Fig. 2C). To determine whether this reflected a batch effect, we annotated clones to their original genome editing pool using data from another iPS2-10X-seq experiment that examined each pool separately (Fig. 5).

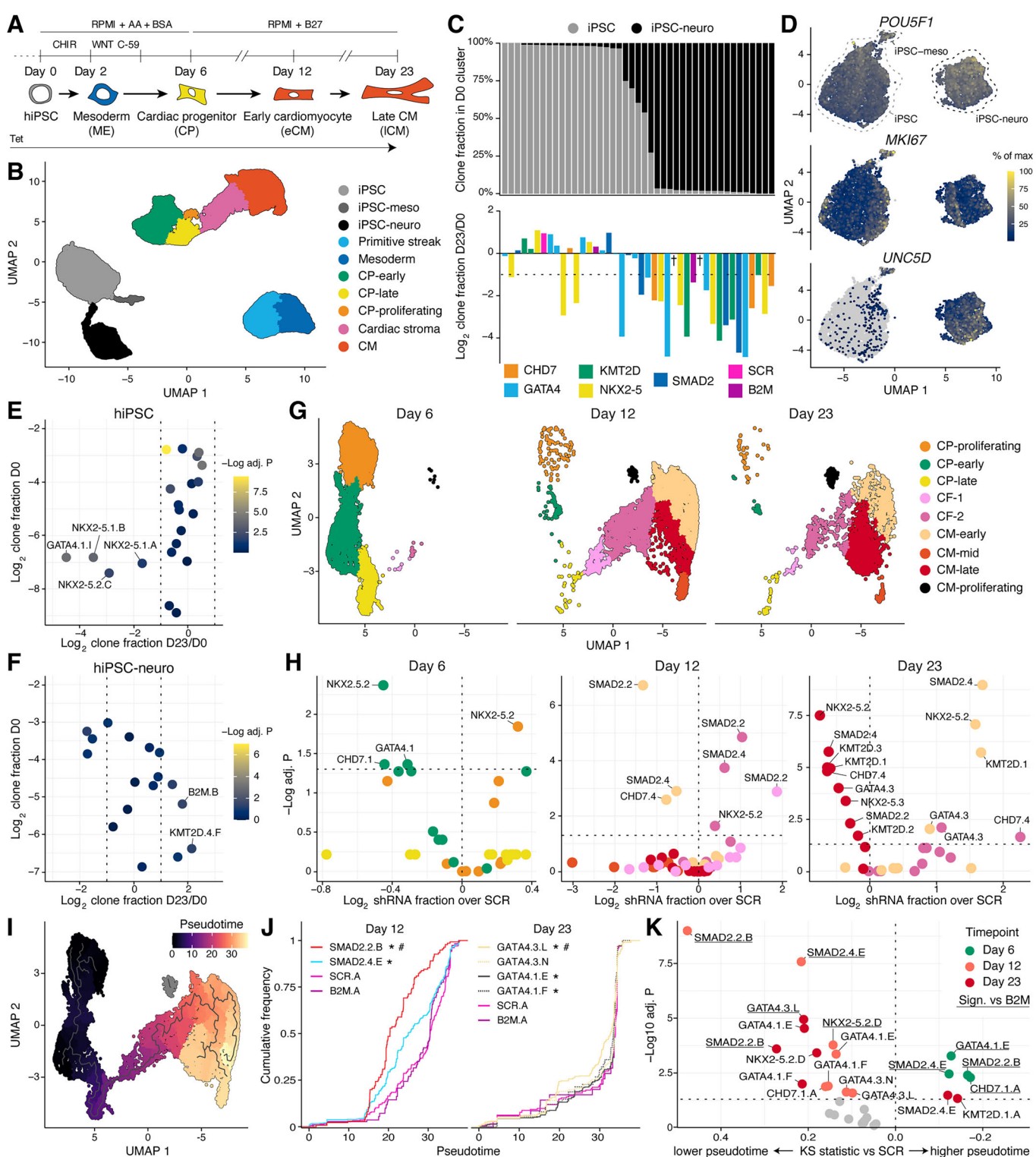

Both pools contributed 10–23% of cells mapping to the depleted cluster (Fig. EV2G), ruling out a simple batch artifact. Subclustering the two day 0 samples showed no substantial differences attributable to pre-differentiation culture conditions. The only consistent difference was a minor subpopulation of mesoderm-primed hiPSCs in cultures exposed to low-dose CHIR99021, a Wnt

pathway activator, for 24 h (Fig. EV2H). In contrast, differential gene expression revealed that clones in the depleted cluster expressed high levels of neuroectodermal transcripts (e.g., *UNC5D*, *ZIC1*, *SIX3*, *GBX2*, *OLIG3*, and *NEUROG3*) despite retaining markers of self-renewal (e.g., *POU5F1/OCT4*) and proliferation (e.g., *MKI67*; Figs. 2D and EV2I,J).

**Figure 2.   iPS2-10X-seq dissects developmental gene function from pluripotency biases.**

(A) iPS2-10X-seq can elucidate gene function at various developmental steps, such as during cardiomyogenesis. (B) Dimensionality reduction and clustering of all datapoints from the time course described in (A). (C) Correlation between the transcriptional state of hiPSC clones and their developmental enrichment or depletion. (D) Subclustering of hiPSCs links clonal neuroectoderm priming to inefficient cardiogenesis (Figs. EV2G–J). (E) Enrichment/depletion analysis limited to hiPSC clones with an unbiased starting transcriptional state; adj. *P* by Fisher test with Benjamini–Hochberg (B–H) correction. Here and elsewhere, numbers and letters indicate different shRNAs and clones, respectively. (F) As in (E), but for neuroectoderm-primed hiPSC clones. Adj. *P* by Fisher test with B–H correction. (G) Subclustering of samples spanning differentiation from cardiac progenitors to cardiomyocytes. (H) Cluster representation changes associated to selected shRNAs (filtered by clone/cluster size); adj. *P* by Fisher test vs. SCR, B–H correction, significance threshold of 0.05. Dots color code matches (G). (I) Pseudotime trajectory quantifying clonal progression along the cardiac differentiation path. (J) Cumulative pseudotime distribution of selected hiPSC clones. */# = adj. *P* < 0.05 vs. SCR/*B2M*, respectively. (K) Pseudotime alterations for selected clones (filtered by numerosity); adj. *P* by two-sided KS) test vs. SCR, B–H correction, significance threshold of 0.05 (underlined clones also significant vs. *B2M*). Directionality from on one-sided KS tests on *x* axis.

To account for these transcriptional differences, we stratified hiPSC clones into "unbiased" and "neuroectoderm-primed" (iPSC-neuro) and repeated the enrichment/depletion analysis. For unbiased clones, results were clearer: only those expressing shRNAs against *GATA4* or *NKX2-5* were significantly depleted from day 6 onward, consistent with roles in cardiac progenitors (Figs. 2E and EV2K). In contrast, iPSC-neuro clones showed more variable trends, limited by lower cell numbers (Fig. 2F). We concluded that genome editing may predispose a subset of hiPSCs toward neuroectodermal lineage bias, reducing their capacity for mesodermal induction. Crucially, iPS2-seq enables detection and control of this effect through clone-level transcriptional profiling. Accordingly, we excluded neuroectoderm-primed clones from downstream analyses to enhance data quality.

We developed two analytical frameworks for iPS2-seq data. First, a cell type representation analysis using multiple comparison-corrected Fisher tests to compare the (sub)clustering distribution of control and knockdown cells, applying filters to ensure adequate statistical power. We applied this test to early time points, and unbiased hiPSCs showed no significant differences, as expected due to limited treatment duration (4 days) and lack of expression for some targets (e.g., *GATA4*, *NKX2-5*). At day 2, subclustering revealed two major cell types: primitive streak (PS; *NODAL +*, *TBXT/Brachyury +*, *EOMES +*) and migrating mesoderm (MES; *PDGFRA +*, *MESP1 +*, *TBX6 +*), further stratified by proliferation state (Fig. EV2L). Surprisingly, a SCR shRNA clone showed a mild delay in PS-to-mesoderm transition versus both *B2M* knockdown and no-shRNA controls (Fig. EV2M). This did not affect downstream differentiation and highlights SCR as a conservative reference. At the same stage, *CHD7* and *NKX2-5* knockdowns showed slight enrichment in proliferative MES cells, the latter despite low *NKX2-5* expression at this stage. Together, these findings underscore three key lessons: (1) clonal variability is high during rapid cell fate transitions; (2) over-reliance on a single control can be misleading; and (3) clustering results should be interpreted with caution.

We next analyzed later stages of CM differentiation, during which all target genes are expressed. Subclustering from day 6 to 23 revealed a main trajectory encompassing cardiac progenitors (CPs; *PDGFRA +*, *TMEM88 +*, and/or *GATA4 +*, with some proliferative cells marked by *TOP2A* and *CENPF*), CFs, and CMs at different maturation stages — comparable to iPS2-sci-seq results (Figs. 2G and EV2N,O). A small subset of CMs remained proliferative. All controls showed similar cell type distributions, while knockdowns of all target genes led to distinct alterations (Fig. EV2P). To assess robustness, we repeated subclustering

representation analysis at the individual shRNA level (Fig. 2H). The clearest effect was observed for *SMAD2*: at day 12, two shRNAs increased CFs at the expense of early CMs; at day 23, the same shRNAs reduced the ratio of mature to immature CMs. We also noted mild but consistent depletion of mature CMs at day 23 for shRNAs targeting *KMT2D* (3 shRNAs), *NKX2-5* (2 shRNAs), and *GATA4* (1 shRNA, also associated with increased cardiac stroma and early CMs). All findings were consistent when using *B2M* or no-shRNA cells as alternative controls.

Our second analytical framework for iPS2-seq goes beyond discrete clustering by leveraging continuous variables to compare cumulative distributions between controls and perturbations using a two-tailed, multiple comparison-corrected Kolmogorov–Smirnov (KS) test (directionality is inferred with two one-sided KS tests). This cell-level approach provides sufficient power to detect clone-specific phenotypes without grouping by shRNA or gene target.

We applied this to later stages of CM differentiation, comparing pseudotime along the differentiation trajectory (Fig. 2I). This identified five clones with significant delays at day 12 or 23 compared to both SCR and *B2M* controls (Fig. 2J,K). Two clones carried distinct shRNAs against *SMAD2*, consistent with cell type representation data. Interestingly, these same clones showed accelerated differentiation at day 6, which rules out an early delay and pinpoints a defect in CP-to-CM specification between days 6 and 12. Additional delayed clones included one targeting *NKX2-5* (day 12) and one targeting *GATA4* (day 23). Grouped gene-level analyses confirmed these findings (Fig. EV2Q). Altogether, these results show that iPS2-10X-seq supports phenotype-agnostic screening during differentiation, capturing perturbation effects on both cell fate and pseudotime while controlling for initial transcriptional biases across clones.

## Multi-omic dissection of clonal variability

The discovery of clonal transcriptional variability in genome-edited hiPSCs raises fundamental questions regarding its stability and origin. We first assessed its persistence in time by comparing early-passage hiPSCs (p3, used for previous experiments) to the same pools after 5 and 10 passages (middle and late; Fig. 3A). DNA sequencing of UCI-BCs and reverse transcription quantitative PCR (RT-qPCR) for neuroectoderm and pluripotency markers revealed substantial shifts in clonal composition: neuroectoderm-primed clones were progressively depleted, matched by reduced expression of markers such as *ZIC1*, while *POU5F1/OCT4* levels remained stable (Figs. 3B,C and EV3A,B). Several unbiased clones expanded over time, one reaching ~22% abundance, highlighting the importance of early passage screening.

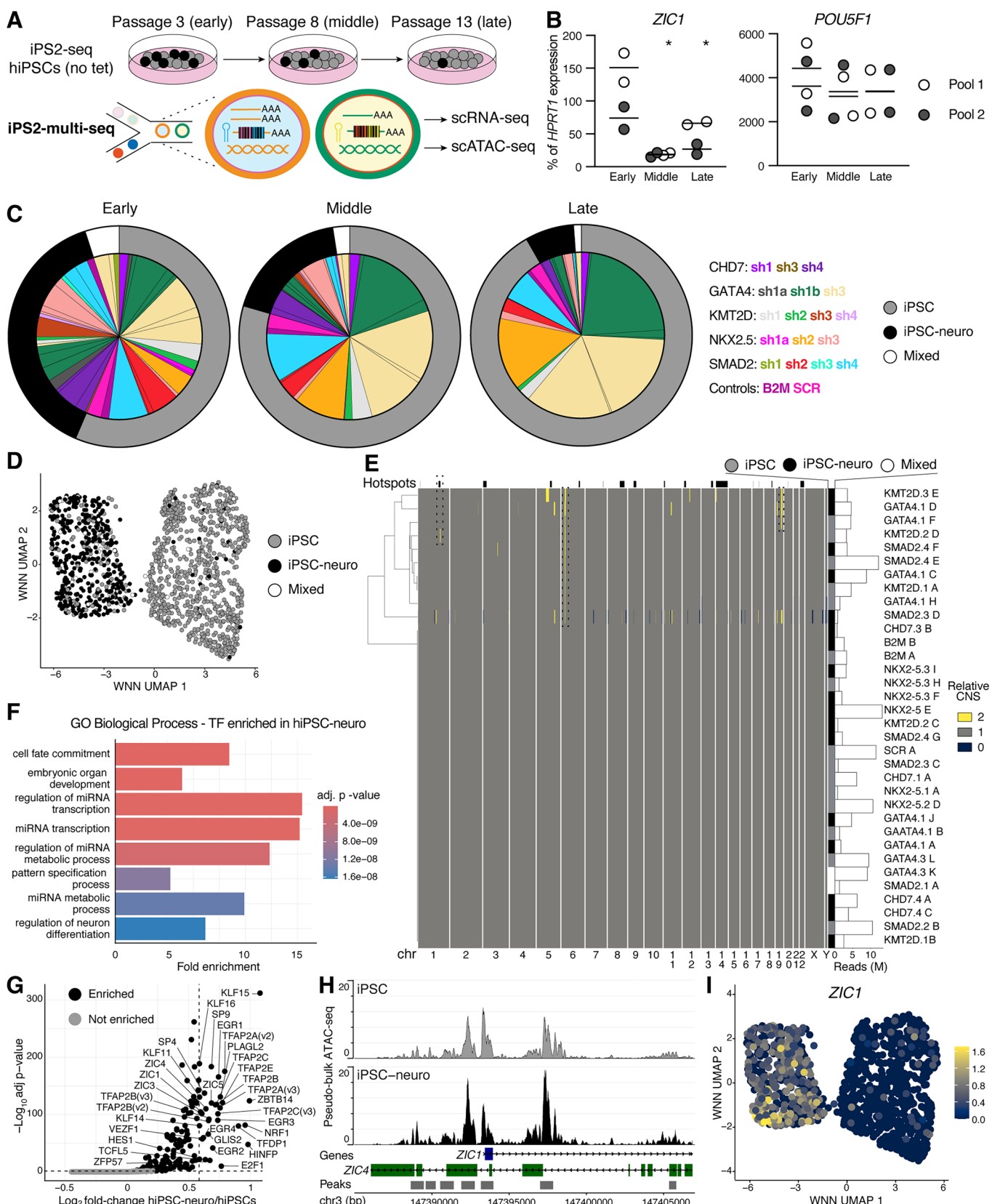

◀ **Figure 3. Dissecting the epigenetic basis of hiPSC clonal biases by iPS2-multi-seq.**

(A) iPS2-multi-seq can reveal the epigenetic basis of transcriptional phenotypes, such as hiPSC clonal biases. (B) RT-qPCR of key neuroectoderm and pluripotency regulators at various passages of iPS2-seq genome-edited hiPSCs. N = 2 cultures (the mean is indicated), * = adj. P 0.036 vs. early by two-way repeated measures (RM) ANOVA with Holm–Šídák's multiple comparisons. (C) Clonal composition by UCI-BC DNA-seq across passages, ordered by clonal bias type (Fig. 2C). (D) Weighted Nearest Neighbor (WNN) plot of integrated nuclear scRNA-seq and scATAC-seq, labeled as in (C). (E) Hierarchical clustering of clonal CNVs inferred from aggregated scATAC-seq signal. Bias type is annotated (no correlation). Known hiPSC mutational hotspots and examples of polyclonal CNVs are marked. (F) Gene ontology enrichment of chromatin regions is more accessible in iPSC-neuro clones. One-sided Fisher's exact test obtained by *enrichGO* function of the clusterprofiler R package. (G) Volcano plot of TF motif enrichment in iPSC-neuro vs. iPSC ATAC peaks. Significance threshold of 0.05. (H) Aggregated chromatin accessibility tracks at the *ZIC1* locus, showing increased accessibility in iPSC-neuro. (I) Expression of *ZIC1* projected on WNN from (D), confirming activation in iPSC-neuro clones. Source data are available online for this figure.

To probe the epigenetic and genetic underpinnings of this variability, we piloted iPS2-multi-seq, a clone-aware, single-cell multi-omic method integrating nuclear RNA-seq and assay for transposable accessible chromatin by sequencing (ATAC-seq) from the same nuclei (Fig. 3A). Although recovery of UCI-BCs from multiomes was lower and noisier than in RNA-based methods, likely due to nuclear RNA loss during permeabilization (Fig. EV3C), we could assign clonal IDs for 1662 out of 6051 cells, with clone proportions consistent with DNA-seq estimates (Fig. EV3D). Individual clustering of transcriptomic and epigenomic profiles reproduced the iPSC versus iPSC-neuro dichotomy observed previously (Fig. EV3E), as did their integration (Fig. 3D), confirming that clonal bias is robustly detectable from nuclear RNA and chromatin accessibility.

To test for potential genetic causes linked to genome instability, we leveraged the ATAC-seq data to estimate copy number variation (CNV) using *epiAneufinder*. Few recurrent CNVs were detected across clones, including known hotspots for hiPSC instability, and these were not associated with neuroectoderm bias (Fig. 3E), suggesting that CNVs are unlikely to explain the observed phenotype.

We next explored the epigenetic basis of clonal divergence. ATAC-seq differential peak analysis revealed strong enrichment for neuroectoderm transcription factor (TF) motifs in iPSC-neuro clones (Fig. 3F,G). Several of these TFs were transcriptionally upregulated and showed increased promoter accessibility (Fig. EV3F), most notably *ZIC1* (Fig. 3H,I). Chromatin accessibility and expression of downstream targets such as *UNC5D* were also increased (Fig. EV3G,H), consistent with prior scRNA-seq data (Fig. 2D). Other TFs such as *TFAP2C* showed increased expression without changes in promoter accessibility, pointing to potential epitranscriptional regulation. Pluripotency markers remained unaffected, and *ZNF528* and *ZNF138* emerged as candidate regulators of the iPSC-neuro and unbiased iPSC epigenetic states, respectively (Fig. EV3G,H). In summary, these observations indicate that clonal variability among genome-edited hiPSCs has a primary epigenetic basis driven by enhanced activity of neuroectoderm TFs. More broadly, these experiments highlight the potential of iPS2-multi-seq in exploring the epigenetic basis of clonal phenotypes in iPSC models.

## Clonal identity as a determinant of differentiation outcomes

The discovery that iPSC-neuro clones are impaired in mesodermal differentiation prompted us to ask whether these same clones also display altered potential within neuroectoderm derivatives. To address this, we performed an iPS2-10X-seq experiment using a well-established protocol for forebrain organoid differentiation (Velasco et al (2019); Fig. 4A). We included control and tet-treated conditions to both investigate clonal differentiation potential in the absence of gene knockdown and confirm the functionality of iPS2-seq in neuronal tissues.

Differentiation followed the expected timeline and morphological milestones, although we observed consistent emergence of both cortical and retinal territories in WTC-11-derived organoids (Figs. 4B and EV4A). scRNA-seq at day 30 verified the presence of various cortical progenitors and neurons, retinal precursors and their derivatives, neural crest subtypes, and an unexpected population of non-neuronal, fibroblast-like cells (Figs. 4C and EV4B,C).

Clone representation analysis revealed that, unlike during cardiac differentiation, iPSC-neuro clones were neither globally nor strongly depleted in brain organoids (Fig. 4D–F). However, the two clonal classes followed strikingly different differentiation trajectories: unbiased iPSCs contributed retinal populations, cortical progenitors, and subcortical neurons, whereas iPSC-neuro clones primarily gave rise to cortical neurons (with very few remaining progenitors), neural crest cells, and the unexpected fibroblast population (Fig. 4G). Notably, tet treatment substantially reduced the representation of retinal populations derived from unbiased iPSCs, while iPSC-neuro-derived lineages were largely unaffected (Fig. 4G).

Although our perturbation library was not tailored to neural contexts, three targeted genes (*SMAD2*, *KMT2D*, and *CHD7*) are expressed in neuroectoderm lineages and linked to syndromes that can include both CHD and neurodevelopmental defects (Van Laarhoven et al, 2015; Wang et al, 2011; Vissers et al, 2004). Only *SMAD2* knockdown led to reduced clonal representation (Fig. EV4D), while all three perturbations had marked effects on cell type composition (Fig. EV4E,F). Retinal precursors were consistently depleted, with *CHD7* knockdown also impairing retinal pigment epithelial fate, consistent with the ocular defects seen in CHARGE syndrome caused by *CHD7* mutations (Krueger and Morris 2022). These shifts were offset by expansions of fibroblasts (*CHD7*), neural crest (*KMT2D*), and cortical progenitors (*SMAD2*). While outside the primary cardiac focus of this study, these findings highlight the broader utility of iPS2-seq in disease-relevant neurodevelopmental contexts. More generally, these experiments demonstrate that epigenetically primed iPSC clones can bias differentiation even within the neuroectoderm, emphasizing the importance of clonal awareness in single-cell studies spanning distinct germ layer derivatives.

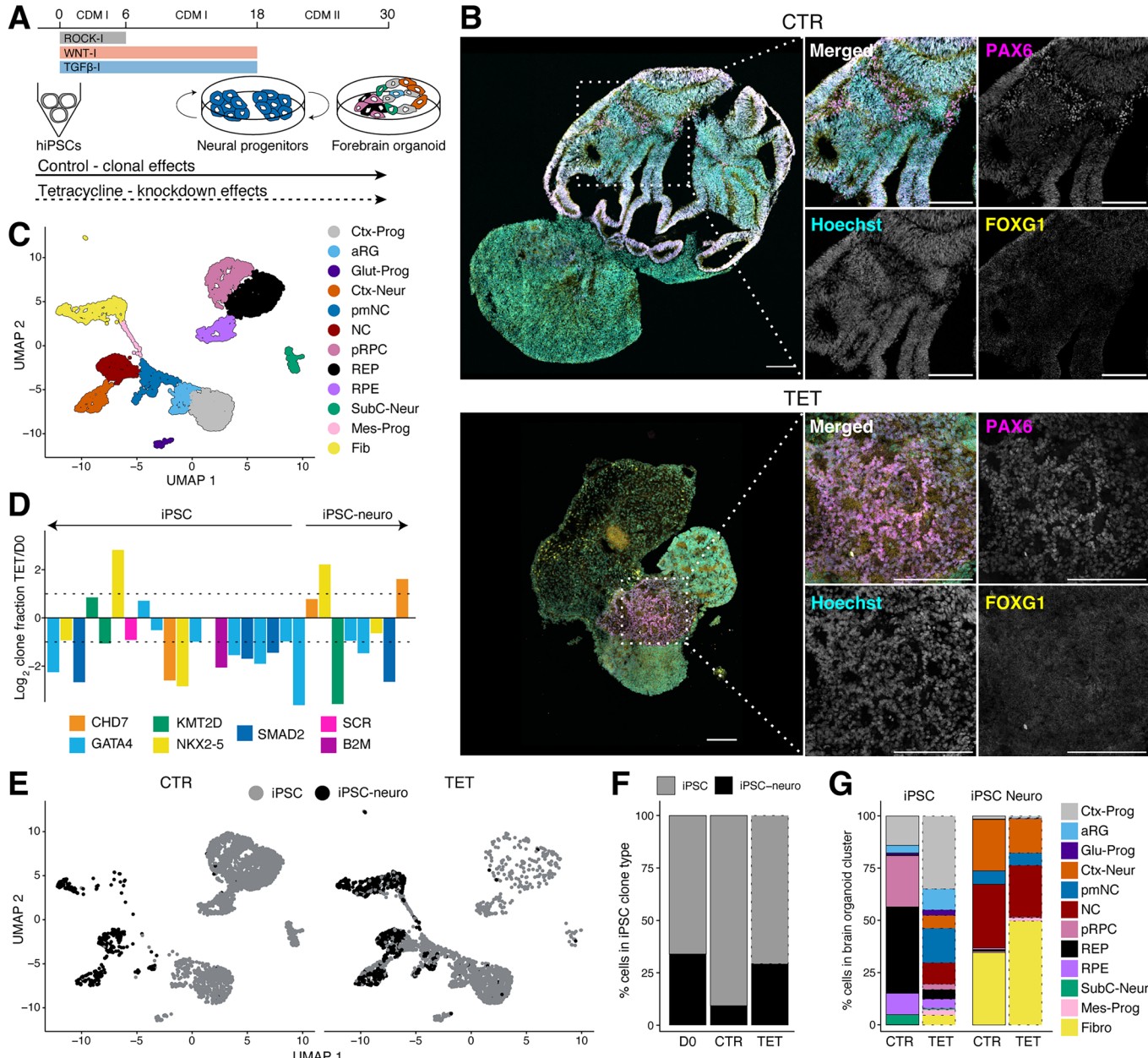

**Figure 4. hiPSC clonal biases alter cell fate in forebrain organoids.**

(A) iPS2-10X-seq can follow iPSC knockdown-independent clonal biases, exemplified in forebrain organoids. (B) Representative immunofluorescences of day 30 control and tet-treated organoids. Scale bars: 150 μm. (C) Dimensionality reduction and clustering of both iPS2-10X-seq datasets. Ctx-Prog cortical progenitors, aRG apical radial glia, Glut-Prog glutamatergic progenitors, Ctx-Neur cortical neurons, pmNC pre-migratory neural crest, NC neural crest, pRPC proliferative retinal progenitor cells, REP retinal epithelial progenitors, RPE retinal pigmented epithelium, SubC-Neur sub-cortical neurons, Mes-Prog mesoderm progenitors, Fibro fibroblast. (D) Clone enrichment/depletion in tet-treated organoids vs. hiPSCs, sorted by clonal bias state as in Fig. 2C. (E) As in (C), shown separately for control and tet-treated samples, with cells color-coded by clonal bias. (F) Clone type changes during control and tet-treated organoid differentiation, based on starting bias state. (G) Cluster representation changes in organoids associated with starting bias state and tet treatment. Source data are available online for this figure.

## Clone- and treatment-aware scRNA-seq screening in organoids

Encouraged by the ability to control clonal variability in neural organoids, we put iPS2-10X-seq to the test in a similarly complex system more directly suited to functionally interrogate our CHD gene list: cardiac organoids (cardioids). This model recapitulates early left ventricular

development (Hofbauer et al, 2021), generating multicellular tissues within a week that break radial symmetry to form inner cavities reminiscent of the early ventricular chamber. To fully exploit the clone-aware design of iPS2-seq, we compared again tet-treated cardioids to matched controls cultured without tet (Fig. 5A). This experimental strategy bypasses the aforementioned issue of relying on a small number of controls and the associated risk of being misled by potential clonal outliers.

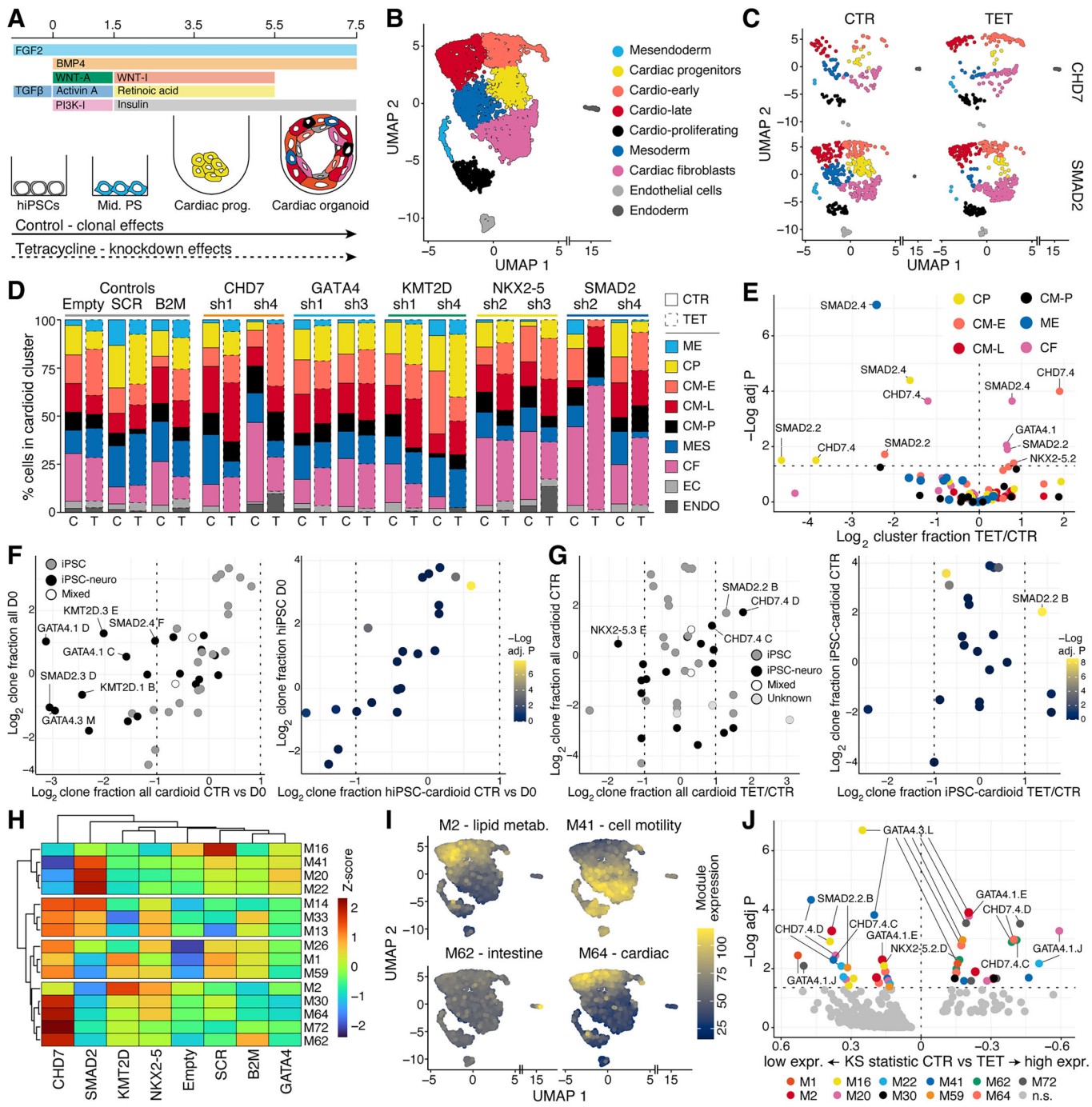

**Figure 5.   Clone- and treatment-aware pooled screening in cardiac organoids.**

(**A**) Clonal biases in organoid morphodifferentiation can be controlled by comparing single-cell transcriptomes for the same clones in paired control (no tet) and tet-treated conditions, exemplified in cardioids. (**B**) Dimensionality reduction and clustering for control and tet-treated day 7.5 left ventricle cardioids. (**C**) As for (**B**), but focusing on cells with gene perturbations to visualize clustering changes. (**D**) Paired assessment of clustering changes for selected shRNAs in control (**C**) vs. tet-treated (**T**) cells. (**E**) Statistical analysis of clustering changes for selected shRNAs (filtered by numerosity); adj. *P* by Fisher test comparing tet vs. control, B–H correction, significance threshold of 0.05. (**F**) Clonal enrichment/depletion in control cardioids vs. the starting iPSC pool, show for all clones (left, color-coded by epigenetic status) and unbiased clones only (right). Adj. *P* by Fisher test with B–H correction. (**G**) As in (**F**), but comparing tet-treated cardioids to their matched controls. Adj. *P* by Fisher test with B–H correction. (**H**) Hierarchical clustering of 15 most variable gene modules based on Z-score differences across perturbations. (**I**) Aggregated expression and functional annotation of selected gene modules (Fig. EV5J; Appendix Table S6). (**J**) Clone-associated alterations in gene module expression (clones filtered by numerosity); adj. *P* by two-sided KS test comparing tet vs. control, B–H correction, significance threshold of 0.05 (labels shown for clones with adj. *P* < 0.01; directionality from one-sided KS tests on *x* axis).

We examined this approach in day 7.5 cardioids, first confirming that tet treatment throughout did not impair morphogenesis or CM yield (Fig. EV5A–D). This supports the expectation that iPS2-seq captures cell-autonomous effects, whereas multicellular morphogenesis may be buffered by non-perturbed neighbors. To offset the increased cost of scRNA-seq when analyzing both control and tet-treated cardioids, we used cell multiplexing oligos (CMOs) to label the two conditions from each hiPSC pool, pooled them, and profiled 23,000 cells in a single 10X run. We obtained 3734 control and 3918 tet-treated cells, ~70% of which were assignable to individual shRNAs, recovering 51 clones (28 with >20 cells).

Dimensionality reduction and clustering revealed the expected cellular diversity in cardioids, including progenitors (EOMES+ mesendoderm, HAND2 + MES, ESRRG + CPs), CMs at various maturation stages, CFs, ECs (CDH5 + , ECSCR + ), and a minor endoderm fraction (APOB + , FOXA2 + ; Figs. 5B and EV5E,F). Cell type proportions differed visibly between control and tet-treated samples for several knockdowns, especially SMAD2 and CHD7 (Fig. 5C). These shifts were statistically significant both at the gene (Fig. EV5G) and shRNA level (Fig. 5D,E). Notably, two SMAD2 shRNAs caused strong depletion of MES, CPs, and/or early CMs, with a corresponding rise in CFs, echoing effects seen in monolayer hiPSC-CMs.

Importantly, we observed marked differences in cell type distributions among tet-unexposed cells expressing different inducible shRNAs, including SCR, B2M, and no shRNA controls (Fig. 5D). Given the previously demonstrated tight control of the tet-ON system in both hiPSCs and derivatives (Bertero et al, 2016), these differences are unlikely due to leaky shRNA expression. Instead, they reflect clonal variability independent of gene knockdown. This was confirmed by benchmarking analyses: comparing perturbations to unmatched controls like SCR or B2M introduced false positives (i.e., significant effects only vs. SCR) and false negatives (i.e., effects missed vs. B2M; Fig. EV5H,I). We also used this control/treatment design to revisit global clonal representation and found that iPSC-neuro clones were depleted in cardioids even without tet treatment, consistent with their poor mesodermal competence (Fig. 5F). Comparing tet-treated cells with their untreated counterparts thus enabled robust correction for this epigenetic bias (Fig. 5G). Together, these findings underscore the value of clone-aware, internally controlled designs enabled by iPS2-seq, especially in multicellular organoids where subtle transcriptional drifts can lead to major cell fate differences.

To uncover subtler effects beyond clustering, we analyzed the most variable gene modules, as pseudotime inference is less reliable in organoids due to complex, branched trajectories. Several modules were altered by SMAD2 and CHD7 knockdown, with additional perturbations linked to KMT2D and NKX2-5 (Fig. 5H). In contrast, GATA4 targeting had little effect, clustering near all controls. Gene ontology enrichment allowed tentative annotation of each module to specific cell types or processes (Figs. 5I and EV5J; Appendix Table S6). To quantify perturbation effects, we ranked cells by median module expression and compared cumulative distributions between control and treated cells. This analysis, performed for the 15 most variable modules at both gene and clone levels, revealed significant changes in 11 modules (Figs. 5J and EV5K). Notably, SMAD2 knockdown reduced cardiac gene expression and lipid metabolism (a hallmark of CM maturation;

Karbassi et al (2020)) while increasing markers of motility and collagen synthesis typical of CFs. CHD7 was linked to repression of endoderm-associated modules, a role masked in monolayer CM differentiation but evident in organoids due to their permissive cell fate diversity. These results confirm that iPS2-10X-seq enables rigorous, clone- and treatment-aware screening of cell type composition and gene module regulation even in complex, self-organizing organoids.

## Validation of screening hits via arrayed polyclonal and clonal analysis

SMAD2 emerged as a top hit in both monolayer and cardioid iPS2-seq screens, yet its role in human cardiac development remains unclear. We therefore selected it for follow-up validation and to showcase an additional application of iPS2-seq: analysis of individual perturbations in arrayed, polyclonal formats while still controlling for clonal variability (Fig. 6A).

To this end, we designed a streamlined protocol for multiplexing control and tet-treated samples within a single microfluidic lane using antibody-based hashing (Fig. EV6A). This strategy, which we term iPS2-CITE-seq, leverages Cellular Indexing of Transcriptomes and Epitopes (CITE) barcoded antibodies for cost-effective single-cell analysis of arrayed perturbations in polyclonal populations, facilitating robust assessment of reproducibility. It also supports optional multi-omic measurement of protein levels, expanding the readout beyond RNA and perturbation identity alone.

To validate the approach, we leveraged TTN-mEGFP reporter hiPSCs to generate polyclonal iPS2-seq cells with SMAD2 or B2M shRNAs, each composed of clones carrying five distinct UCIs (Fig. EV6B). These lines were used to generate cardioids, which showed that SMAD2 knockdown significantly reduced their size (Fig. EV6C). We performed iPS2-10X-seq on pooled organoids across all four experimental conditions (control/tet, SMAD2/B2M shRNA) using a single microfluidic channel (Fig. EV6D). Transcriptomic data confirmed knockdown, and CITE-seq for B2M further validated protein-level depletion (Fig. EV6E,F).

Cell type composition analysis revealed that SMAD2-silenced clones exhibited consistent depletion of CMs and CPs, with a reciprocal increase in CFs (Fig. EV6G,H), matching prior observations. Unexpectedly, these clones also gave rise to a population of epicardial-like cells, suggesting a non-cell-autonomous role of SMAD2 that went unappreciated in pooled screens. These experiments demonstrate that iPS2-CITE-seq enables efficient follow-up of pooled screen hits through arrayed polyclonal analyses.

We next isolated genotyped SMAD2 and B2M iPS2-seq clones (both heterozygous and homozygous; Fig. EV6I) and generated cardioids from each. Homozygous SMAD2 shRNA expression resulted in the smallest organoids (Figs. 6B and EV6J), and reduced expression of TTN-mEGFP (Fig. 6C,D). Of note, this clone yielded larger organoids under no-tet conditions and, like the B2M control, showed a few CNVs (Appendix Table S7), underscoring the importance of matched controls to account for clonal differences.

scRNA-seq analysis of this clone confirmed SMAD2 knockdown-associated depletion of CMs and CPs, and expansion of CFs. This experiment revealed an even stronger epicardial signature (~11% of cells), expressing markers such as PDPN, TBX18, WT1, and SFRP5 (Figs. 6E,G and EV6K).

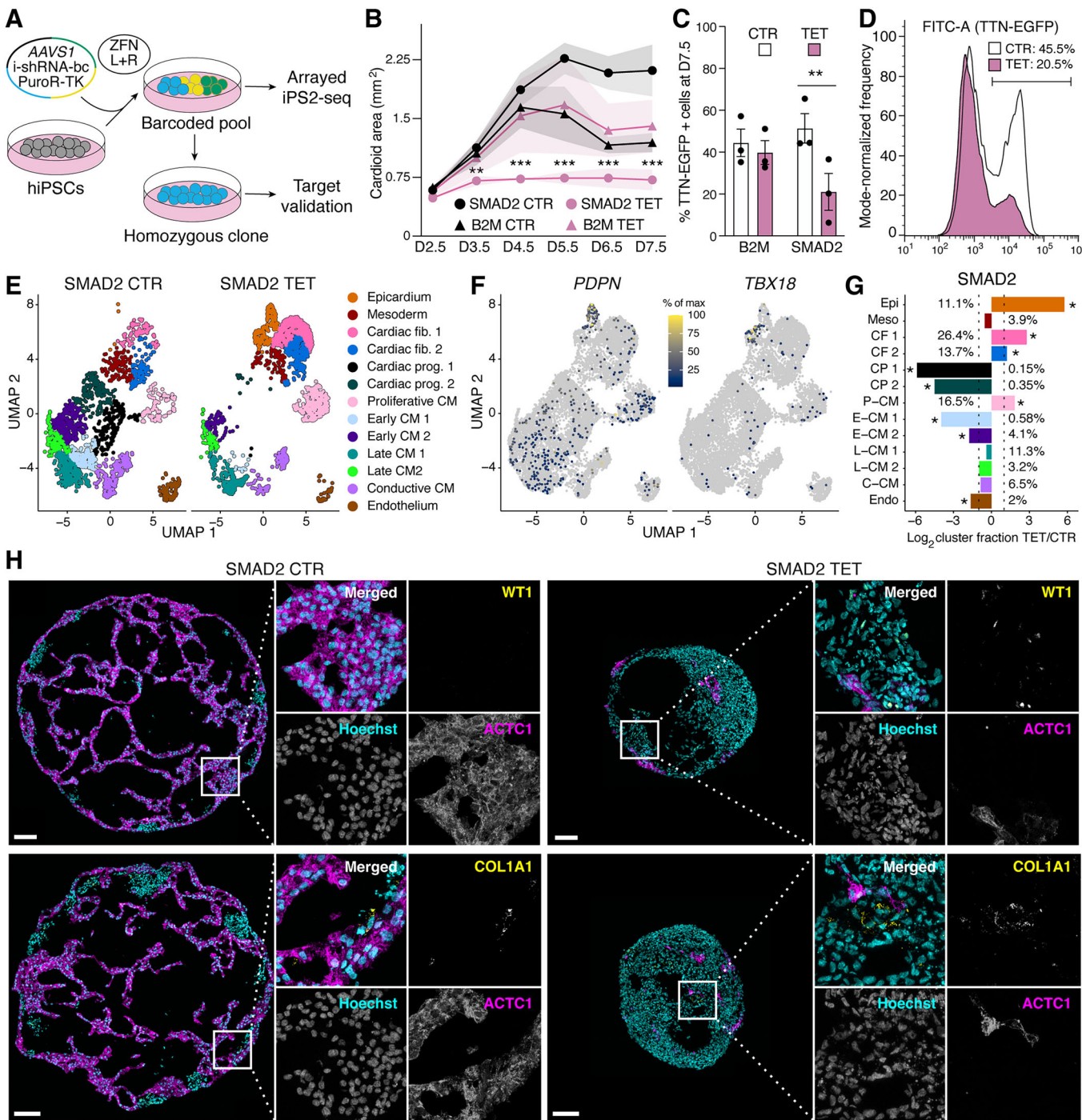

**Figure 6. *SMAD2* knockdown impairs cardiac organoid morphodifferentiation.**

(**A**) iPS2-seq supports both arrayed screening in polyclonal pools (Fig. EV6) and targeted validation in clones. (**B**) Time course analysis of cardioid size in iPS2-seq *SMAD2* and *B2M* homozygous clones in control or tet-treated conditions. Here and in (**C**), $N = 3$ cultures. **, *** = adj. *P* of 0.006, <0.001 vs. control by two-way RM ANOVA with Holm–Šídák's multiple comparisons. Day 7.5 cardioids were analyzed in the rest of the figure. (**C**) Flow cytometry analysis of cardiomyocyte differentiation efficiency based on a TTN-mEGFP knock-in reporter; ** = adj. *P* of 0.0018, by two-way RM ANOVA with Holm–Šídák's multiple comparisons. (**D**) Representative log fluorescence flow cytometry histograms for *SMAD2* knockdown. (**E**) Dimensionality reduction and clustering of *SMAD2* control and tet-treated cardioids, shown separately. (**F**) Epicardial markers expression projected on the dimensionality reduction from (**E**). (**G**) Quantification of clustering changes; * = adj. *P* < 0.05 by Fisher test comparing tet vs. control, B–H correction. Specifically, in order of appearance from top to bottom, $P = 7.5 \times 10^{-58}$ (* Epi), $7.32 \times 10^{-104}$ (* CF 1), $1.04 \times 10^{-6}$ (* CF 2), $5.71 \times 10^{-63}$ (* CP 1), $1.36 \times 10^{-45}$ (* CP 2), $2.03 \times 10^{-29}$ (* P-CM), $2.66 \times 10^{-48}$ (* E-CM 1), $4.02 \times 10^{-27}$ (* E-CM 2), $2.19 \times 10^{-2}$ (* Endo). (**H**) Representative immunofluorescence for cardiomyocytes (ACTC1), epicardial cells (WT1), and cardiac fibroblasts (COL1A1). Scale bars: 100 μm. Source data are available online for this figure.

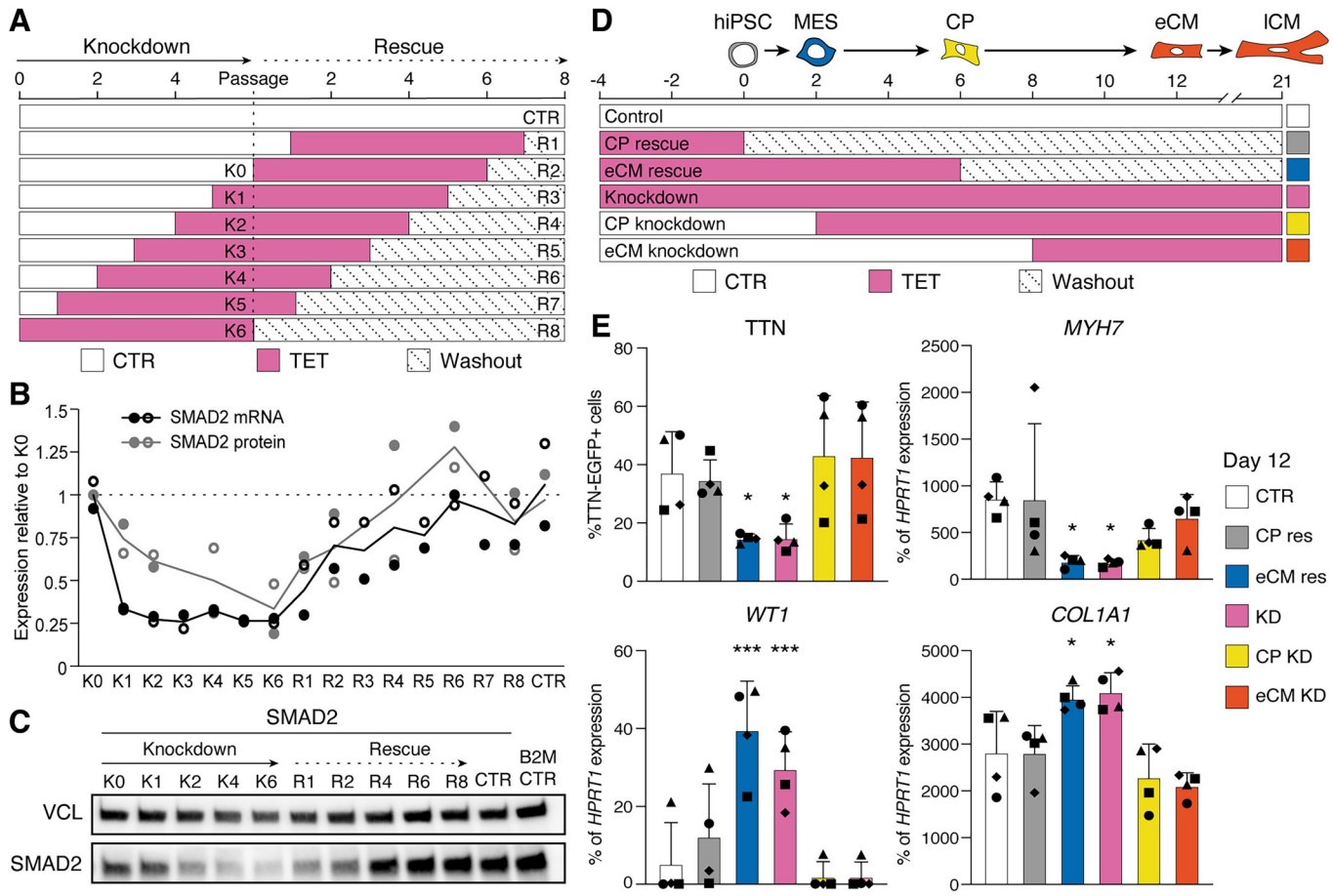

**Figure 7. SMAD2 is required for cardiac progenitor patterning and specification.**

(**A**) iPS2-seq enables inducible and reversible gene knockdown (K) and rescue (R), exemplified here in iPSCs. (**B**) Time course of *SMAD2* silencing and re-expression in an iPS2-seq homozygous clone examined according to the strategy of (**A**). mRNA analyses by RT-qPCR, and protein quantification by western blot. N = 2 cultures. (**C**) Representative western blot quantified in (**B**). VCL (vinculin) serves as a loading control; *B2M* iPS2-seq homozygous clone in control conditions rules out inducible system leakiness. (**D**) Experimental design to dissect the developmental requirement for *SMAD2* during monolayer cardiac differentiation (Fig. 2A). K and R windows are 4 days based on (**A–C**). (**E**) TTN-mEGFP reporter expression (top left) and mRNA quantification for markers of cardiomyocytes (*MYH7*), epicardial cells (*WT1*), and cardiac fibroblasts (*COL1A1*), at day 12 of differentiation. N = 4 differentiations (symbols); *, *** = adj. *P* vs. control by one-way RM ANOVA with Dunnett's multiple comparisons. Specifically, in order of appearance, *P* = 0.04 (* *TTN*), 0.048 and 0.043 (* *MYH7*), <0.001 (*** *WT1*), 0.041 and 0.021 (* *COL1A1*). Error bars represent mean ± SD. Source data are available online for this figure.

Immunofluorescence further demonstrated that *SMAD2*-silenced cardioids were smaller, expressed less ACTC1, more COL1A1, and displayed discrete WT1+ cell clusters not present in controls or *B2M* knockdown organoids (Fig. 6H). These phenotypes were confirmed by RT-qPCR in bulk samples from biological triplicates (Fig. EV6L).

## Stage-specific dissection of gene function

The emergence of epicardial and stromal populations in *SMAD2* knockdown cardiac organoids prompted us to examine when *SMAD2* activity is required during cardiac development. Given its well-known role downstream of Activin, Nodal, and TGFβ signaling pathways in pluripotent and early mesendodermal stages (Bertero et al, 2018a; Bertero et al, 2015), we first asked whether its silencing affected pluripotency exit or PS induction. RT-qPCR in

day 1.5 organoids showed comparable downregulation of pluripotency genes and induction of PS markers in *SMAD2* knockdown conditions (Fig. EV6M), suggesting early differentiation was largely intact.

We next leveraged the inducibility and reversibility of iPS2-seq to map stage-specific requirements for *SMAD2*. RT-qPCR and western blot showed that *SMAD2* could be robustly silenced within four days of tet treatment, and re-expressed with similar kinetics upon tet washout (Fig. 7A–C). Up to 6 days of *SMAD2* knockdown had no effect on pluripotency marker expression, confirming that this extent of silencing did not disrupt hiPSCs (Fig. EV7A).

To dissect functional timing, we turned to 2D cardiac differentiation, which proceeds more gradually than organoid-based protocols and supports precise perturbation windows (Fig. 7D). We designed four conditions: (1) early knockdown with rescue at CP stage; (2) late knockdown with rescue at early

cardiomyocyte (eCM) stage; (3) knockdown from CP onward; and (4) knockdown from eCM onward. In all cases, *SMAD2* silencing was confirmed and early mesoderm and CP differentiation remained unaffected (Fig. EV7B–D).

CM maturation was assessed by TTN-mEGFP expression, which was consistently reduced by *SMAD2* knockdown. The phenotype was not rescued by eCM-stage *SMAD2* restoration, but was fully prevented by CP-stage rescue (Figs. 7E and EV7E,F), indicating that *SMAD2* is required specifically between the CP and eCM stages. Notably, knockdown only at the CP stage was insufficient to replicate the phenotype, suggesting that early *SMAD2* activity (e.g., during mesoderm induction) primes cells for subsequent CP-stage competence.

Transcript analysis confirmed that reduced TTN-mEGFP reflected decreased *TTN* and *MYH7* expression, alongside upregulation of epicardial (*WT1*, *TBX18*) and stromal (*COL1A1*, *COL5A2*) markers (Figs. 7E and EV7E,F). These results underscore the stage-specific requirement for *SMAD2* in promoting CM fate and suppressing alternative lineage trajectories.

In sum, this study showcases iPS2-seq as a multi-purpose platform for pooled LoF screening, arrayed and multiplexed hit validation, and mechanistic dissection of gene function across developmental time.

## Discussion

We report iPS2-seq, a technology optimized for robust phenotype-agnostic pooled screens in hPSC models. Key features include the ability to follow individual hiPSC clones to account for clonal genetic and epigenetic variability, compatibility with both microfluidics and split-pool scRNA-seq protocols to capture differentiation heterogeneity and asynchrony—including multi-omics strategies, isogenic engineering to prevent transgene silencing or variability, post-transcriptional LoF perturbations that bypass genotoxicity, and stage-specific, reversible perturbation. The method can be applied directly to any hiPSC line without prior genetic modification, enabling rapid deployment across diverse models. Paired with its dedicated analysis pipeline, *catcheR*, iPS2-seq democratizes functional single-cell genomics in stem cell laboratories.

The power of scRNA-seq unbiased phenotyping is exemplified by our study of genes implicated in CHD, a condition defined by complex morphological phenotypes poorly suited to traditional pooled dropout screens. Simple depletion analysis during cardiogenesis yielded limited hits (*GATA4*, *NKX2-5*), whereas transcriptomic signatures induced by gene LoF revealed additional genotype–phenotype associations for *SMAD2*, *CHD7*, and *KMT2D*, not only in fate decisions, but also in gene and module-level expression. Such "transcriptional perturbomes"—collections of single-cell transcriptomes under perturbation—enables flexible, reusable phenotype-agnostic screens. Indeed, they can be reanalyzed to interrogate emerging gene signatures from, for example, clinical datasets.

We also highlight the advantage of rapid generation of isogenic hPSC lines expressing individual perturbations. Molecular cloning and genome editing of a new line could be completed in ~1 month by an experienced operator, yielding ~80% of cells expressing a single shRNA (~42–72% of all sequenced cells; Appendix Table S3). Genome editing efficiency was not limiting as additional clones could have been readily produced; instead, as anticipated, the main bottleneck was obtaining sufficient scRNA-seq coverage per clone. Although we did not profile enough clones to rigorously compare isogenic lines with single versus multiple shRNA integrations, we observed reproducible findings across clones carrying the same shRNA (e.g., GATA4.1 and GATA4.3) and across distinct shRNAs targeting the same gene (e.g., SMAD2.2 and SMAD2.4). Notably, as few as ~50 cells per clone were sufficient to detect significant gene module associations, even within a multicellular system encompassing at least 10 cell types.

Undoubtedly, the strongest asset of iPS2-seq is its clonal awareness, enabling transcriptome comparisons within the same clone before and after LoF induction. This supports a more robust statistical framework than conventional perturbation versus control comparisons. Our day 2 cardiac differentiation data exemplify the limitations of standard approaches: variability in control clone differentiation speed could have been accounted for by including unperturbed counterparts. We addressed this in our cardioid experiments, where a control versus treatment design enabled reproducible, statistically significant findings despite moderate cell numbers and high model complexity.

The ability to follow clones was also critical for an unexpected biological finding: a stable, epigenetically-driven neuroectoderm bias in a subset of genome-edited iPSC clones. This priming involved persistent upregulation of neuroectodermal transcription factors, such as *ZIC1*, and was associated with impaired cardiomyocyte differentiation. Unexpectedly, it also skewed differentiation outcomes within neuroectodermal lineages, likely because biased clones failed to respond to timed external cues guiding unbiased cells. Among the transcriptional changes, *UNC5D* emerged as a top upregulated gene and surface marker, raising the possibility of selectively depleting biased clones by magnetic or fluorescence-activated cell sorting. While clonal variability in hPSC cultures has long been recognized, our study shows that it can be systematically measured, tracked, and mitigated using iPS2-seq. Critically, although previous studies have described transcriptional heterogeneity or dynamic states— including *ZIC1*-high populations (Nguyen et al, 2018)—to our knowledge, no screen has systematically tracked genome-edited hPSC clones before and after differentiation. iPS2-seq closes this methodological gap and opens the door to further biological discovery.

Our findings on *SMAD2* in cardiogenesis exemplify the flexibility of iPS2-seq. In the adult heart, *SMAD2* mediates pathological remodeling under pressure overload (Bjørnstad et al, 2012) and promotes fibrosis (Khalil et al, 2017), but its developmental role remains poorly understood. While this manuscript was under revision, the Seidman lab reported that *SMAD2* haploinsufficiency in iPSCs disrupts transcription factor binding and chromatin interactions critical for cardiovascular development (Ward et al, 2025). Our findings confirm and extend these results in an orthogonal system, narrowing the critical requirement for *SMAD2* to the patterning of cardiac progenitors and their specification into cardiomyocytes. Such temporal resolution would be difficult to achieve using irreversible CRISPR-Cas9 knockouts, highlighting the advantages of inducible, reversible post-transcriptional silencing. *SMAD2* silencing did not impair early

cardiac development, despite the well-established role of Activin/ Nodal/TGFβ signaling in hPSCs and for primitive streak induction (Bertero et al, 2015). This suggests that *SMAD3* may compensate specifically at this stage. *SMAD2* knockdown promoted a shift toward the epicardial lineage, potentially explaining why ALK5 inhibition empirically supports epicardium maintenance (Hofbauer et al, 2021). Of note, our data in cardiac organoids suggest a role for *SMAD2* in morphogenesis of the left ventricle, consistent with its mutation being associated with hypoplastic left heart syndrome and other severe CHD forms (Zaidi et al, 2013; Granadillo et al, 2018). Future studies will be required to establish the causal role of *SMAD2* more definitively, for example, through knockdown rescue experiments or orthogonal validation using inducible knockout systems.

While the features of iPS2-seq make it a powerful tool for functional genomics, we emphasize the value of a diverse toolbox tailored to specific experimental needs. It is therefore important to highlight key distinctions between iPS2-seq and other scRNA-seq screening approaches in hiPSC models (reviewed in Balmas et al (2023) and Li et al (2023b)). In contrast to our nimble shRNA-based silencing, most published methods rely on CRISPR-Cas9 systems requiring stable expression of Cas9 proteins or catalytically inactive variants fused to transcriptional regulators: strategies that are technically demanding, time-consuming, and prone to silencing (Karbassi et al, 2024). These systems also depend on lentiviral delivery of sgRNAs either in hiPSCs—raising concerns about silencing, insertional mutagenesis, and positional effect—or in hiPSC-derived cells, restricting implementation to cell types amenable to efficient transduction and selection. iPS2-seq instead relies on isogenic editing of an established genomic safe harbor, which not only mitigates screening variability but also enables a predictable path to hit validation, as demonstrated by our *SMAD2* silencing experiments. Inducibility is another major differentiator: most CRISPR-based approaches are constitutive, with the exception of the recently reported CHOOSE knockout system (Li et al, 2023a). While inducible CRISPRi has been used for dropout screens (Tian et al, 2019; Dräger et al, 2022), it has yet to be adapted for single-cell readouts. We also note that nearly all successful scRNA-seq screens in hPSCs to date focus on brain lineages, which are less prone to transgene silencing (Bertero et al, 2016). Last, and perhaps most importantly, iPS2-seq was designed with built-in clone awareness. Although this is conceptually achievable with CHOOSE, where the clonal barcode is constitutively expressed, direct matching of unperturbed versus perturbed clones is impractical due to the very large number of lentivirally derived clones, and of limited value given the risk of variegated expression from lentiviral silencing. Conversely, CRISPR-based, lentiviral methods remain more scalable for large libraries, particularly when a complete knockout is necessary to elicit a phenotype. In sum, iPS2-seq offers a robust and flexible platform to deploy focused scRNA-seq screens for haploinsufficient genes across diverse hPSC lines, particularly in lineages and organoid models less accessible to CRISPR-Cas9-based methods.

Despite their flexibility and reusability, pooled scRNA-seq–based screens are not optimal for all experimental contexts. They are primarily suited to detecting cell-autonomous effects, whereas non-cell-autonomous phenomena—such as those

mediated by secreted factors or extracellular matrix components —are often masked by unperturbed neighboring cells. To address this limitation, iPS2-seq can also be applied in an arrayed format, albeit at reduced throughput. Second, gene expression signatures do not fully capture cellular function, particularly in cases involving post-transcriptional regulation. They may also fall short in offering straightforward biological interpretation for complex phenotypes, such as those involving cell morphology or intercellular signaling. While this is a broader limitation of transcriptomics, it underscores the value of integrating scRNA-seq with orthogonal approaches such as proteomics, metabolomics, or functional assays. In addition to these general constraints shared with similar approaches, iPS2-seq presents specific features that should be considered when designing an experiment.

First, while shRNAs remain a powerful tool, even compared to more recent CRISPR-Cas9 approaches, their limitations are well known and include potential off-target effects, mitigated by rigorous controls and use of multiple shRNAs, incomplete knockdown, which may be actually advantageous when studying essential genes or modeling recessive mutations, and slower kinetics compared to alternative degron-based systems. Of note, validation of gene knockdown is inherently limited by the sensitivity of scRNA-seq, and protein-level validation by CITE-seq is not broadly scalable. While indirect validation via analysis of downstream targets is theoretically possible, such targets are typically unknown for the majority of genes being screened, precisely because their function is poorly characterized. As a result, false negatives should not be overinterpreted, as they may simply reflect poorly active shRNAs. Conversely, knockdown efficiency should not be used as the primary criterion for hit selection; rather, robust hits should be prioritized based on phenotypes that are reproducible across multiple shRNAs and independent clones.

Second, iPS2-seq is optimized for one perturbation per cell and does not readily support analysis of multi-gene interactions. While co-selection strategies (e.g., biallelic *AAVS1* targeting) could enrich for dual shRNA expression, engineering specific perturbation pairs remains challenging.

Third, iPS2-seq is best suited for screens targeting up to ~100 genes, primarily due to cost constraints. For higher-throughput applications, homology-directed repair-based genome editing, which achieves an efficiency of approximately 50 clones per million nucleofected cells, may become a bottleneck. Should the economy of scRNA-seq change in the future, iPS2-seq could be readily scaled further by implementing recombinase-mediated editing to streamline shRNA delivery, although it requires prior engineering of a landing pad cell line and plasmid adaptation. Of note, iPSC lines with recombination landing pads in the *AAVS1* and *CLYBL* genomic safe harbors have been reported and are publicly available (Blanch-Asensio et al, 2023). These lines may be used with an adapted version of the iPS2-seq plasmids for this purpose.

Furthermore, cost remains a practical barrier. We provide two protocols offering different trade-offs between cost and data richness. While several other scRNA-seq platforms exist, our optimization steps can likely be adapted to many of them. For example, the recently published sci-RNA-seq3 protocol (Martin et al, 2023) offers improved performance at low cost and could be readily implemented with minor modifications to iPS2-sci-seq.

# Methods

**Reagents and tools table**

| Reagent/resource | Reference or source | Identifier or catalog number |
|---|---|---|
| **Experimental models** | | |
| WTC-11 hiPSC line | Coriell | GM25256 |
| TTN-mEGFP hiPSC line | Coriell | AICS-0048-039 |
| **Recombinant DNA** | | |
| pAAV-Puro_siKD2.0 | This study - submitted to Addgene | #220536 |
| pAAV-Neo_CAG | This study - submitted to Addgene | #220537 |
| pZFN-AAVS1_ELD | Addgene | #159297 |
| pZFN-AAVS1-KKR | Addgene | #159298 |
| pAAV-Puro_siKD | Addgene | #86695 |
| MV-PGK-Puro-TK | Hera BioLabs | #SGK-005 |
| **Antibodies** | | |
| Alexa Fluor 647 Mouse monoclonal Anti-Cardiac Troponin T | BD Pharmigen | #565744 |
| Alexa Fluor 647 Mouse monoclonal IgG1, κ Isotype Control | BD Pharmigen | #566011 |
| Anti-Wilms Tumor Protein antibody -Rabbit monoclonal | Abcam | #ab89901 |
| Anti-Sarcomeric Alpha Actinin antibody -Mouse monoclonal | Abcam | #ab9465 |
| Anti-COL1A1 antibody -Rabbit polyclonal | Invitrogen | #PA5-29569 |
| Anti-FOXG1 antibody -Rabbit polyclonal | Abcam | #ab18259 |
| Anti-PAX6 antibody -Mouse monoclonal | GeneTex | #GTX634863 |
| Goat anti-Rabbit IgG (H + L) Alexa Fluor 633 | Invitrogen | #A-21070 |
| Goat anti-Mouse IgG (H + L) Alexa Fluor 488 | Invitrogen | #A-11001 |
| Goat anti-Mouse IgG (H + L) Alexa Fluor 568 | Invitrogen | #A-11004 |
| Goat anti-Rabbit IgG (H + L) Alexa Fluor 488 | Invitrogen | #A-11008 |
| Anti-Rabbit IgG SMAD2 antibody | Cell Signaling Technology | #5339 |
| Anti-mouse IgG1 Vinculin antibody | Sigma-Aldrich | #SAB4200080 |
| Goat Anti-Mouse IgG (H + L)-HRP Conjugate | Bio-Rad | #1706516 |
| Goat Anti-Rabbit IgG (H + L)-HRP Conjugate | Bio-Rad | #1706515 |
| TotalSeq-B0057 anti-human β2-microglobulin antibody | BioLegend | #316325 |
| TotalSeq-B0251 anti-human Hashtag 1 antibody | BioLegend | #394631 |
| TotalSeq-B0252 anti-human Hashtag 2 antibody | BioLegend | #394633 |
| **Oligonucleotides and other sequence-based reagents** | | |
| ssDNA oligo pool | IDT | oPools |
| iPS2-seq_dsDNA_R (5′-GCTGATCAGCGAGCTAC-3′) | Eurofins Genomics | DNA Oligo desalted |
| iPS2-seq_step1QC_F (5′-CGAACGCTGACGTCATCAACC-3′) | Eurofins Genomics | DNA Oligo desalted |
| iPS2-seq_QC_R (5′-CCAGCATGCCTGCTATTCTC-3′) | Eurofins Genomics | DNA Oligo desalted |
| iPS2-seq_step2QC_F (5′-CCTGGAACTGATCATCTGCG-3′) | Eurofins Genomics | DNA Oligo desalted |
| iPS2-seq_dsDNA_F (5′-AGTTCCCTATCAGTGATAGAGATCCC-3′) | Eurofins Genomics | DNA Oligo desalted |
| iPS2-seq_step1NGS_F (5′-GTGACTGGAGTTCAGACGTGTGCTCTTCCGATCTAGTTCCCTATCAGTGATAGAGATCCC-3′) | Eurofins Genomics | DNA Oligo desalted |
| iPS2-seq_NGS_R (5′-CTACACGACGCTCTTCCGATCT[12 N]GCTGATCAGCGAGCTAC-3′) | Eurofins Genomics | DNA Oligo desalted |

| Reagent/resource | Reference or source | Identifier or catalog number |
|---|---|---|
| iPS2-seq_step2NGS_F (5′-GTGACTGGAGTTCAGACGTGTGCTCTTCCGATCTAGAAGCAGCTGAAGTGCGAGAG-3′) | Eurofins Genomics | DNA Oligo desalted |
| Locus_fw (5′-CTGTTTCCCCTTCCCAGGCAGGTCC-3′) | Eurofins Genomics | DNA Oligo desalted |
| Locus_rev (5′-TGCAGGGGAACGGGGCTCAGTCTGA-3′) | Eurofins Genomics | DNA Oligo desalted |
| Puro_rev (5′-TCGTCGCGGGTGGCGAGGCGCACCG-3′) | Eurofins Genomics | DNA Oligo desalted |
| OPTtetR_fw (5′-CCACCGAGAAGCAGTACGAG-3′) | Eurofins Genomics | DNA Oligo desalted |
| Neo_rev (5′-GTGCCCAGTCATAGCCGAAT-3′) | Eurofins Genomics | DNA Oligo desalted |
| Backbone_rev (5′-ATGCACCACCGGGTAAAGTT-3′) | Eurofins Genomics | DNA Oligo desalted |
| iPS2-sci-seq_pT (5′-ACGACGCTCTTCCGATCT[8 N][RT_index][30 T]VN-3′) | Eurofins Genomics | DNA Oligo NGS grade desalted |
| iPS2-sci-seq_tetR (5′-ACGACGCTCTTCCGATCT[8 N][RT_index]TCGAGGCTGATCAGCGAGCTAC3′) | Eurofins Genomics | DNA Oligo NGS grade desalted |
| iPS2-sci-seq_P5 (5′-AATGATACGGCGACCACCGAGATCTACAC[i5]ACACTCTTTCCCTACACGACGCTCTTCCGATCT-3′) | Eurofins Genomics | DNA Oligo NGS grade desalted |
| iPS2-sci-seq_P7 (5′-CAAGCAGAAGACGGCATACGAGAT[i7]GTCTCGTGGGCTCGG-3′) | Eurofins Genomics | DNA Oligo NGS grade desalted |
| iPS2-sci-seq_P7A (5′-CAAGCAGAAGACGGCATACGAGAT[i7]GTCTCGTGGGCTCGGAGATGTGTATAAGAGACAGCGGCTCCCCCAGATGAACG -3′) | Eurofins Genomics | DNA Oligo NGS grade desalted |
| iPS2-10X-seq_truseq (5′-CTACACGACGCTCTTCCGATCT-3′) | Eurofins Genomics | DNA Oligo NGS grade desalted |
| iPS2-10X-seq_inner (5′-GTGACTGGAGTTCAGACGTGTGCTCTTCCGATCTCGGCTCCCCCAGATGAA-3′) | Eurofins Genomics | DNA Oligo NGS grade desalted |
| iPS2-sci-seq_P7B (5′-CAAGCAGAAGACGGCATACGAGAT[i7]GTCTCGTGGGCTCGGAGATGTGTATAAGAGACAGAAGCAGCTGAAGTGCGAGAG-3′) | Eurofins Genomics | DNA Oligo NGS grade desalted |
| iPS2-sci_P7B_short (5′-AGAAGCAGCTGAAGTGCGAGAG-3′) | Eurofins Genomics | DNA Oligo NGS grade desalted |
| iPS2-10X-seq_outer (5′-CTGTTCGGCCTGGAACTGAT-3′) | Eurofins Genomics | DNA Oligo NGS grade desalted |
| HPRT1_fw (5′-TGACACTGGCAAAACAATGCA-3′) | Eurofins Genomics | DNA Oligo desalted |
| HPRT1_rev (5′-GGTCCTTTTCACCAGCAAGCT-3′) | Eurofins Genomics | DNA Oligo desalted |
| ZIC1_fw (5′-CGCAAACACATGAAGGTCCAC-3′) | Eurofins Genomics | DNA Oligo desalted |
| ZIC1_rev (5′-AGGGCGATAAGGAGCTTGTG-3′) | Eurofins Genomics | DNA Oligo desalted |
| SIX3_fw (5′-TCACTCCCACACAAGTAGGC-3′) | Eurofins Genomics | DNA Oligo desalted |
| SIX3_rev (5′-CTGGTGCTGGAGCCTGTTC-3′) | Eurofins Genomics | DNA Oligo desalted |
| NEUROG3_fw (5′-CGCCGGTAGAAAGGATGACG-3′) | Eurofins Genomics | DNA Oligo desalted |
| NEUROG3_rev (5′-GTCACTTCGTCTTCCGAGGC-3′) | Eurofins Genomics | DNA Oligo desalted |
| POU5F1_fw (5′-AGTGAGAGGCAACCTGGAGA-3′) | Eurofins Genomics | DNA Oligo desalted |
| POU5F1_rev (5′-ACACTCGGACCACATCCTTC-3′) | Eurofins Genomics | DNA Oligo desalted |
| NANOG_fw (5′-CATGAGTGTGGATCCAGCTTG-3′) | Eurofins Genomics | DNA Oligo desalted |
| NANOG_rev (5′-CCTGAATAAGCAGATCCATGG-3′) | Eurofins Genomics | DNA Oligo desalted |
| SOX2_fw (5′-TGGACAGTTACGCGCACAT-3′) | Eurofins Genomics | DNA Oligo desalted |
| SOX2_rev (5′-CGAGTAGGACATGCTGTAGGT-3′) | Eurofins Genomics | DNA Oligo desalted |
| SMAD2_fw (5′-AAAGTATGGACACAGGCTCTCC-3′) | Eurofins Genomics | DNA Oligo desalted |
| SMAD2_rev (5′-TGCTATCGAACACCAAAATGCAG-3′) | Eurofins Genomics | DNA Oligo desalted |
| B2M_fw (5′-AGATGAGTATGCCTGCCGTG-3′) | Eurofins Genomics | DNA Oligo desalted |
| B2M_rev (5′-TGCGGCATCTTCAAACCTCC-3′) | Eurofins Genomics | DNA Oligo desalted |
| TBXT_fw (5′-TGCTTCCCTGAGACCCAGTT-3′) | Eurofins Genomics | DNA Oligo desalted |
| TBXT_rev (5′-GATCACTTCTTTCCTTTGCATCAAG-3′) | Eurofins Genomics | DNA Oligo desalted |

| Reagent/resource | Reference or source | Identifier or catalog number |
|---|---|---|
| MESP1_fw (5′-TCGAAGTGGTTCCTTGGCAGAC-3′) | Eurofins Genomics | DNA Oligo desalted |
| MESP1_rev (5′-CCTCCTGCTTGCCTACAAAGTGTC-3′) | Eurofins Genomics | DNA Oligo desalted |
| EOMES_fw (5′-CTATCAGTACAGCCAGGGGG-3′) | Eurofins Genomics | DNA Oligo desalted |
| EOMES_rev (5′-AAGGAAACATGCGCCTGCC-3′) | Eurofins Genomics | DNA Oligo desalted |
| NKX2.5_fw (5′-GAGCCGAAAAGAAAGCCTGAA-3′) | Eurofins Genomics | DNA Oligo desalted |
| NKX2.5_rev (5′-CACCGACACGTCTCACTCAG-3′) | Eurofins Genomics | DNA Oligo desalted |
| GATA4_fw (5′-AATCTAAGACACCAGCAGCTCCTTC-3′) | Eurofins Genomics | DNA Oligo desalted |
| GATA4_rev (5′-CATGGCCAGACATCGCACT-3′) | Eurofins Genomics | DNA Oligo desalted |
| PDGFRA_fw (5′-CCGGCGTTCCTGGTCTTAG-3′) | Eurofins Genomics | DNA Oligo desalted |
| PDGFRA_rev (5′-GCTCACTTCACTCTCCCCAAAG-3′) | Eurofins Genomics | DNA Oligo desalted |
| MYH7_fw (5′-AGACTGTCGTGGGCTTGTATCAG-3′) | Eurofins Genomics | DNA Oligo desalted |
| MYH7_rev (5′-GCCTTTGCCCTTCTCAATAGG-3′) | Eurofins Genomics | DNA Oligo desalted |
| WT1_fw (5′-CCAGGACTCATACAGGTGAAAAG-3′) | Eurofins Genomics | DNA Oligo desalted |
| WT1_rev (5′-CTGATGCATGTTGTGATGGCG-3′) | Eurofins Genomics | DNA Oligo desalted |
| TBX18_fw (5′-GCTAAAGGCTTCCGAGACTCC-3′) | Eurofins Genomics | DNA Oligo desalted |
| TBX18_rev (5′-GAACTTGCATTGCCTTGCTTG-3′) | Eurofins Genomics | DNA Oligo desalted |
| TTN_fw (5′-GTAAAAAGAGCTGCCCCAGTGA-3′) | Eurofins Genomics | DNA Oligo desalted |
| TTN_rev (5′-GCTAGGTGGCCCAGTGCTACT-3′) | Eurofins Genomics | DNA Oligo desalted |
| COL1A1_fw (5′-GGAGGAATTTCCGTGCCTGG-3′) | Eurofins Genomics | DNA Oligo desalted |
| COL1A1_rev (5′-CAATCCTCGAGCACCCTGA-3′) | Eurofins Genomics | DNA Oligo desalted |
| COL5A2_fw (5′-CCCACAGCTGACTTCATGGT-3′) | Eurofins Genomics | DNA Oligo desalted |
| COL5A2_rev (5′-CACCATATCCTTCATCCTCGTC-3′) | Eurofins Genomics | DNA Oligo desalted |
| CDH5_fw (5′-TTGGAACCAGATGCACATTGAT-3′) | Eurofins Genomics | DNA Oligo desalted |
| CDH5_rev (5′-TCTTGCGACTCACGCTTGAC-3′) | Eurofins Genomics | DNA Oligo desalted |
| TrueGuide sgRNA custom AAVS1 (5′-GUCACCAAUCCUGUCCCUAG-3′) | Invitrogen | #A35534 |
| **Chemicals, enzymes, and other reagents** | | |
| Nuclease-free water | Invitrogen | #AM9937 |
| Elution buffer | QIAGEN | #19086 |
| Deoxynucleotide (dNTP) Solution Mix (10 mM) | New England Biolabs | #N0447 |
| MgSO$_4$ (100 mM) | New England Biolabs | #B1003S |
| Isothermal Amplification Buffer II (10X) | New England Biolabs | #B0374S |
| Bst 3.0 DNA Polymerase (8 U/μL) | New England Biolabs | #M0374S |
| Exonuclease I (20 U/μL) | New England Biolabs | #M0293S |
| FastDigest MluI | Thermo Scientific | #FD0564 |
| FastDigest BglII | Thermo Scientific | #FD0083 |
| FastAP Thermosensitive Alkaline Phosphatase (1 U/μL) | Thermo Scientific | #EF0654 |
| FastDigest Green Buffer (10X) | Thermo Scientific | #B72 |
| SYBR Safe DNA Gel Stain (10,000X) | Invitrogen | #S33102 |
| NEBuilder HiFi DNA Assembly Master Mix (2X) | New England Biolabs | #M5520A |
| GoTaq DNA Polymerase (5 U/μL) | Promega | #M300A1 |
| Colorless GoTaq Flexi Buffer (5X) | Promega | #M890A |
| MgCl2 (25 mM) | Promega | #A351H |
| Shrimp Alkaline Phosphatase (1 U/μL) | New England Biolabs | #M0371 |
| FastDigest SalI | Thermo Scientific | #FD0644 |

| Reagent/resource | Reference or source | Identifier or catalog number |
|---|---|---|
| FastDigest SgsI (AscI) | Thermo Scientific | #FD1894 |
| Rapid DNA Ligation kit | Invitrogen | #K1423 |
| SPRIselect beads | Beckman Coulter | #B23318 |
| ThermoPol Reaction Buffer (10X) | New England Biolabs | #B9004S |
| Deep Vent DNA Polymerase (2 U/µL) | New England Biolabs | #M0258S |
| Q5 Hot Start High-Fidelity DNA Polymerase (2 U/µL) | New England Biolabs | #M0493 |
| Q5 Reaction Buffer (5X) | New England Biolabs | #B9027 |
| NEBNext high-fidelity 2× PCR master mix | New England Biolabs | #M0541L |
| Essential 8 Medium | Gibco | #A1517001 |
| Geltrex LDEV-Free hESC-qualified | Gibco | #A1413302 |
| Thiazovivin | Cayman chemicals | #14245-25 |
| StemPro Accutase Cell Dissociation Reagent | Gibco | #A1110501 |
| TrueCut Cas9 Protein v2 | Invitrogen | #A36498 |
| Polyamine Supplement (1000X) | Sigma-Aldrich | #P8483-5ML |
| Chroman 1 (10 µM) | MedChemExpress | #HY-15392 |
| Emricasan (5 mM) | Sigma-Aldrich | #SML2227-5ML |
| trans-ISRIB (1.4 mM) | Sigma-Aldrich | #SML0843-5MG |
| EDTA (0.5 M) | MerckMillipore | #4055-OP |
| DPBS (no calcium, no magnesium) | Euroclone | #ECM4004 |
| Puromycin Dihydrochloride | Gibco | #A1113803 |
| Geneticin Selective Antibiotic (G418 Sulfate) (50 mg/mL) | Gibco | #10131035 |
| Fialuridine | Sigma-Aldrich | #SML0632 |
| LongAmp Taq DNA Polymerase (2.5 U/µL) | New England Biolabs | #M0323S |
| LongAmp Taq Reaction Buffer (5X) | New England Biolabs | #B0323S |
| Paraformaldehyde 4% (PFA) | Electron Microscopy Sciences | #15735-95 |
| YOYO-1 Iodide | Invitrogen | #Y3601 |
| NaCl (5 M) | Invitrogen | #AM9759 |
| MgCl2 (1 M) | Invitrogen | #AM9530G |
| Tris-HCl pH 7.5 (1 M) | Thermo Scientific Chemicals | #15493699 |
| IGEPAL CA-630 | Sigma-Aldrich | #I8896-50ML |
| Bovine serum albumin (BSA, 20 mg/mL) | New England Biolabs | #B9000S |
| SUPERase In RNase Inhibitor (20 U/µL) | Invitrogen | #AM2694 |
| Superscript IV Reverse Transcriptase (with 0.1 mM DTT and 5× SSIV Buffer) | Invitrogen | #18090200 |
| RNaseOUT Recombinant Ribonuclease Inhibitor | Invitrogen | #10777019 |
| Spermidine | MP bio | #0215206801 |
| DAPI | Invitrogen | #D3571 |
| NEBNext Ultra II Non-Directional RNA Second Strand Synthesis Module | New England Biolabs | #E6111L |
| Nextera DNA Sample Preparation kit (with Tagmentase enzyme and buffer) | Illumina | #FC-121-1030 |
| DNA binding buffer | Zymo | #D4004 |
| Glycerol (50%) | VWR | #BIOVB1012-100 |
| Tween 20 (10%) | Bio-Rad | #1662404 |
| Low TE Buffer | Invitrogen | #12090-015 |
| Bovine serum albumin (BSA for FACS, IF, WB, and CellPlex) | Sigma-Aldrich | #90-48-46-8 |

| Reagent/resource | Reference or source | Identifier or catalog number |
|---|---|---|
| Fixable Viability Dye eFluor 780 | Invitrogen | #65-0865-14 |
| SdaI (SbfI) (10 U/μL) | Thermo Scientific | #ER1192 |
| Bsp119I (BstBI) (10 U/μL) | Thermo Scientific | #ER0121 |
| mTeSR Plus | STEMCELL Technologies | #100-0276 |
| Corning Matrigel hESC-Qualified Matrix, LDEV-free | Corning | #354277 |
| Opti-MEM | Gibco | #31985047 |
| GeneJuice | Sigma-Aldrich | #70967 |
| Saponin | Sigma-Aldrich | #8047-15-2 |
| Fixable Viability Dye eFluor 450 | Invitrogen | #65-0863-14 |
| Y-27632 (hydrochloride) | Cayman chemicals | #10005583 |
| RPMI-1640 with glutamine | Gibco | #11835030 |
| Bovine serum albumin (BSA for cardiac differentiations) | STEMCELL Technologies | #100-0177 |
| Ascorbic acid | Waco-chemicals | #321-44823 |
| CHIR99021 | Cayman chemicals | #13122 |
| Wnt-C59 | Cayman chemicals | #16644 |
| B27 supplement | Gibco | #17504044 |
| Trypsin | Gibco | #25200056 |
| Fetal bovine serum (FBS) | Gibco | #A5670701 |
| TrypLE Select Enzyme | Gibco | #12563029 |
| Iscove's Modified Dulbecco's Medium (IMDM) | Gibco | #21980032 |
| Ham's F-12 Mutrient Mix with Glutamax | Gibco | #31765068 |
| Chemically Defined Lipid Concentrate | Gibco | #11905031 |
| 1-Thioglycerol, BioReagent | Sigma-Aldrich | #M6145 |
| Optiferrin | InVitria | #777TRF029 |
| FGF2-G3 | Qkine | #Qk025 |
| Activin A | Qkine | #Qk001 |
| BMP4 | Qkine | #Qk038 |
| LY294002 | Selleckchem | #S1105 |
| Insulin | Biocom (Gibco) | #A11382II |
| Retinoic acid | Sigma-Aldrich | #R2625 |
| D(+)-Sucrose | Carlo Erba | #477187 |
| Tissue-Tek O.C.T. compound | Sakura | #62550 |
| Goat Serum | Bio-Rad | #C07SA |
| Triton-X-100 | Sigma-Aldrich | #T9284 |
| Pap Pen | ThermoFisher Scientific | #R3777 |
| Hoechst 33342 | Invitrogen | #H1399 |
| ProLong Glass Antifade Mountant | Invitrogen | #P36982 |
| Glasgow's MEM (GMEM) | Gibco | #11710035 |
| KnockOut Serum Replacement | Gibco | #10828028 |
| MEM Non-Essential Amino Acids Solution (100X) | Gibco | #11140050 |
| Sodium Pyruvate (100 mM) | Gibco | #11360070 |
| 2-Mercaptoethanol (50 mM) | Gibco | #31350010 |
| IWR-1 | Sigma-Aldrich | #I0161 |
| SB-431542 | Tocris | #1614 |

| Reagent/resource | Reference or source | Identifier or catalog number |
|---|---|---|
| DMEM/F-12 | Gibco | #11320082 |
| GlutaMAX Supplement | Gibco | #35050061 |
| Penicillin–Streptomycin (10,000 U/mL) | Gibco | #15140122 |
| N-2 Supplement (100X) | Gibco | #17502001 |
| Tetracycline hydrochloride | Sigma-Aldrich | #T7660 |
| DPBS (with calcium, with magnesium) | Euroclone | #ECB4053 |
| High-Capacity cDNA Reverse Transcription Kit | Applied Biosystems | #4368813 |
| PowerUp SYBR Green Master Mix | Applied Biosystems | #A25777 |
| Protease Inhibitor Cocktail | Roche | #11873580001 |
| NuPAGE LDS Sample Buffer (4×) | Invitrogen | #NP0008 |
| PageRuler Plus Prestained Protein Ladder | Thermo Scientific | #26619 |
| 4–20% Mini-PROTEAN TGX Precast Protein Gels | Bio-Rad | #4561096 |
| Amersham Protran 0.45 NC nitrocellulose Western blotting membrane | Cytiva | #GE10600002 |
| Skin Milk Powder | Milipore | #70166 |
| Clarity Western ECL Substrate | Bio-Rad | #1705061 |
| Trypan Blue | Sigma-Aldrich | #T8154 |
| MACS BSA Stock Solution | Miltenyi Biotec | #130-091-376 |
| Digitonin (5%) | Invitrogen | #BN2006 |
| Nuclei Buffer (20×) | 10X Genomics | #2000207 |
| RNase Inhibitor (40×) | 10X Genomics | #2001488 |
| DNase I (1 U/µL) | Thermo Scientific | #EN0521 |
| DTT | Sigma-Aldrich | #646563 |
| **Software** | | |
| catcheR | https://github.com/alessandro-bertero/catcheR This study-GitHub | |
| Docker | https://docs.docker.com/manuals/ | |
| CREDOengine | https://github.com/alessandriLuca/CREDOengine | |
| Cellranger version 7.1.0 | https://www.10xgenomics.com/support/software/cell-ranger/downloads/previous-versions | |
| cellranger7hedge | https://hub.docker.com/repository/docker/hedgelab/cellranger7hedge This study-Docker image | |
| bbi-sci | https://github.com/bbi-lab/bbi-sci | |
| Seurat v5 | https://github.com/satijalab/seurat | |
| Monocle 3 | https://github.com/cole-trapnell-lab/monocle3 | |
| R version 4.3.1 | cran.r-project.org | |
| custom R scripts for data analysis after catcheR | https://marialuisaratto.github.io/catcheRdocs/ This study-GitHub | |

| Reagent/resource | Reference or source | Identifier or catalog number |
|---|---|---|
| scMAGeCK | https://bitbucket.org/weililab/scmageck/src/master/demo/ | |
| ggplot2 | https://ggplot2.tidyverse.org | |
| pheatmap | https://github.com/raivokolde/pheatmap | |
| viridislite | R/scale-viridis.R | |
| Incucyte Spheroid Analysis Software Module | Sartorius | #9600-0019 |
| GenomeStudio v.2 | Illumina | – |
| cnvPartition CNV Analysis Plugin version 3.2.1 | Illumina | – |
| FlowJo version 10 | BD Biosciences | – |
| Prism 10 | GraphPad | – |
| **Other** | | |
| NEB 5-alpha Competent *E. coli* (High Efficiency) | New England Biolabs | #C2987H |
| QIAquick PCR Purification Kit | QIAGEN | #28104 |
| QIAEX II Gel Extraction Kit | QIAGEN | #20021 |
| QIAfilter Plasmid Midi Kit | QIAGEN | #12243 |
| EndoFree Plasmid Buffer Set | QIAGEN | #19048 |
| P3 Primary Cell 4D-Nucleofector X Kit L | Lonza | #V4XP-3024 |
| P3 Primary Cell 4D-Nucleofector X Kit S | Lonza | #V4XP-3032 |
| Monarch Spin gDNA Extraction Kit | New England Biolabs | #T3010 |
| Qubit dsDNA Broad Range kit | Invitrogen | #Q32850 |
| Qubit dsDNA HS kit | Invitrogen | #Q32851 |
| Qubit tubes | Invitrogen | #Q32856 |
| High Sensitivity D1000 Screen Tape and reagents | Agilent | #5067-5584 |
| High Sensitivity D5000 Screen Tape and reagents | Agilent | #5067-5592 |
| Zymo DNA Clean & Concentrator-25 kit | Zymo | #D4033 |
| Chromium Next GEM Single Cell 3′ Kit v3.1 dual index, 4 rxns | 10X Genomics | #1000269 |
| Chromium Next GEM Chip G Single Cell Kit 16 rxns | 10X Genomics | #1000127 |
| 3′ Feature Barcode Kit. 16 rxns | 10X Genomics | #1000262 |
| 3′ CellPlex Kit Set A | 10X Genomics | #1000261 |
| Dual Index Kit TT Set A, 96 rxns | 10X Genomics | #PN-1000215 |
| Dual Index Kit NN Set A, 96 rxns | 10X Genomics | #PN-1000243 |
| Dual Index Kit NT Set A, 96 rxns | 10X Genomics | #PN-1000242 |
| NextSeq 1000/2000 P2 reagents v3 (100 cycles) | Illumina | #20046811 |
| NextSeq 2000 P3 reagents (100 cycles) | Illumina | #20040559 |
| NextSeq 500 High output (75 cycles) | Illumina | #20024906 |
| MiSeq Reagent Kit v3 (150-cycle) | Illumina | #MS-102-3001 |
| Quick-RNA MiniPrep Kit | Zymo | #R1055 |
| Pierce BCA Protein Assay Kits | Thermo Scientific | #23227 |
| Papain Dissociation Kit | Worthington Biochemical Corporation | #LK003150 |
| Plasmid verification service | Plasmidsaurus | https://www.plasmidsaurus.com |
| Sanger sequencing service | Eurofins Genomics | https://www.eurofinsgenomics.eu |

| Reagent/resource | Reference or source | Identifier or catalog number |
|---|---|---|
| NanoDrop Spectrophotometer | Thermo Scientific | #ND-ONE-W |
| 4D-Nucleofector Core Unit | Lonza | #AAF-1003B |
| 4D-Nucleofector X Unit | Lonza | #AAF-1003X |
| Qubit 2.0 Fluorometer | Invitrogen | #Q32866 |
| Tapestation 2200 | Agilent | #G2964AA |
| PCR machine (96-well) Eppendorf Mastercycler X50s | Eppendorf | #6311000010 |
| Centrifuge 5430 R | Eppendorf | #5430-R |
| Magnetic Separator | 10X Genomics | #230003 |
| Bright-Line Hemacytometer | Merck | #Z359629 |
| SH800S Sorter | Sony Biotech | Blue (488 nm)/Red (640 nm) |
| 1.5 ml DNA LoBind tube | Eppendorf | #30108051 |
| 96-well DNA LoBind plate | Eppendorf | # 30129679 |
| Microseal 'B' seal Seals | Bio-Rad | #MSB1001EDU |
| Cell strainer 100 μm | Falcon Corning | #431752 |
| Mini cell strainer 40 μm | Pluriselect | #43-10040-51 |
| Flowmi cell strainer, 40 μm | Bel-Art | #H13680-0040 |
| Chromium X | 10X Genomics | #1000331 |
| Nunclon sphera-treated, U-shaped-bottom 96-well microplate | Thermo Scientific | #174925 |
| PrimeSurface, V-shaped-bottom 96-well microplate | Sumitomo Bakelite | #MS-9096VZ |
| Coverslips | VWR | #631-0137 |
| Glass slides | Globe Scientific | #1354 W |
| Incucyte SX5 Live-Cell Analysis System | Sartorius | – |
| FACSCelesta cell analyzer | BD Biosciences | Blue (488 nm)/Red (640 nm)/Violet (405 nm) |
| FACSVerse cell analyser | BD Biosciences | Blue (488 nm)/Red (640 nm)/Violet (405 nm) |
| QuantStudio 6 Flex Real-Time PCR system | Applied Biosystems | #4485694 |
| GloMax Discover Microplate Reader | Promega | #GM3000 |
| ChemiDoc MP imager system | Bio-Rad | #12003154 |
| Confocal microscope | Leica Biosystems | #SP8 |
| Cryostat | Leica Biosystems | #CM1950 |
| Molecular karyotyping service | Life & Brain | https://www.lifeandbrain.com/ |

## hiPSC culture

Apparently healthy male hiPSCs (RRID: CVCL_Y803, commonly known as WTC-11) were a kind gift of Bruce Conkin (J. David Gladstone Institutes). The WTC-11 TTN-mEGFP reporter (RRID: CVCL_UD16) was developed at the Allen Institute for Cell Science (allencell.org/cell-catalog) and is available through Coriell. Cell line authentication by short tandem repeat (STR) profiling was not performed after receipt. hiPSCs were cultured feeder-free on either hESC-qualified Matrigel coating (5 μg cm$^{-2}$) in mTeSR Plus (iPS2-seq experiments) or Geltrex coating (5.2 μg cm$^{-2}$) in Essential 8 media (*SMAD2* knockdown studies). For routine passaging, colonies were dissociated as small clumps with 0.5 mM EDTA in DPBS, and replated for 16 h in media supplemented with 2 μM Thiazovivin. hiPSCs were maintained at 37 °C in a humidified atmosphere with 5% $CO_2$ and routinely screened and tested negative for mycoplasma.

## Plasmids

pAAV-Puro_siKD2.0 is a modification of pAAV-Puro_sikD (Bertero et al, 2018b). The SbfI-BstBI restriction fragment (containing the 5′ *AAVS1* homology arm and the puromycin resistance gene trap with its bovine growth hormone polyadenylation signal) was excised and replaced with the following fragments (listed in 5′–3′ direction): (1) a truncated thymidine kinase cDNA followed by SV40 polyadenylation signal, encoded on the negative strand (Yusa et al, 2013); (2) a

PGK-EM7 promoter, encoded on the negative strand; (3) the 5' *AAVS1* homology arm and the puromycin resistance gene trap without polyadenylation signal; and (4) the SV40 polyadenylation signal. Fragments 1, 2, and 4 were PCR amplified from MV-PGK-Puro-TK; fragment 3 was PCR-amplified from pAAV-Puro_sikD; all fragments contained short overlaps that allowed directional Gibson assembly. pAAV-Neo_CAG was obtained through a combination of gene synthesis and blunt ligation of a restriction fragment encoding the CAG promoter. *AAVS1* locus ZFN plasmids (pZFN-AAVS1_ELD and pZFN-AAVS1_KKR) were previously described (Bertero et al, 2016). All plasmids were sequence verified in their entirety by Plasmidsaurus.

Plasmid pools with barcoded inducible shRNAs were designed, generated, and quality controlled according to methods detailed in Appendix Protocol 1. Specifically, for each gene we chose three to four shRNAs from the TRC library based on available validation data (Appendix Table S1). We included the top predicted shRNAs against *GATA4* and *NKX2-5* twice to ensure their representation despite an anticipated strong negative selective pressure due to the established role of these two genes in cardiomyocyte development (Anderson et al, 2018; Ang et al, 2016). Lastly, we included three negative controls (scrambled—SCR, *EGFP*, and *B2M*), for a total of 23 shRNAs. A pool of 23 ssDNA oligonucleotides, encoding shRNAs paired to unique BCs and random UCIs (Appendix Table S1), was converted to dsDNA by primer extension with Bst3.0 DNA polymerase, gel extracted, and assembled with NEBuilder HiFi DNA Assembly Master Mix into BglII-MluI cut and dephosphorylated pAAV-Puro_siKD2.0. The resulting plasmid pool was cut with AscI-SalI and ligated to the SalI-MluI restriction fragment with CAG-OPTtetR from pAAV-Puro_siKD2.0. After each cloning step, a subset of bacterial clones was sacrificed for colony PCR to assess cloning efficiency qualitatively (by DNA electrophoresis) and quantitatively (by Sanger sequencing). Plasmid pools were further quality controlled through NGS with Illumina NextSeq 1000 and P2 Reagents (100 cycles) v3. These libraries were sequenced as a single read 1 of 118 bp. The analysis was done with *catcheR_step1QC* and *catcheR_step2QC* as described in Appendix Protocol 4. Finally, endotoxin-free plasmid pools were prepared for transfection.

Two ssDNA oligos with shRNAs for *B2M* or *SMAD2* from the original pool were ordered separately, with a minor change to the barcode of SMAD2.2 shRNA to "ACTCGAGA". These were cloned in parallel into the pAAV-Puro_siKD2.0 plasmid following Appendix Protocol 1. Once the final cloning step was completed, bacterial colonies were individually screened by PCR for correct integration, and then Sanger sequenced with iPS2-seq_step1QC_F and OPTtetR_fw primers to QC the shRNA and UCI regions, respectively. For each shRNA, five clones containing unique UCIs and mutation-free shRNAs were selected and pooled equally to create a 5-UCI pool for a single target.

## Genome editing

WTC-11 hiPSCs were engineered with inducible barcoded shRNA pools according to the method detailed in Appendix Protocol 1, except that plasmids were delivered by transfection instead of nucleofection (which we recommend for larger screens due to the higher efficiency). Subconfluent iPSC colonies were dissociated to very small clumps with 0.5 mM EDTA in DPBS and seeded at a density of $3 \times 10^4$ cells cm$^{-2}$ in six-well culture dishes using media supplemented with 2 µM Thiazovivin. Each well was immediately exposed to a transfection mix based of 100 µL of Opti-MEM and containing 6 µL of GeneJuice and 2 µg of plasmids (divided equally between the two targeting vectors [pool of pAAV-Puro_siKD2.0 with the 23 barcoded shRNAs and pAAV-Neo_CAG], and the ZFN plasmids [pZFN-AAVS1_ELD and pZFN-AAVS1_ELD]). Media was refreshed after 16 h, and biallelic *AAVS1* targeted clones were selected using 0.5 µg mL$^{-1}$ puromycin and 25 µg mL$^{-1}$ Geneticin from day 2 through 5 post transfection; 2 µM Thiazovivin was added for the first 48 h of selection. Clones were passaged at day 12 post-transfection, and immediately counter-selected with 200 nM fialuridine (FIAU) for 4 days, to eliminate clones carrying random integrations of the inducible shRNA plasmid. Genome editing was performed twice to generate independent replicates of hiPSC clonal pools. In the first experiment, puromycin and Geneticin-selected clones were picked in order to correlate FIAU resistance with genotype; for this, a small portion of the colony was seeded in a separate dish, expanded, and analyzed by PCR according to the optional procedure detailed in Appendix Protocol 1. FIAU-resistant clones were subsequently pooled. In the second experiment, puromycin and Geneticin-selected clones were passaged and FIAU selected as a pool, our recommended approach.

For the *SMAD2* knockdown experiments, TTN-mEGFP WTC-11 hiPSCs were gene edited following Appendix Protocol 1 with the following changes. Two parallel nucleofections were performed to engineer inducible shRNAs targeting either *SMAD2* or *B2M*. To allow for homozygous shRNA integration, the pAAV-Neo_CAG plasmid was excluded, delivering 5 µg of the shRNA targeting vector (5 UCI pool of pAAV-Puro_siKD2.0 of *SMAD2* or *B2M* shRNAs) along with 2.5 µg of each ZFN plasmid. After selection with 0.5 µg mL$^{-1}$ puromycin, a subset of colonies were picked, expanded, and analyzed by PCR to identify monoclonal hiPSC lines with homozygous or heterozygous *AAVS1* integration. The remaining colonies were maintained as polyclonal pools.

## Molecular karyotyping

Parental WTC-11 hiPSCs stocks were analyzed by comparative genomic hybridization (KaryoStat+ Genetic Stability Assay Service by ThermoFisher), confirming the expected diploid karyotype with a single segmental loss on chromosome Y (p11.2), a benign feature present in the healthy donor genome. The genomic stability of clones with homozygous *SMAD2* or *B2M* shRNAs was compared to the parental WTC-11 hiPSCs (Appendix Table S7). Genomic DNA was analyzed for ≥700,000 markers by low-resolution karyotyping through Illumina microarray (Life & Brain GmbH). Data was processed using GenomeStudio v.2 with the cnvPartition CNV Analysis Plugin version 3.2.1. Data were filtered for a minimum threshold of 6 for probe counts, 40 for confidence, and 300,000 for CNV length.

## Monolayer cardiomyocytes

WTC-11 hiPSC-CM differentiation was performed with minor changes of a validated protocol (Burridge et al, 2014; Bertero et al, 2019b). hiPSCs were dissociated as single cells in 0.5 mM EDTA in DPBS, and seeded at a density of $8.5 \times 10^4$ cells cm$^{-2}$ on 12-well Matrigel-coated culture dishes in mTeSR Plus supplemented with

5 μM Y-27632 (differentiation day, DD, -2). After 16 h, the media was refreshed and supplemented with 1 μM CHIR99021 (mesoderm priming, DD-1). From DD0 to DD6, media was switched to RBA (RPMI-1640 with glutamine, 500 μg mL$^{-1}$ BSA, and 213 μg mL$^{-1}$ ascorbic acid), which was supplemented with 6 μM CHIR99021 from DD0 to DD2 (mesoderm induction) and with 2 μM Wnt-C59 from DD2 to DD4 (cardiac mesoderm specification); no supplement was added from DD4 to DD6 (cardiac progenitor commitment). From DD6 onward media was switched to RPMI-B27 (RPMI-1640 with glutamine with 1× B27 supplement), with media changes every other day (cardiomyocyte specification). Spontaneous beating was first observed between DD7 to DD9. At DD14, hiPSC-CM were dissociated using 0.25% trypsin and 0.5 mM EDTA in DPBS and replated at a density of $1 \times 10^5$ cells cm$^{-2}$ on six-well Matrigel-coated culture dishes in RPMI-B27 with 5% FBS and 5 μM Y-27632. Media was replaced with RPMI-B27 after 24 h, and changed every other day through DD23 (cardiac maturation).

For TTN-mEGFP WTC-11 hiPSCs, differentiation was adapted to a seeding density of $2.5 \times 10^5$ cells cm$^{-2}$ onto Geltrex-coated plates in Essential 8 supplemented with 2 μM Thiazovivin (DD -2). CHIR99021 was reduced to 4 μM for mesoderm induction (DD0 to DD2), and cells were not replated.

For flow cytometry analyses, DD23 WTC-11 hiPSC-CMs were dissociated in 0.05% trypsin and 0.5 mM EDTA in DPBS, and live cells were stained with fixable viability dye eFluor 450 for 15 min at RT, followed by two washes with FACS buffer (5% FBS in DPBS). Cells were fixed with 4% PFA in DPBS for 15 min at RT, washed three times with FACS buffer, and once with FACS buffer containing 0.75% saponin. Cells were incubated with AF647-conjugated mouse anti-cTnT antibody or isotype-matched control antibody (both diluted 1:100) for 40 min at RT. Following two washes, cells were resuspended in FACS buffer and analyzed on a FACSCelesta flow cytometer, acquiring at least $1 \times 10^4$ events per sample. Data were analyzed with FlowJo to quantify single live cells expressing cTnT based on isotype control gating. DD12 and DD21 hiPSC-CMs derived from the TTN-mEGFP reporter line were instead only stained with fixable viability dye eFluor 780, and TTN-mEGFP reporter expression was analyzed on a FACSVerse flow cytometer.

## Cardiac organoids

hiPSCs were differentiated into left ventricle cardioids with minor modifications of a recently described protocol (Schmidt et al, 2023). hiPSCs were dissociated in TrypLE and seeded at a density of $2.5 \times 10^4$ cells cm$^{-2}$ on 24-well Matrigel/Geltrex-coated culture dishes in mTeSR Plus/Essential 8, supplemented with 5 μM Y-27632 (DD -1). 24 h post seeding, media was replaced with chemically defined media (CDM: 50% IMDM, 50% Ham's F-12 with Glutamax, 3 mg mL$^{-1}$ BSA, 1% chemically defined lipids, 0.004% 1-thioglycerol, and 15 μg mL$^{-1}$ Optiferrin) containing 6 ng mL$^{-1}$ FGF2-G3, 5 μM LY294002, 5 ng mL$^{-1}$ Activin A, 8 ng mL$^{-1}$ BMP4, and 5 μM CHIR99021 (DD0). After 36 h (DD1.5), primitive streak cells were dissociated in TrypLE, and 15,000 cells were seeded per well of low-attachment 96-well U-bottom plate in CDM containing 1.6 ng mL$^{-1}$ FGF2-G3, 8 ng mL$^{-1}$ BMP4, 10 μg mL$^{-1}$ insulin, 2 μM Wnt-C59, 50 nM retinoic acid, and 5 μM Y-27632. Cells were immediately

aggregated by centrifugation at 140 ×$g$ for 4 min. Media was refreshed daily, and Y-27632 was removed from the media from DD2.5, while Wnt-C59 and retinoic acid were removed from DD5.5.

Cardioids were imaged daily on an Incucyte SX5 using the spheroids module and a ×4 objective. Image segmentation was performed with filters for a minimum object area of $3 \times 10^4$ μm$^2$ and a maximum eccentricity of 0.9, and the largest object area was quantified. For flow cytometry analyses, DD7.5 cardioids were pooled and dissociated in 0.5% trypsin and 0.5 mM EDTA in DPBS; cells were then stained and analyzed on a Sony SH800S cell sorter exactly as described for hiPSC-CMs.

For immunofluorescence staining, DD7.5 cardioids were fixed in 4% PFA for 30 min at RT on a shaker, washed in DPBS twice, and incubated overnight at 4 °C in 30% sucrose/DPBS. Organoids were embedded in optimal cutting temperature (OCT) compound, then sectioned at a thickness of 12 μm, placed on a glass slide, and stored at −20 °C. For staining, section slides were washed for 15 min in DPBS at RT and permeabilized for 30 min at RT with a 4% goat serum, 1% Triton-X in DPBS solution (DPBST-G). After creating a hydrophobic barrier with a pap pen, primary antibodies— rabbit anti-WT1 (1:200), mouse anti-sarcomeric α-actinin (1:100), and rabbit anti-COL1A1 (1:100)—were diluted in DPBST-G and incubated in the dark in a humidity chamber for 3 h at RT. Slides were then washed in DPBS with 0.1% Tween 20 (DPBST2) for 30 min and incubated in the dark in a humidity chamber for 1 h at RT with secondary antibodies—goat anti-rabbit Alexa Fluor 633 (1:500) and goat anti-mouse Alexa Fluor 488 (1:500)—diluted in DPBST2. Following DPBST2 washes, nuclei were stained with Hoechst 33342 (1:2000 in DPBS) for 5 min in the dark, then washed again. Slides were air-dried for 5–10 min then mounted with ProLong Glass Antifade Mountant. Images were acquired with a Leica SP8 confocal microscope at ×63 magnification (objective: HC PL APO CS2 63X/1.40 oil) with the locator function to image the whole organoid (laser lines: UV for 405, Argon for 488, HeNe for 633).

## Forebrain organoids

Forebrain organoid differentiations from hiPSCs were performed as previously described (Velasco et al, 2019). Briefly, on differentiation day 0 (DD0), hiPSCs of 80–90% confluence were dissociated to single cells with Accutase, and $1.1 \times 10^4$ cells were aggregated per well of an ultra-low attachment 96-well V-bottom plate. From DD0 until DD18, cells were kept in Glasgow's MEM media supplemented with 20% KnockOut Serum Replacement, 0.1 mM MEM non-essential amino acids solution, 1 mM Sodium Pyruvate, 0.1 mM 2-Mercaptoethanol, 5 μM SB-431542, 3 μM IWR-1, and 20 μM Y-27632 (only until DD6). On DD18, organoids were transferred to a 10-cm dish and kept in suspension on an orbital shaker and in DMEM/F-12 supplemented with 1% GlutaMAX, 1% penicillin–streptomycin, 1% N2 supplement, and 1% chemically defined lipid concentrate. Media was prepared fresh and added every 3 days.

Immunofluorescence staining and imaging were performed on DD30 forebrain organoids following the methods described for the cardioids, with the following changes. Embedded organoids were sectioned at 20 μm thickness, washed three times with a 0.1% Tween 20 solution, then incubated for 1 h at RT in a solution of 0.3% Triton-X-100 and 6% BSA. Primary antibodies—rabbit anti-FOXG1

(1:1000) and mouse anti-PAX6 (1:400)—were incubated overnight at 4 °C followed by a 2 h incubation at RT with secondary antibodies—goat anti-mouse Alexa Fluor 568 and goat anti-rabbit Alexa Fluor 488. A solution of 0.1% Triton-X-100 and 2.5% BSA was used for antibody dilutions and washes before and after the secondary antibody incubation. Finally, sections were counterstained with DAPI, and coverslips were mounted with Mowiol. Images were acquired with a Leica SP8 confocal microscope at ×20 magnification (objective: HC PL APO CS2 20X/0.75 dry).

## Inducible gene knockdown

hiPSC pools were passaged not more than three times after genome editing, to minimize clonal drifting. Expression of inducible shRNAs in genome-edited hiPSCs was activated by adding 1 µg mL$^{-1}$ of tetracycline (tet) to the culture media. hiPSCs were tet-treated for 4 days before collection, and DD23 hiPSC-CMs were treated from DD0 through DD23. Samples collected at DD0 (primed hiPSCs), DD2 (mesoderm), DD6 (cardiac progenitors), and DD12 (early CMs) of hiPSC-CM differentiation were tet-treated from DD-4 through their collection. DD7.5 cardioids and DD30 forebrain organoids were either maintained in control conditions or tet-treated from DD-4 through their collection. For the study of knockdown kinetics, tet treatment was added and withdrawn at staggered intervals in hiPSCs and hiPSC-CMs (Fig. 7A,D). After tet withdrawal, wells were washed twice with DPBS containing Ca2+ and Mg2+, before addition of tet-free media. Control conditions included scrambled (SCR) and *B2M* shRNAs as negative controls to control for potential tet-related transcriptional challenges, and paired uninduced versus tet-induced comparisons to control for clonal variability.

## RT-qPCR

RNA was isolated from hiPSCs, hiPSC-CMs (DD0, 2, 6, 12, 21), and cardioids (DD1.5, DD7.5) using the Zymo Quick-RNA MiniPrep Kit. Up to 500 ng of RNA was transcribed using the High-Capacity RT cDNA kit following the manufacturer's protocol. RT-qPCR reactions containing 10 ng of cDNA, 1× PowerUp SYBR green master mix, and 0.5 µM forward and reverse primers were run on a QuantStudio 6 Flex Real-Time PCR system. *HPRT1* was used as the housekeeping gene, and all primers are listed in the reagents and tools table.

## Western blot

For assessment of tet knockdown and rescue kinetics, hiPSC were collected after 6 or 8 days, respectively (Fig. 7A). Cells were lysed from 12-well plates in radioimmunoprecipitation assay (RIPA) buffer containing 10 µL mL$^{-1}$ phosphatase inhibitor cocktail (PIC). Protein concentrations were quantified on a GloMax microplate reader using a Pierce BCA Protein Assay Kit. In total, 15 µg of protein diluted in lithium dodecyl sulfate (LDS) sample buffer and a prestained protein ladder were loaded onto 4–20% Polyacrylamide gels to be separated and then transferred onto a nitrocellulose membrane. Membranes were blocked for 1 h at RT in Tris-buffered saline with 0.3% Tween 20 (TBS-T0.3) with 5% Milk. Primary antibodies—rabbit anti-SMAD2 (1:1000) or mouse anti-Vinculin (1:10,000)—were diluted in TBS-T0.3 with 1% BSA and incubated

overnight at 4 °C. Membranes were washed three times with TBS-T0.3, then incubated for 1 h at RT with HRP-conjugated secondary antibodies—goat anti-rabbit (1:3000) and goat anti-mouse (1:5000)—diluted in TBS-T0.3. After the addition of Clarity Western ECL Substrate, the signal was detected on a ChemiDoc imager system. Band intensity was quantified using the Bio-Rad Image Lab software, and after which SMAD2 values were normalized to the Vinculin loading control.

## Clonal drifting assessment

Two batches of genome-edited hiPSC pools were kept in culture for extended passaging and collected as frozen cell pellets or lysed for RNA at early passage (P:3), middle passage (P:3 + 5), and late passage (P:3 + 10). DNA was extracted from the frozen cell pellet using the Monarch Spin gDNA Extraction Kit, following the manufacturer's instructions. DNA was eluted in 70 µL EB, then quantified and assessed for purity by Qubit fluorometry and TapeStation (D1000). The following steps are outlined in more detail in Appendix Protocol 2. In brief, 500 ng of gDNA was PCR amplified, dual-size-selected, quantified, and then indexed with dual-index primers (10X plate set A). Indexed libraries were then pooled with another gene expression library to increase the complexity, and 1 M reads were allocated to each library. Indexed libraries were pooled and sequenced using an Illumina MiSeq and one set of V3 reagents (150 cycles; R1: 76 cycles; i7: 10 cycles; i5: 10 cycles; R2: 76 cycles). *catcheR_step2QC* was applied to the FASTQ files to identify clones from the genomic PCR sequencing library and to assess their distribution across samples and time points. Clonal annotations were matched to the hiPSC phenotypes from the iPS2-10X-seq time course experiment (Fig. 2C), classifying clones as iPSC, iPSC-neuro, or mixed phenotype. Only previously annotated clones with assigned phenotypes were retained for analysis. The percentage of each clone and phenotype was calculated relative to the total number of clones identified at each time point, providing a measure of clonal heterogeneity over time. These analyses were performed on the two iPSC genome editing batches separately or combined.

## iPS2-sci-seq

iPS2-seq based on 2-level indexing sci-RNA-seq was performed according to the method detailed in Appendix Protocol 2, a custom modification of an established protocol (Cao et al, 2017). Specifically, DD23 hiPSC-CM nuclei were extracted, fixed in 4% PFA, and snap frozen in liquid nitrogen. Nuclei from two independent differentiations were thawed, dispensed at 5000 nuclei per well of a 96-well plate (48 wells each), and subjected to in situ reverse transcription (RT) using 96 barcoded and UMI-containing polythymidine (pT) and OPTtetR-specific (tetR) primers (mixed at 1:10 ratio and sharing indexes, well by well). Nuclei were pooled, DAPI-stained, and FACS-sorted at 25 nuclei per well into four 96-well plates. Following second-strand synthesis, tagmentation with Nextera Tn5, and bead cleanup, samples were consolidated in a 384-wp and gene expression and UCI-BC libraries were prepared by multiplexed PCR in each well using unique combinations of dual indexed P5 and P7 primers supplemented with P7A primers specific for the 3' end of OPTtetR cDNA (sharing indexes with P7, well by well, and mixed at 1:2 ratio). Libraries were pooled, purified,

size-selected, and subjected to NGS on an Illumina NextSeq 500 using a High Output Kit v2.5 (75 cycles).

## iPS2-10X-seq

iPS2-seq based on 10X Genomics 3' end scRNA-seq was performed according to the method detailed in Appendix Protocol 3, a custom modification of the manufacturer's protocol, with the following specifics. An optimization run was performed on two pools of hiPSCs and DD23 hiPSC-CMs, analyzed separately. The two batches were dissociated with TrypLE, washed three times in ice-cold culture media, filtered through a 40-μm cell strainer, counted (>95% vitality in all samples), pooled at equal amounts, and loaded in a chip G microfluidics reaction with $1.6 \times 10^4$ cells. Gene expression (GEX) libraries were generated according to the manufacturer's protocol. Briefly, following indexed RT of cells captured in Gel Beads-in-Emulsion (GEMs), the cDNA was purified, pre-amplified, fragmented, end-repaired and A-tailed, ligated to TruSeq adapter, and amplified by PCR with dual indexed primers. UCI-BC libraries were generated in parallel, starting from pre-amplified cDNA, which was amplified with an OPTtetR-specific primer carrying the TruSeq adapter (inner) and a reverse primer for all indexed cDNAs (Hill et al, 2018). GEX and UCI-BC libraries were pooled at 100:1 molar ratio and sequenced on an Illumina NextSeq 500 using two High Output Kits v2.5 (150 cycles).

For the time course experiment, the two batches of genome-edited hiPSC pools were combined together during differentiation and samples were collected at DD0, 2, 6, and 12 of hiPSC-CM differentiation. Cells were dissociated and processed as described above, except that $3.3 \times 10^4$ cells were loaded in a chip G microfluidics reaction, and GEMs were frozen after RT to be processed for library prep altogether. GEX and UCI-BC libraries were pooled at 100:1 molar ratio and sequenced on a NextSeq 2000 using one set of P3 Reagents (100 cycles).

For the DD7.5 cardioids in control conditions or tet-treated, the two batches of genome-edited hiPSCs pools were differentiated separately. In all, 24 cardioids per condition were pooled, dissociated with 0.5% trypsin and 0.5 mM EDTA in DPBS, washed twice in ice-cold DPBS with 1% BSA, and filtered through a 40 μm cell strainer. $2 \times 10^6$ cells per condition were labeled with barcoded, lipid-conjugated cell multiplexing oligos (CMOs), following the manufacturer's recommendations. Briefly, cells were washed once in room temperature DPBS 0.04% BSA, incubated for 5 min at RT in 100 μL of CMO solution, and washed three times with ice-cold DPBS 1% BSA. Dead cells were stained with fixable viability dye eFluor 780, and live cells were isolated by FACS with a Sony SH800S equipped with a 100 μm microfluidics chip and calibrated in targeted mode. An equal number of cells from the four conditions were pooled and loaded in a chip G microfluidics reaction with $2.3 \times 10^4$ cells. GEX, CMO, and UCI-BC libraries were pooled at 60:10:3 molar ratios and sequenced on an Illumina NextSeq 1000 using one set of P2 Reagents (100 cycles).

Forebrain organoids were differentiated in control or tet-treated conditions as two separate batches of genome-edited hiPSCs. Organoids were collected following a step-by-step protocol for human brain organoid single-cell dissociation (Arlotta et al, 2017). Briefly, 5–6 organoids at 1 month of age were transferred to a 60 mm dish containing a prewarmed papain and DNase solution. Organoids were minced with razors and incubated for 30 min at 37 °C on an orbital shaker and then mixed with a 1 mL tip several

times and returned for another 5–10 min to incubation at 37 °C. Next, the pieces were triturated 5–10 times with a 10 mL pipette and transferred to a 15-ml conical tube containing 8 ml final inhibitor solution and DNase. The tube was inverted a few times and left upright for a few minutes for larger debris to settle, and then the single-cell suspension was transferred to a new conical tube and centrifuged for 7 min at $300 \times g$ at 4 °C. The cell pellet was resuspended in 0.001% BSA–DPBS and passed through a 40-μm cell strainer. Live-cell counts were determined using a hemocytometer and trypan blue staining. CMO labeling was performed on 2 million cells per condition according to the manufacturer's protocol, plus an additional washing step described above for DD7.5 cardioids. An equal number of cells from the two conditions were pooled and loaded in a chip G microfluidics reaction with $4.9 \times 10^4$ cells. GEX, CMO, and UCI-BC libraries were pooled at 60:10:3 molar ratios and sequenced on an Illumina NextSeq 1000 using one set of P2 XLEAP Reagents (100 cycles).

All single-cell libraries were prepared using the Chromium Next GEM Single Cell 3' v3.1 protocol (CG000388). Indexed libraries were sequenced at a loading concentration of 650 pM using the recommended cycle configuration (R1: 28 cycles; i7: 10 cycles; i5: 10 cycles; R2: 90 cycles) with a 1% PhiX spike-in.

## iPS2-CITE-seq

Monoclonal homozygous or polyclonal hiPS2-seq *SMAD2* and iPSCs were collected for CITE-seq combined with Chromium Next GEM Single Cell 3' v3.1' at DD7.5 of cardioid differentiation. For the monoclonal clones, multiplexing was performed by CMOs to distinguish the tet treatment from the control conditions, as well as the two shRNA targets. The polyclonal pool was instead only multiplexed for treatment condition using TotalSeq-B hashtags. Beta-2-microglobulin () surface expression was assessed in both experiments by using the TotalSeq-B0057 antibody.

Cardioids were dissociated into a single-cell suspension in 0.05% trypsin and 0.5 mM EDTA in DPBS. Cells were washed and resuspended in DPBS with 0.04% BSA and then counted with a hemocytometer and trypan blue to assess cell viability. Antibody labeling was performed following 10X Genomics protocol (CG00149 Rev D) with the following adjustments. In total, $1 \times 10^6$ live cells per population were stained in 100 μL of TotalSeq-B0057 antibody (1:50) diluted in PBS and 1% BSA and incubation at 4 °C for 30 min, followed by a PBS and 1% BSA wash. The monoclonal samples were then labeled with CMOs following the 10X Genomics protocol (CG000391 Rev A). The polyclonal samples were instead incubated with TotalSeq-B0251 hashtag (control-condition) or TotalSeq-B0252 hashtag (tet-treated) for 30 min at 4 °C. Cells were washed three times with PBS and 1% BSA, then incubated for 15 min at room temperature with fixable viability dye eFluor 780. Cells were washed again, and live cells were sorted with a SONY SH800S sorter with a 100 μm chip in FACS tubes, then pooled together in equal quantities and loaded in a chip G microfluidics reaction with $2.98 \times 10^4$ cells.

Depending on the type of multiplexing applied, the single-cell gene expression (GEX), multiplexing (CMO), and antibody-derived tag (ADT) libraries were generated following two slightly varying workflows. For the monoclonal experiment, GEM formation, reverse transcription (RT), and post-RT cleanup were performed by following the Chromium Next GEM Single Cell 3' v3.1

workflows CG000390 Rev C. cDNA amplification used Feature Barcode Primer 3 and was performed for 11 cycles. Sample Index PCR was performed as recommended with indexes TT (GEX library), NN (CMO library), and NT (ADT library).

The polyclonal experiment instead followed workflow CG000317 Rev E and used Feature cDNA primers 2 and sample indexes TT and NN for the GEX and ADT libraries, respectively. In addition, the polyclonal ADT library underwent 1.2× single-sided size selection for construction of the UCI-BC library from the amplified cDNA (6.34 ng input) as described in Appendix Protocol 2. The first UCI-BC PCR was performed for 16 cycles, then, after 1.8× SPRIselect purification, the PCR was diluted to ~0.1 ng μL$^{-1}$ and indexed with dual-indexes TT.

All monoclonal and polyclonal libraries underwent a final purification with a 0.7X–0.9X SPRIselect size selection, then quantified by TapeStation and Qubit dsDNA HS assays. Libraries were pooled at 1:6 (CMO), 1:10 (UCI-BC), and 1:6 (ADT) of the GEX library. Sequencing was performed on an Illumina NextSeq 1000 using a set of P2 XLEAP reagents (100 cycles) for each experiment, exactly as described for iPS2-10X-seq.

## iPS2-multi-seq

iPS2-seq based on 10X Genomics multiome (scRNA-seq and scATAC-seq) was performed according to the method detailed in the manufacturer's protocol (CG000338, Rev F), and enrichment of the iPS2-seq barcode library was performed according to specific adaptations to the iPS2-10X-seq protocols as described in Appendix Protocol 3. Preparation of the hiPSCs for single-cell multiome was performed following the 10X Genomics demonstrated protocol for nuclei isolation (CG000365, Rev D). Two pools of iPS2-seq targeted hiPSC at early passage (P3) were detached using with 0.5 mM EDTA in DPBS, and incubated with DNase I (200 U mL$^{-1}$) to reduce clumping. They were then washed and filtered through a 40-μm flowmi cell strainer. Cell number and viability were assessed using trypan blue staining on a hemocytometer. hiPSC transfection pools were kept separately, and nuclei were pooled only prior to loading on the 10X Chromium. After cell lysis using freshly prepared buffer containing 0.1% Tween 20, IGEPAL CA-630, 0.01% digitonin, 1% BSA, and 1 U mL$^{-1}$ RNAse inhibitor, nuclei were isolated, washed twice with chilled wash buffer, and counted using both trypan blue and YOYO-1 (1:1000).

Nuclei were pooled 1:1, filtered through a 40-μm flowmi strainer, diluted to 7000 nuclei μL$^{-1}$ in 10× Nuclei buffer, and loaded into a chip J microfluidics reaction with $3 \times 10^4$ cells. Sequencing was performed on an Illumina NextSeq 1000 using two sets of P2 XLEAP reagents (100 cycles). ATAC libraries were sequenced at 400 million reads with 650 pM loading concentration and 1% PhiX spike-in (R1: 50 bp; i7: 8 bp; i5: 24 bp; R2: 49 bp). The GEX library was pooled with the UCI-BC library at a 20:1 ratio. Final pooled libraries were quantified using Qubit and loaded at 650 pM with 1% PhiX spike-in (R1: 28 bp; i7: 10 bp; i5: 10 bp; R2: 90 bp).

## Bioinformatics analyses

### Raw data processing

Illumina base calls for iPS2-sci-seq were converted to fastq and demultiplexed with bcl2fastq v2.20.0.422, using i7 and i5 PCR

indexes. Gene expression analyses were performed as previously described (Cao et al, 2017) using the bbi-sci pipeline, which was implemented in Docker and run with the *catcheR_scicount* function described in Appendix Protocol 4. Briefly, reads 1 and 2 were combined, adapter trimmed with TrimGalore 0.6.10, aligned with STAR 2.5.1b to human GRCh38 (GENECODE v21/Ensembl77) gene annotation, and filtered to have a MAPQ score equal of higher than 30 with samtools 1.7. Uniquely mapping reads were extracted, and reads with identical UMI, RT index, and tagmentation site were collapsed as PCR duplicates. Samtools and bedtools 2.27.1 were used to calculate the percentage of ribosomal RNA and to assign reads to genes. Reads were finally assigned to nuclei by further demultiplexing with the RT indexes. Cells with less than 500 UMI were filtered out as noise/low quality. Potential doublets were also filtered by removing nuclei with a total UMI count >2 standard deviations from the mean total UMI count in all cells. The remaining sparse count matrix was converted to a complete count matrix with the *catcheR_scicsv* function employing Monocle 2.22.

Illumina base calls for iPS2-10X-seq and iPS2-CITE-seq were processed with Cell Ranger 7.1.0 (Zheng et al, 2017) and Cell Ranger 9.0.1, respectively, starting with *cellranger mkfastq* for demultiplexing of gene expression (GEX), cell multiplexing oligo (CMO), antibody-derived tag (ADT) libraries, and unique clonal identifier and shRNA barcode (UCI-BC) libraries into fastq format. Individual GEX libraries were analyzed with *cellranger count*, while multiplexed GEX libraries were analyzed with *cellranger multi*, leveraging on CMO or ADT hashtag barcode counts for sample demultiplexing. In all cases, alignment to the human genome was performed with the STAR index human GRCh38.p13 (GENCODE v32/Ensembl98). Samples from the same experiment were merged with *cellranger aggr* with –normalize = none, adding experimental batch information to the aggregated feature matrix for subsequent batch correction.

Illumina base calls for iPS2-MULTI-seq were processed with Cell Ranger ARC 2.0.2, starting with *cellranger-arc count* that allows the combined cell barcode calling and the alignment of both the Gene-barcode matrix and the Peak-barcode matrix. This tool generates the feature-barcode matrix.h5 files that can be used for downstream analyses. Alignment to the human genome was performed with STAR index human GRCh382024A downloaded from the 10× download center.

### Perturbation assignment

Cells were assigned to shRNA perturbations using *catcheR*, following the steps detailed in Appendix Protocol 4. The package is comprised of R functions that execute all computation inside a Docker container produced with the Dockerfile generator CREDO (Alessandri et al, 2024), ensuring reproducible analyses. For iPS2-sci-seq, *catcheR_scicatch* was used to extract nuclei expressing single UCI-BCs identified from all fastq files. The same was achieved for iPS2-10X-seq, but using *catcheR_10Xcatch* to match cells to UCI-BC found in the dedicated fastq files. In both cases, for each read pair, read 2 was reverse-complemented and concatenated to read 1 (to simplify handling), and cell identifiers (IDs), unique molecular identifiers (UMIs), UCIs, and shRNA BCs were obtained by positional slicing, all using bash scripts. First, read 1 was sliced into UMIs and RT indexes, which represented the cell IDs for iPS2-10X-seq and part of the cell ID for iPS2-sci-seq (the rest being constituted by the PCR indexes). To remove noise, cell IDs were

compared to those in the cell by gene expression matrix for the same experiment, and only reads matching to a valid nucleus or cell were retained. Secondly, reads matching the expected sequence from the OPTtetR 3' end were selected by examining the start of read 2 (the end of the concatenated read), filtering reads not matching to a 23 bp reference sequence (5'-CGGCTCCCCCA-GATGAACGCGCC-3'). The sequences immediately following this reference (preceding it on the concatenated read) were sliced into shRNA BCs (8 bp) and UCIs (6 bp). All of these informations were loaded into an R dataframe, and the name of the shRNA associated to the BC was added. Reads containing an invalid BC were discarded. Reads were deduplicated by collapsing those containing the same cell IDs, UMIs, UCIs, and BC. PCR artifacts consisting of reads with the same UMI and cell ID but associated with different UCI-BCs were resolved by eliminating the less common read (by number of duplicates). UCI-BCs for a given cell ID were filtered based on three criteria: (1) the minimal number of supporting UMIs (more than 5 for iPS2-sci-seq and more than 10 for iPS2-10X-seq and iPS2-CITE-seq); (2) the minimal fraction of UMIs for a given UCI-BC compared to the total UMIs for all UCI-BCs matching to the same cell ID (more than 15% for all experiments); and (3) the minimal ratio between the UMIs for a given UCI-BC and the UMIs for the second most abundant UCI-BC in the same cell ID (5 for all experiments). The resulting list of UCI-BCs identified cell IDs reliably associated to a single shRNA, which were used to filter and annotate the cell by gene expression matrix for downstream analyses.

For iPS2-10X-seq, cells expressing no shRNA (integration of empty pAAV-Puro_siKD2.0) were identified using *catcheR_10X-nocatch*. Concatenated reads 1 and 2 were filtered based on a 23 bp reference sequence differing from the one of UCI-BC-containing cassettes by the last two nucleotides (5'-CGGCTCCCCCAGAT-GAACGCGTA-3'). After deduplication, empty BCs were filtered based on a UMI threshold of more than 5 for all experiments. Finally, cell IDs with no valid shRNA UCI-BC (all below the first two thresholds) but containing a valid empty BC were annotated as empty on the cell by gene expression matrix.

Reassignment of perturbations for clones engineered with a plasmid encoding an shRNA with barcode swap was performed by combining *catcheR_step1QC* and *catcheR_sortcatch*. The first function parsed the results of NGS quality control for the first pooled cloning step (Appendix Protocol 1) to verify the expected pairing of shRNAs and their barcode in individual bacterial clones. Reads were partially deduplicated based on diversity indexes (DI, used during the low-cycle PCR preceding library prep) and positionally sliced into shRNAs and UCI-BCs. To filter out NGS artifacts, only UCI-BCs counted above a certain threshold were retained (100 for our experiment), and the matching shRNAs were counted. UCI-BCs for which the most common shRNA was counted at least ten times more often than any other shRNA were assigned to such shRNA (this ratio was applied to control for PCR artifacts, as most UCI-BCs showed some degree of background pairing to many shRNAs, presumably as a result of template switching during library prep; Potapov and Ong 2017). *catcheR_sortcatch* used the output of *catcheR_step1QC* to reassign hiPSC clones (cell IDs expressing individual barcode-swapped UCI-BCs) to the correct shRNA (1 hiPSC clone in our experiment), changing the matrix annotation.

The distribution of perturbations in cells was compared to that of the final plasmid pool using *catcheR_step2QC*, which counted UCI-BCs in the NGS quality control data for the second pooled cloning step (Appendix Protocol 1). Reads were partially dedupli-cated based on DIs, positionally sliced to obtain UCI-BCs, and UCI-BCs counted below a certain threshold were removed to filter out NGS artifacts (1000 for our experiment).

### Cell by gene expression matrix annotation and filtering

Secondary data analysis was performed in R using software and custom scripts listed in the reagents and tools table and subsequently wrapped into *catcheR* functions (Appendix Protocol 4—Perturbation effect analysis). Cell by gene expression matrices annotated with shRNA perturbations and clonal IDs (UCI-BCs) were loaded on Seurat v5 (Hao et al, 2024). These were the output of *catcheR_scicatch* for iPS2-sci-seq, or the filtered feature bc matrix from *cellranger aggr* annotated and filtered based on the output of *catcheR_10Xcatch* run for individual iPS2-10X-seq samples. For iPS2-10X-seq experiments, clones were also annotated with the genome-edited hiPSC batch (determined from the cardioid experiment based on CMO labeling) and with the starting pluripotent transcriptional state (determined from the time course experiment as described below): clones composed of cells that clustered in iPSC or iPSC-neuro at 75% or more were annotated with the respective labels, while remaining clusters were annotated as "mixed".

Seurat *PercentageFeatureSet* was used to obtain the percentage of mitochondrial and ribosomal reads, and cells were filtered to retain only those with more than 200 total UMI, more than 200 expressed genes, and less than 30% or 20% mitochondrial reads (for iPS2-sci-seq or iPS2-10X-seq, respectively). Only protein-coding genes with at least 10 total counts in the dataset were kept. For iPS2-10X-seq, cells with a clonal ID counted less than 20 times in at least one of the time points or conditions were set aside. The data was scaled and normalized with Seurat *ScaleData* and *NormalizeData*. Seurat *FindVariable-Features* and *ElbowPlot* were used to enumerate the principal components (PCs) that explained the main source of data variance (15 for iPS2-sci-seq and 20 or 50 for iPS2-10X-seq experiments). Seurat *CreateAssayObject* was used to obtain the antibody capture of the iPS2-CITE-seq dataset as an additional layer of the Seurat object.

### Enrichment/depletion

At every time point, the fraction of each hiPSC clone was calculated over the total number of shRNA-expressing cells (that is, excluding empty cells since these are not wholly comparable). Changes in clonal fractions at a given time point were calculated relative to the experiment-matched baseline (hiPSCs for the DD23 hiPSC-CMs; DD0 for the DD2, 6, and 12 differentiation time course; DD0 of the time course for 3D cardioids; hiPSCs of iPS2-MULTI-seq for forebrain organoids). Significant changes were calculated per clone with a Fisher test on cell numbers considering the variables "time point versus baseline" and "clone of interest versus other clones", and the $P$ values were adjusted by the number of multiple comparisons using the Benjamini and Hochberg (B–H) method (significance cutoff of 0.05).

These analyses were performed either considering all hiPSC clones altogether, or only for clones with the same starting pluripotent transcriptional state (iPSC or iPSC-neuro).

Additionally, for the iPS2-10X-seq neural organoid experiment, only clones derived from perturbations targeting *CHD7*, *KMT2D* or *SMAD2* were considered for this analysis.

### Cell clustering

Filtered data from the Seurat v5 pipeline was transferred to Monocle 3 (Trapnell et al, 2014; Cao et al, 2019; Qiu et al, 2017) as a SingleCellExperiment object, pre-processed using *preprocess_cds* with 30 dimensions, and processed using *reduce_dimensions* to perform uniform manifold approximation and projection (UMAP) dimensionality reduction (Healy and McInnes 2024). Monocle 3 *align_cds* was used to correct batch effects for the two experiments aggregated into the iPS2-10X-seq time course; all the other datasets were not batch-corrected.

Cells or nuclei were clustered with Monocle *cluster_cells* using Leiden-based clustering, K nearest neighbor of 20, and resolution of $1 \times 10^{-3}$, $1 \times 8^{-4}$, or $1 \times 5^{-4}$ (for iPS2-sci-seq, iPS2-10X-seq on cardioids, or iPS2-10X-seq on brain organoids, respectively; Levine et al (2015)). For the time course iPS2-10X-seq experiment, two main hiPSC clusters were identified and labeled as unbiased hiPSCs (iPSC) and neuroectoderm-primed hiPSCs (iPSC-neuro), based on marker and differential gene expression analyses described below.

The time course iPS2-10X-seq experiment was also subclustered after subsetting samples by time point: hiPSCs and DD0 (both corresponding to a comparable undifferentiated state), DD2, and DD6, 12, and 23 (which formed a continuous trajectory), using the same parameters. Monocle 3 *top_markers* was run to identify the top 5 markers describing each cluster, and *fit_models* was used to perform gene regression analyses based on a negative binomial distribution for modeling and comparing gene expression differences across clusters. Monocle 3 *plot_cells* was used to visualize marker expression across clusters. This collective information allowed the tentative assignment of clusters to cell types.

### Cell type representation

Given a clustering or subclustering, cell type representation for a condition of interest was enumerated as the fraction of cells for each cluster in said condition. This distribution was compared to that of a control condition, cluster by cluster, to obtain fold changes. Statistically significant changes were identified through multiple Fisher tests on cell numbers, one per cluster of interest, considering the variables "condition of interest versus control condition" and "cluster of interest versus all other clusters"; the *P* values were adjusted by the number of multiple comparisons using the B–H method (significance cutoff of 0.05). This analysis was performed after subsetting cells by the perturbed gene (that is, aggregating all cells expressing one shRNA against the same gene) or by individual shRNAs. For the latter, only shRNAs represented by at least 70 cells in at least one of the time points were considered, to reduce the number of underpowered comparisons and thus increase the sensitivity of the analysis. For the same reason, only clusters represented by a minimum number of cells for at least one condition at one of the time points being analyzed were selected for statistical assessment.

For the time course experiment, the control condition was the *B2M* shRNA (DD2) or SCR shRNA (DD6, 12, and 23), and only clusters with 70 or more cells were statistically assessed. For the cardioid experiment, the control condition was the same clones of the condition of interest but cultured in the absence of tetracycline, and only clusters with 40 or more cells were statistically assessed. Additionally, the

cardioid experiment allowed for benchmarking the importance of tet versus control clone-matched analysis in comparison to using *B2M* or SCR controls. *P* values for each analysis were ranked, and individual comparisons were assessed for significance (B–H adj. $P \leq 0.05$) in none, one, two, or all the control settings.

For the iPS2-10X-seq forebrain organoid experiment, clusters derived from "iPSC" or "iPSC-neuro" were kept separately, and only data derived from perturbations targeting *CHD7*, *KMT2D*, or *SMAD2* were considered for further cluster analysis.

### Pseudotime

For the time course experiment, Monocle 3 *order_cells* was run on the subclustering of samples from DD6, 12, and 23, to create a pseudotime trajectory whose start point was selected based on cell type annotation (proliferating cardiac progenitors). Each cell from these samples was ordered by the resulting pseudotime, and this was used as a continuous variable to compare the distribution of cells in a condition of interest with that in a control condition. This analysis was performed after subsetting cells by the perturbed gene or in individual hiPSC clones, using SCR and *B2M* shRNAs as separate controls. Significant changes were assessed by a two-sided Kolmogorov–Smirnov (KS) test, and the resulting p-values were adjusted for multiple comparisons by the B–H method. To determine the magnitude and directionality of change, two B–H-corrected, one-sided KS tests were performed, and the KS statistic was plotted as is (significantly "higher" cumulative frequency distribution, indicating lower pseudotime) or as a negative number (significantly "lower" cumulative frequency distribution, indicating higher pseudotime). A significance cutoff of 0.05 was used for all tests.

### Gene modules

For iPS2-sci-seq, Monocle 3 *find_gene_modules* was run with resolution of $1 \times 10^{-2}$ to identify 59 gene expression modules (Traag et al, 2019). The expression of genes constituting each module was aggregated at the level of individual gene targets using Monocle 3 *aggregate_gene_expression*, and the difference between each gene target and the SCR control was calculated and Z-scored across the various targets. The maximum absolute Z-score per gene module was used to rank them based on variability within the experiment, and the top 10 were selected.

For the cardioid iPS2-10X-seq experiment, a similar analysis was performed, except that Monocle 3 *find_gene_modules* was run with a resolution range from $1 \times 10^{-6}$ to $1 \times 10^{-1}$ (identifying 72 gene expression modules), that the difference in gene module aggregated expression was calculated between control and tet-treated cells for each gene target, and that the top 15 modules were selected. Cell-level aggregated gene module expression was used as a continuous variable to compare the distribution of cells in a condition of interest versus a control condition. This was done at the gene target and clonal level by combining B–H corrected, two-sided, and one-sided KS tests as described for the pseudotime analysis, except that the control condition was represented by the same group of clones or the individual clone of the condition of interest, but cultured in the absence of tetracycline.

### scMAGeCK

scMAGeCK (Yang et al, 2020) was adapted for iPS2-sci-seq by providing shRNA barcodes as input (instead of CRISPR guide RNAs, as originally intended for this tool) and by considering the SCR shRNA as a negative control. *scMAGeCK-LT* was used to assess the correlation between gene-level perturbations and all

genes in the 10 most variable gene modules from the analysis described above (2007 total genes examined for 5 perturbations, each with 4 shRNAs). In total, 306 genes were identified as perturbed at a false discovery rate of less than 0.05.

### iPS2-multi-seq

Perturbation assignment was performed using *catcheR* with a threshold of 6 UMIs and a minimum of 15% of total UMIs per cell supporting the same UCI, requiring a ratio of 3 for confident assignment. These relaxed thresholds were necessary due to the inherent noise in the data, particularly the lower average UMI counts per UCI (~20).

Preprocessing and integration of scRNA-seq and scATAC-seq data were conducted using Signac v1.8.0 and Seurat v5. Chromatin accessibility data and transcription data from filtered_feature_-bc_matrix.h5 and atac_fragments.tsv.gz (generated by *cellranger-arc* previously described) were imported using *CreateSeuratObject* and *CreateChromatinAssay*. Quality control was performed using *PercentageFeatureSet*, *NucleosomeSignal* and *TSSEnrichment*, retaining cells with <40% mitochondrial content, >1800 ATAC counts, nucleosome signal <2, and TSS enrichment >3. scRNA-seq data were scaled with *ScaleData* and normalized with *NormalizeData* and pre-processed as follows. Highly variable genes were identified with *FindVariableFeatures* and selected the top 2000, and PCA was run with *RunPCA* using these features. The top 50 principal components were selected (based on the elbow plot) and used for UMAP embedding via *RunUMAP*. For scATAC-seq, peak features present in at least 5 cells were retained with *FindTopFeatures*. Dimensionality reduction was performed using TF-IDF normalization with *RunTFIDF* followed by dimensionality reduction via SVD with *RunSVD*. The first 20 LSI components were used to compute UMAP embeddings with *RunUMAP*. The two modalities were integrated using *FindMultiModalNeighbors* using the top 50 PCA (RNA) and 20 LSI (ATAC) components to build a joint UMAP representation. After integration, *catcheR*-annotated cells were subset for downstream analyses.

Motif analysis was performed by scanning accessible peaks using position frequency matrices (PFMs) from the JASPAR2020 database with functions *CreateMotifMatrix* and *AddMotifs*. Then motif deviation scores were computed with *RunChromVAR* to quantify TF activity across experimental groups. Marker gene analysis was conducted using *FindMarkers* on both RNA and ATAC assays to identify features differentially expressed or accessible between "iPSC" and "iPSC-neuro" groups. This function employed a Wilcoxon test and a log2 fold-change threshold of 1 for significance in the RNA data, and a logistic regression framework for the ATAC data. Significantly differential genes and peaks were retained for downstream motif enrichment.

Significant regions were filtered with *AccessiblePeaks*, and GC-matched background peaks were generated using *MatchRegionStats*. Then, TF motifs in group-specific accessible regions were tested using the *FindMotifs* function. RNA-derived markers highly expressed in "iPSC-neuro" were integrated with motif-level accessibility results to identify TFs that were both transcriptionally enriched and had significantly enriched motif accessibility in the "iPSC-neuro" group. Finally, GO enrichment analysis was conducted on TFs motifs enriched after ranking based on fold change in iPSC-neuro, then using the clusterProfiler package with the *gseGO* function, and the top 10 GO results were visualized using the

*barplot* function. Coverage plots were generated with Seurat v5 with the function *CoveragePlot* on the significant regions, and gene expression plots were generated with the *FeaturePlot* on the significant or selected genes.

### ATAC-based karyotyping

We used *epiAneufinder* to estimate CNVs from scATAC-seq data, following a recent publication (Ramakrishnan et al, 2023). For each clone included in the final results, a corresponding BAM file was generated based on cell IDs. Single-cell resolution was insufficient due to low ATAC read depth (85,356 average reads per cell, compared to 707,188 used by Ramakrishnan and colleagues), but aggregation by clonal ID allowed robust CNV profiling (median 3,366,857, range 650,907–13,228,813 fragments per clone). To exclude outliers, only BAM files containing between 600,000 and 15,000,000 reads were retained. These files were converted to SAM format, and cell IDs were replaced with clone names in the annotation. The files were then reconverted to BAM and merged. The resulting BAM file was processed with *sinto* 0.10.1 to generate an ATAC segment file, which was used as input for *epiAneufinder* with standard parameters, and *windowSize* of $1 \times 10^{-5}$, *minsizeCNV* set to 1, and $k$ set to 6.

## Data visualization and statistical analyses

Graphs were generated with R v4.3.1 (ggplot2, pheatmap, and viridis packages), which was also used for Fisher and KS tests. Additional graphs and statistical analyses were obtained with GraphPad Prism 10; the type of test and number of replicates are indicated in the figure legends. No blinding, randomization, or formal sample size estimation were performed, as all experiments involved standardized differentiation protocols and quantitative molecular assays not requiring these procedures.

## Resource availability

Plasmids generated in the study and required to implement the technology have been deposited to Addgene and are listed in the reagents and tools table. The custom shRNA pool and/or the engineered hiPSCs will be made available on request with appropriate MTA.

## Data availability

The dataset and computer code produced in this study are available in the following databases: iPS2-sci-seq—hiPSC-CMs: BioStudies E-MTAB-14102. iPS2-10X-seq— monolayer differentiation: BioStudies E-MTAB-14065. iPS2-10X-seq—cardioids: BioStudies E-MTAB-14066. iPS2-seq—hiPSCs clonal drift: BioStudies E-MTAB-15303. iPS2-multi-seq—hiPSCs: BioStudies E-MTAB-15332. iPS2-10X-seq—neural organoids: BioStudies E-MTAB-15308. iPS2-10X-seq - monoclonal cardioids: BioStudies E-MTAB-15307. iPS2-CITE-seq—polyclonal cardioids: BioStudies E-MTAB-15309. Monocle3 CellDataSet class files: Zenodo https://doi.org/10.5281/zenodo.11085619. Computer scripts and software: GitHub (https://github.com/alessandro-bertero/catcheR/tree/dev).

The source data of this paper are collected in the following database record: biostudies:S-SCDT-10_1038-S44320-025-00172-8.

## Peer review information

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

## Acknowledgements

We are grateful to Giancarlo Bonora for pilot analyses during the optimization experiments. We also thank Davide Cacchiarelli and Anna Osnato for critical feedback, and Francesca Anselmi for next-generation sequencing support (NGS UniTo Platform). This work was funded by the University of Washington Institute for Stem Cell and Regenerative Medicine (Innovation Pilot Award 2019; AB), the Giovanni Armenise-Harvard Foundation (Career Development Award 2021; AB), Additional Ventures (Single Ventricle Research Fund 2022; SM and AB), and the European Union (ERC, TRANS-3, 101076026; AB). Views and opinions expressed are however those of the author(s) only and do not necessarily reflect those of the European Union or the European Research Council Executive Agency. Neither the European Union nor the granting authority can be held responsible for them.

## Author contributions

**Elisa Balmas**: Conceptualization; Data curation; Software; Formal analysis; Investigation; Visualization; Writing—review and editing. **Maria L Ratto**: Software; Formal analysis; Writing—review and editing. **Kirsten E Snijders**: Investigation; Writing—review and editing. **Silvia Becca**: Investigation. **Carla Liaci**: Investigation. **Irene Ricca**: Investigation. **Giorgio R Merlo**: Resources. **Raffaele A Calogero**: Resources. **Luca Alessandri**: Supervision. **Sasha Mendjan**: Resources; Supervision. **Alessandro Bertero**: Conceptualization; Resources; Supervision; Funding acquisition; Investigation; Visualization; Methodology; Writing—original draft; Project administration.

Source data underlying figure panels in this paper may have individual authorship assigned. Where available, figure panel/source data authorship is listed in the following database record: biostudies:S-SCDT-10_1038-S44320-025-00172-8.

## Disclosure and competing interests statement

The IMBA filed a patent application (No. 21712188.8) on multi-chamber cardioids with SM named as inventor. SM is a co-founder and a SAB member of HeartBeat.bio AG, the IMBA cardioid drug discovery platform spin-off. The remaining authors declare no competing interests.

# Expanded View Figures

**Figure EV1.  Implementation of the iPS2-sci-seq pipeline.**

(**A**) Diagram of the two pooled cloning steps to generate *AAVS1* targeting vectors containing matched shRNAs and UCI-BCs. H1-TO: tetracycline inducible Pol III promoter; CAG: constitutive promoter; HA: homology arm. (**B**) Representative efficiency of pooled cloning step 1 measured in individual bacterial clones through colony PCR. (**C**) As in (**B**) but for bacterial clones resulting from pooled cloning step 2. -ve: parental vector negative control. (**D**) Genotyping of hiPSC clones for the *AAVS1* locus (loss-of-allele implies biallelic targeting), the expected targeting cassette junctions (OPTtetR indicates inducible barcoded shRNA targeting; NeoR signals co-targeting of the second allele), and random shRNA plasmid integrations (which can convey sensitivity to FIAU); the number of shRNAs expressed by each surviving clone (no crosses) can be thus estimated. Refer to Appendix Fig. S1 for details on the optimization steps related to (**B–D**). (**E**) Diagram of the key molecular biology steps tested for and implemented in iPS2-sci-seq. Briefly, during reverse transcription (RT), shRNA UCI-BCs are both captured by the standard pT primer also used for the rest of the transcriptome (exemplified on the right) and specifically enriched by an OPTtetR-specific primer (tetR). Following second-strand synthesis (RT) and tagmentation with Tn5 (which is only needed for the rest of the transcriptome), UCI-BCs are enriched again during PCR. P7A primer is the one ultimately chosen for the final protocol; P7B was tested both in its full length and a truncated version with no overhangs (P7BT). At each step, primers introduce the overhangs required for the subsequent steps (color-coded). The three fragment types are processed in parallel (except for one of the tests described in Appendix Fig. S2), and the same RT indexes and PCR dual indexes (i5 and i7) are inserted in molecules from a given nucleus, out of a large variety of combinations. Following NGS, UCI-BC are matched to single nuclei sharing the same indexes. (**F**) Sequence of the tetR-primed UCI-BC library described in (**E**); the pT-primed UCI-BC library is equivalent except for a larger insert encompassing the ~109 bp bGH polyadenylation site and ~30 bp polyadenosine tail. Primers used for paired-end sequencing on Illumina instruments, with index reads based on the reverse complement workflow. Read 1 is exactly 18 bp, while read 2 is at least 52 bp to support parallel sequencing of the transcriptome. (**G**) Evaluation of hiPSC-CM purity through flow cytometry for cTnT (TNNT2) for the two batches of CMs utilized for the experiments of the rest of this figure and Fig. 1E–K. Gates set on isotype control. (**H**) Filtering of valid nuclei barcodes based on the number of associated UMIs in an iPS2-sci-seq experiment with ~9,600 input hiPSC-CM nuclei (half from each batch from (**G**)). The threshold was set at 500 UMIs. (**I**) Dimensionality reduction and clustering of single nuclei transcriptomes, separated by CM replicate (compare to Fig. 1H). CM1-3: three subsets of CMs; P-CM: proliferating CMs; CF: cardiac fibroblasts. (**J**) Aggregated list of top 5 gene markers per cluster determined with Monocle 3. (**K**) Expression patterns of selected cluster markers: *TNNT2,* pan CM; *NPPB,* early CM; *TNNI1* and *MYH6,* mid CM; *MYH7* and *RYR2,* mature CM; *MKI67,* proliferating cells; and *FBN1,* cardiac fibroblasts. (**L**) Expression of selected hits from scMAGeCK analyses of genes significantly associated with LoF perturbations among those in highly variable gene modules (Fig. 1I–K). Expression presented as log normalized for examples of genes negatively regulated (*CACNA1C,* particularly in *NKX2-5* LoF) or positively regulated (*CTNNA1* and *MYOZ2,* in *KMT2D* and *CHD7* LoF, respectively). Source data are available online for this figure.

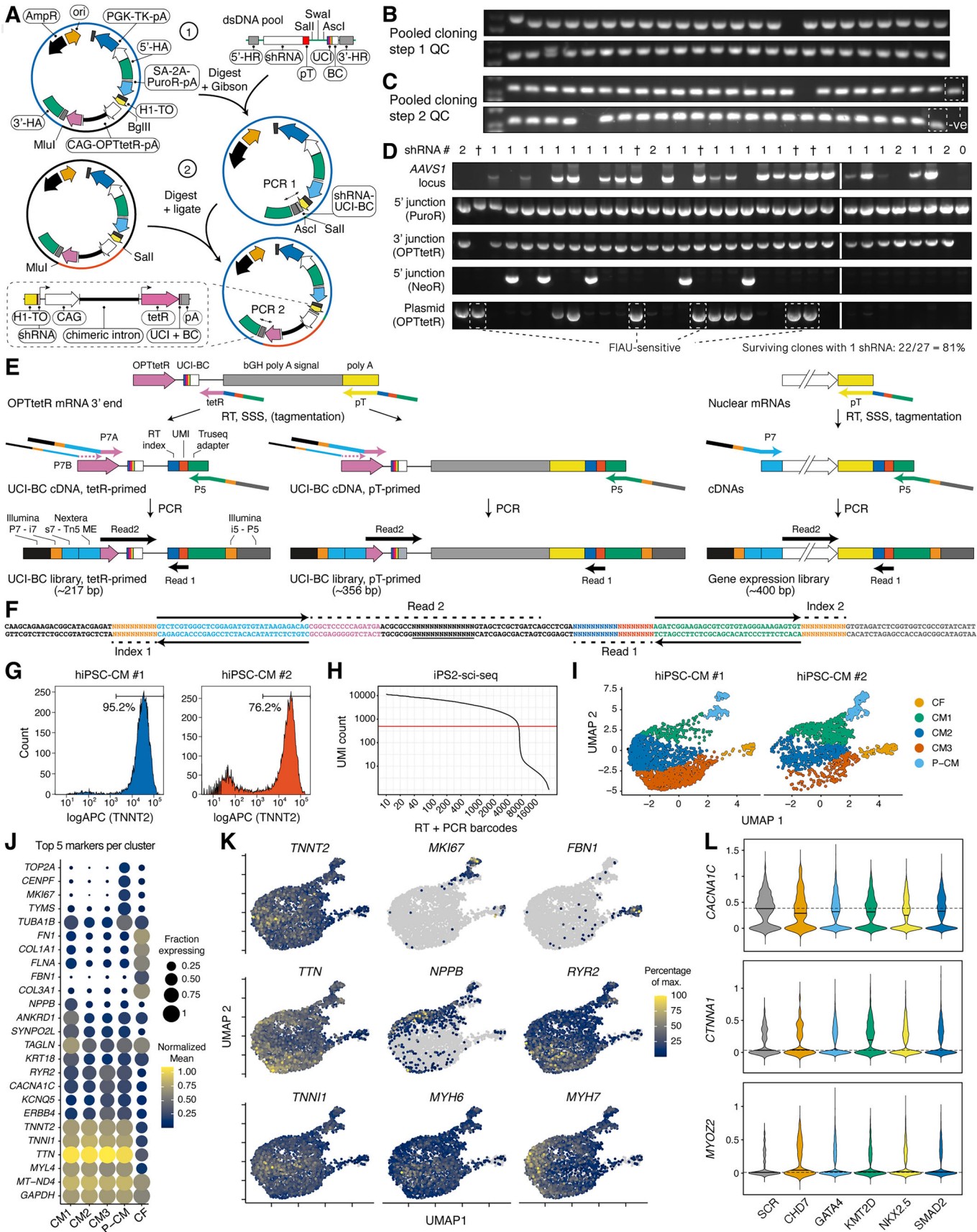

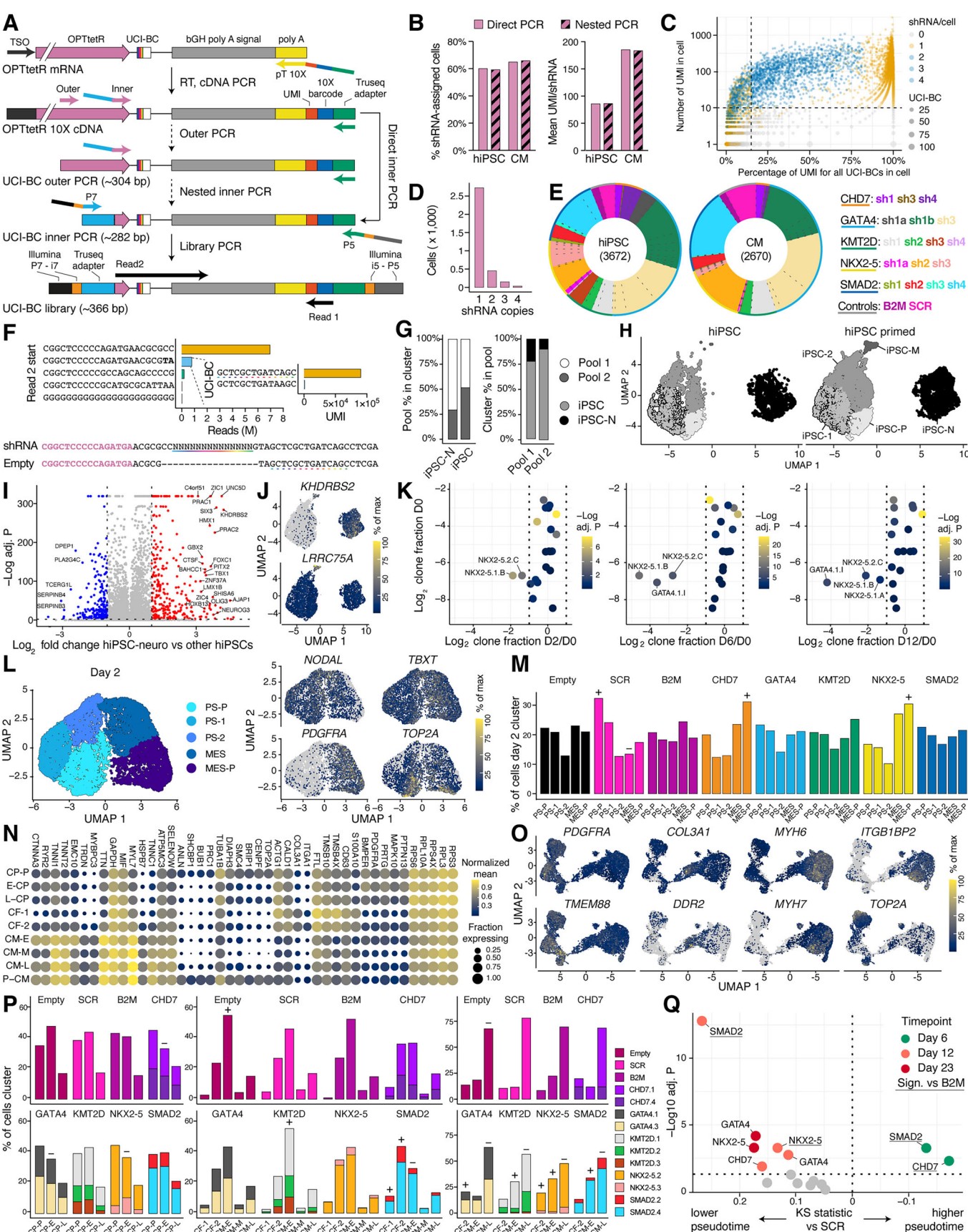

**Figure EV2. Optimization and implementation of iPS2-10X-seq.**

(A) Diagram of the molecular biology protocols tested for and implemented in iPSC-10X-seq. Briefly, an aliquot of 10X Genomics cell-barcoded and pre-amplified cDNA is processed to generate an shRNA UCI-BC library. UCI-BCs are amplified either in three steps ("nested" PCR with outer then inner primers, followed by NGS library prep), or in two steps ("direct" PCR with inner primer followed by NGS library prep). The resulting NGS libraries share the same structure, which is identical to the transcriptome-wide expression library obtained in parallel following the supplier's standard protocol, except that all read 2 sequences start in the same position of the OPTtetR 3′ UTR (same sequence described in Fig. EV1F). UCI-BC reads are matched to their originating single cell through the 10X Genomics cell barcodes sequenced as part of read 1. Both strategies are effective, as described in (B), but since the direct PCR is more expedient it is our reference protocol. (B) Comparison of direct and nested PCR strategies to detect UCI-BCs in the same cDNA samples from hiPSCs and CMs, as assessed by the fraction of cells that could be assigned to individual shRNAs (*catcheR_10Xcatch* with UMI count threshold >10) and the mean UMI count for the relevant UCI-BCs. (C) Filtering of UCI-BC counts (UMIs); representative results in day 23 CMs (compare to Fig. 1E). (D) Quantification of shRNA expression per cell; representative results in day 23 CMs (compare to Fig. 1F). (E) Distribution of clones expressing individual shRNAs in hiPSCs and CMs (compare to Fig. 1C and Fig. 1G). (F) Assessment of read 2 types in the UCI-BC NGS library: besides the expected sequence upstream of UCI-BCs (shRNA), the second most common sequence matches the unmodified pAAV-Puro_siKD2.0 plasmid (Empty) and thus identifies integrations lacking an shRNA; representative results from day 23 CMs. (G) Relationship between genome editing pool of origin and segregation of hiPSC clones in two main transcriptional clusters (iPSC-N: neuroectoderm-primed hiPSCs, globally depleted during CM differentiation; Fig. 2C). (H) Subclustering of hiPSCs (Fig. 2B) based on sample of origin (hiPSCs: standard pluripotency media; hiPSC primed: day 0 of differentiation, following one day of mesoderm priming in pluripotency media supplemented with 1 μM CHIR99021); iPSC-P: proliferating iPSCs; iPSC-M: mesoderm-primed hiPSCs; iPSC-1/2: unbiased hiPSCs. (I) Differential gene expression analysis for iPSC-N vs. all other iPSC subclusters (adj. P by negative binomial test with B–H correction, significance threshold of 0.05, and absolute fold-change cutoff of 2). (J) Expression patterns of selected markers of hiPSC subclusters: *KHDRBS2*, early neuroectoderm; *LRRC75A*, early mesoderm (also see Fig. 2D). (K) Enrichment/depletion analysis for unbiased hiPSC clones (not in the hiPSC-N subcluster); day 23 in Fig. 2E. Adj. P by Fisher test with B–H correction. (L) Subclustering of CM differentiation day 2 cells (Fig. 2B) and expression patterns of selected markers; PS-1/2: primitive streak; PS-P: proliferative primitive streak; MES: mesoderm; MES-P: proliferative mesoderm. (M) Gene knockdown-associated cell clustering changes at day 2 of differentiation; + or − indicate significantly increased or decreased cluster representation (adj. P < 0.05 by Fisher test vs. *B2M* with B–H correction). (N) Aggregated list of top 5 gene markers per subcluster in cells from day 6, 12, and 23 of differentiation (Fig. 2G). (O) Expression patterns of selected markers of day 6 to day 23 subclusters: *PDGFRA* and *TMEM88*, cardiac progenitors; *COL3A1* and *DDR2*, cardiac fibroblasts; *MYH6*, early CM; *ITGB1BP2*, mid CM; *MYH7*, mature CM; and *TOP2A*, proliferating cells. (P) Gene knockdown-associated cell clustering changes at differentiation day 6 (left), 12 (middle), and 23 (right); + or - indicate significantly increased or decreased cluster representation (adj. P < 0.05 by Fisher test vs. SCR with B–H correction; individual shRNA shown just in reference to Fig. 2H); CP-P/E/L: proliferating/early/late cardiac progenitor; CF-1/2: early/late cardiac fibroblasts; CM-E/M/L: early/mid/late cardiomyocyte. (Q) Gene knockdown-associated pseudotime alterations at differentiation day 6, 12, and 23 (refer to Fig. 2I–K); adj. P by two-sided Kolmogorov–Smirnov (KS) test of pseudotime cumulative frequency vs. SCR with B–H correction and significance threshold of 0.05 (underlined clones also significant vs. *B2M*; x axis based on one-sided KS tests).

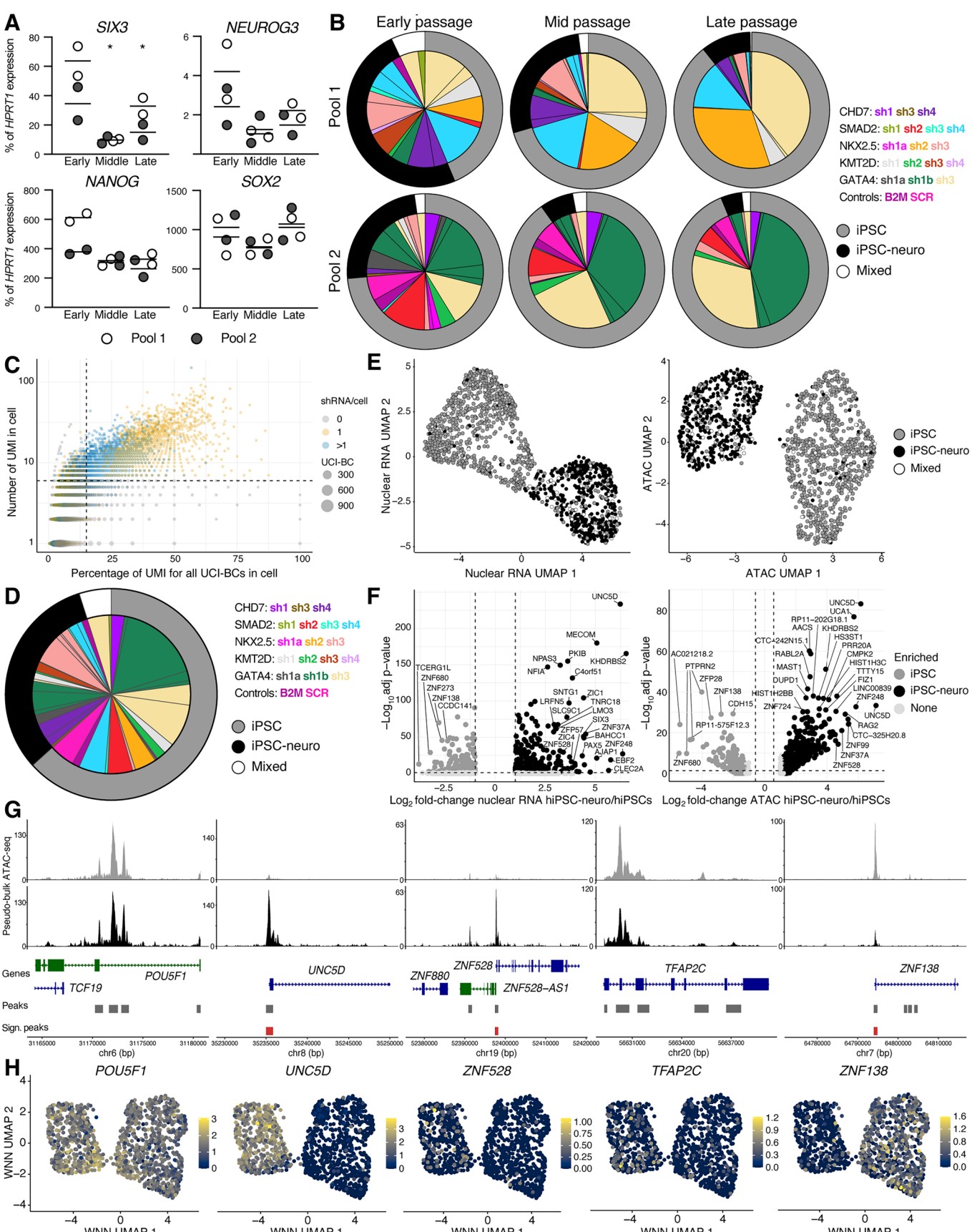

◀ **Figure EV3.   Dissecting the epigenetic basis of hiPSC clonal biases by iPS2-multi-seq.**

(A) Additional RT-qPCR of key regulators of neuroectoderm (*SIX3*, *NEUROG3*) and pluripotency (*NANOG*, *SOX2*) at various passages (early - p3, middle - p8, late - p13) of iPS2-seq genome-edited hiPSCs (see Fig. 3B). $N = 2$ cultures (the mean is indicated), * = adj. *P* of 0.031 and 0.022 vs. early, RM two-way ANOVA with Holm–Šídák's multiple comparisons. (B) Clonal composition for the two iPS2-seq hiPSC pools by UCI-BC DNA-seq across passages, plotted separately to show reproducibility of trends in Fig. 3C. Clones are ordered by clonal bias type (Fig. 2C). (C) Filtering of UCI-BC counts (UMIs) in the hiPSC pools analyzed at p3 by iPS2-multi-seq (Fig. 3A). (D) Clonal composition determined from iPSC-multi-seq, plotted as in (B). (E) Dimensionality reduction and clustering of matched nuclear RNA-seq (left) and ATAC-seq (right) from iPS2-multi-seq. Cells are color-coded by the clonal bias type (Fig. 2C). iPS-neuro: neuroectoderm-biased hiPSCs. (F) Volcano plots showing differential gene expression (left) and chromatin accessibility (right) between hiPSC-neuro and iPSCs from matched nuclear RNA-seq and ATAC-seq, respectively ($N = 1$). Wilcoxon Rank Sum test generated by *FindMarkers* function of Seurat package. (G) Aggregated chromatin accessibility tracks at exemplary loci showing no changes (*POU5F1/OCT4*, pluripotency marker; *TFAP2C*, neuroectoderm TF), increased accessibility in iPSC-neuro (*UNC5D*, neural differentiation marker; *ZNF528*, putative neuroectoderm TF), or increased accessibility in iPSC (*ZNF138*, putative iPSC TF). (H) Expression of the genes reported in (G) projected on WNN of integrated nuclear RNA-seq and ATAC-seq data (Fig. 3D). Source data are available online for this figure.

                                                  

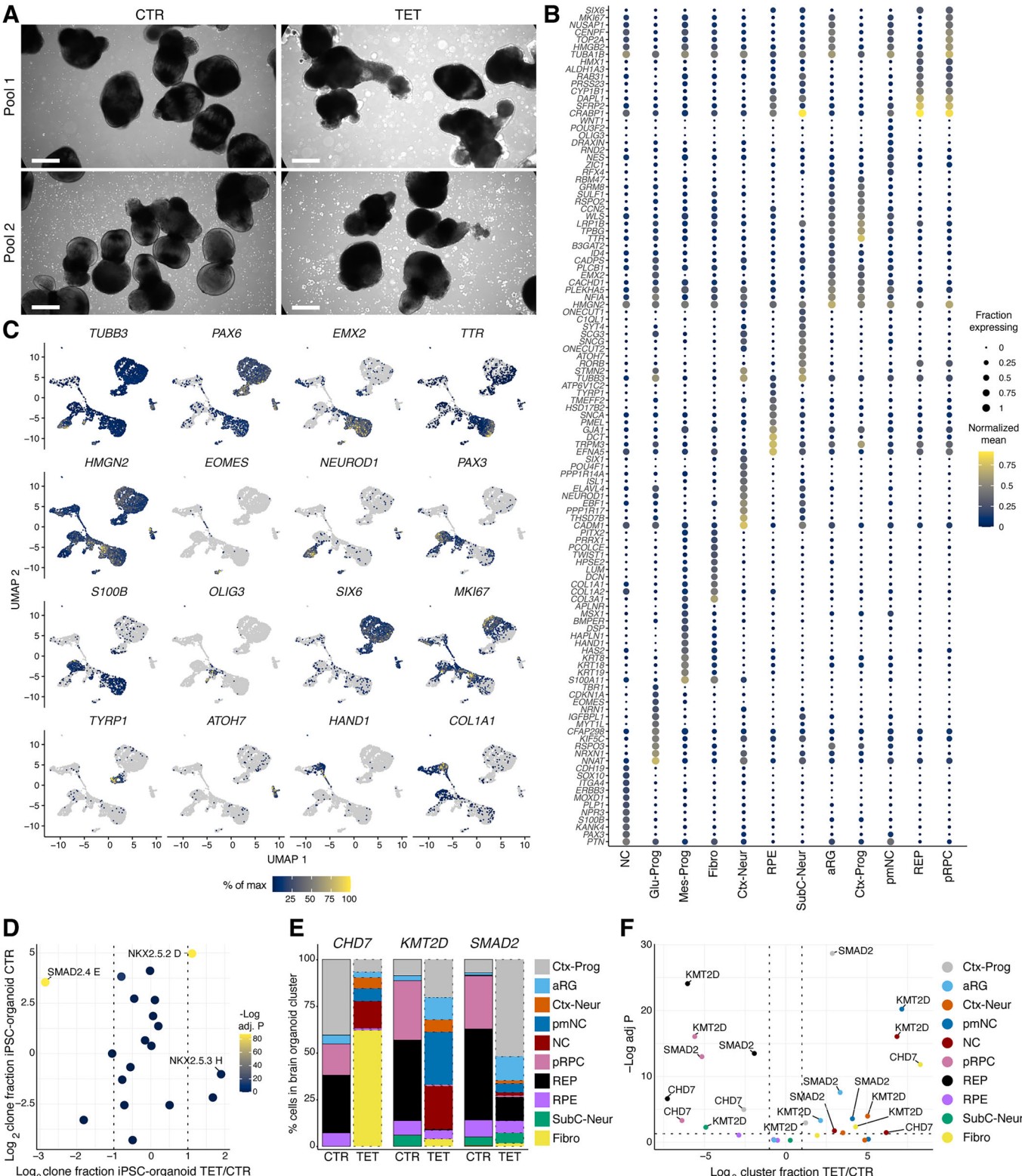

**Figure EV4.  hiPSC clonal biases alter cell fate in forebrain organoids.**

(A) Representative phase contrast images of day 23 forebrain organoids from two iPS2-seq genome-edited hiPSC pools under control and tet-treated conditions, illustrating reproducible morphogenesis. Retinal and cortical regions appear as dark and light multilayered zones, respectively; tet treatment impacts overall structure. Scale bars: 1 mm. (B) Aggregated list of top 10 gene markers per cluster from iPS2-10X-seq of day 30 forebrain organoids (Fig. 4C). Ctx-Prog: cortical progenitors; aRG: apical radial glia; Glut-Prog: glutamatergic progenitors; Ctx-Neur: cortical neurons; pmNC: pre-migratory neural crest; NC: neural crest; pRPC: proliferative retinal progenitor cells; REP: retinal epithelial progenitors; RPE: retinal pigmented epithelium; SubC-Neur: sub-cortical neurons; Mes-Prog: mesoderm progenitors; Fibro: fibroblast. (C) Expression patterns of selected markers: *TUBB3*, pan-neuronal; *PAX6*, neuroectoderm and retina; *EMX2*, dorsal telencephalon; *TTR*, Ctx-Prog; *HMGN2*, aRG; *EOMES*, Glut-Prog; *NEUROD1*, Ctx-Neur; *PAX3*, pan-NC; *S100B*, NC; *OLIG3*, pmNC; *SIX6*, pan-retinal; *MKI67*, proliferating cells (e.g., pRPC); *TYRP1*, RPE; *ATOH7*, SubC-Neur; *HAND1*, Mes-Prog; *COL1A1*, Fibro. (D) Clone-level enrichment/depletion in control *vs.* tet treatment for unbiased hiPSC clones Fig. 4D. Adj. *P* by Fisher test with B–H correction. (E) Paired clustering representation changes after tet-induced knockdown of genes expressed in forebrain. (F) Statistical analysis of clustering changes in (E); adj. *P* by Fisher test comparing tet *vs.* control, B–H correction, significance threshold of 0.05. Source data are available online for this figure.

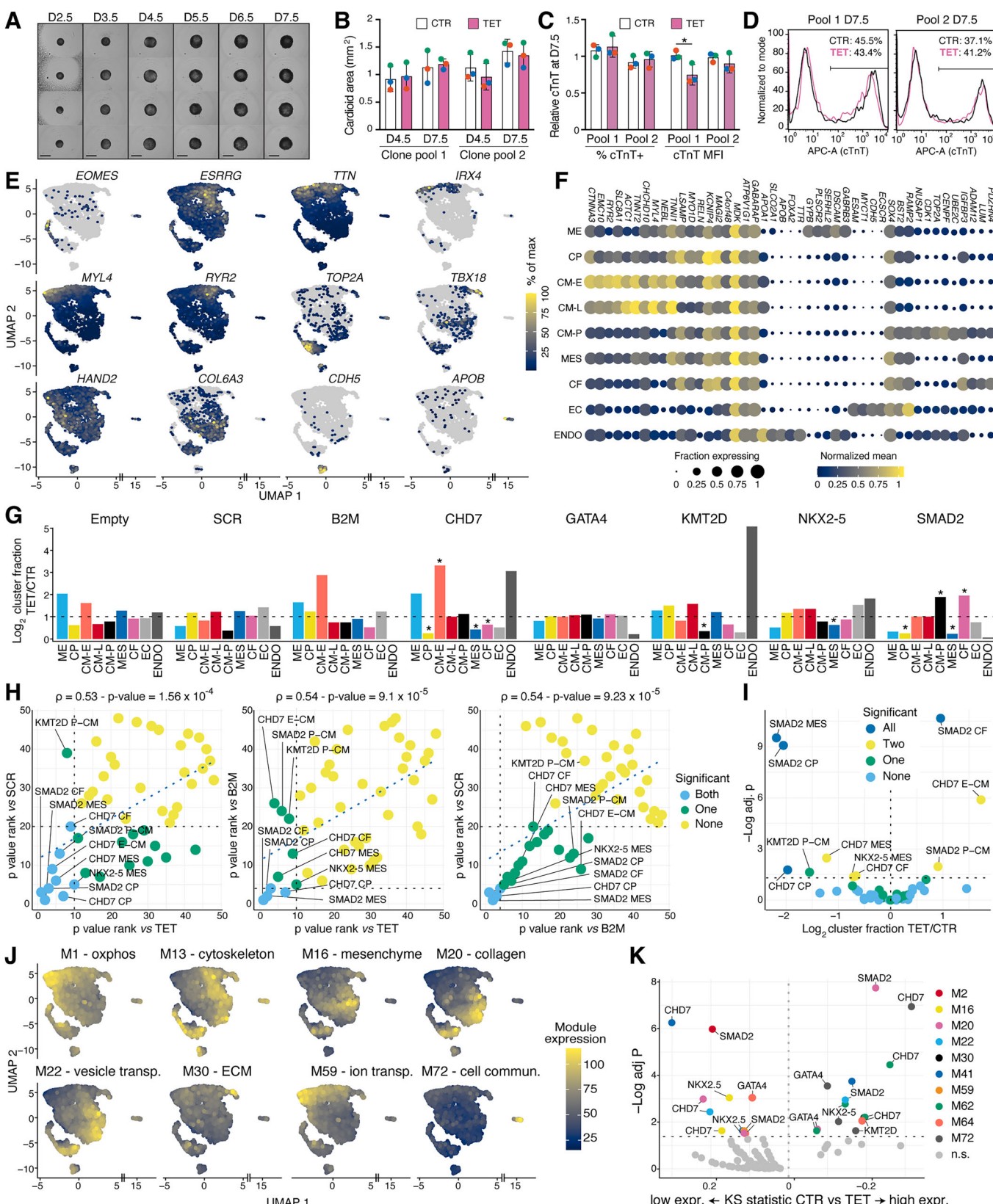

◄ **Figure EV5.  Application of iPS2-10X-seq in cardiac organoids.**

(**A**) Representative phase contrast images of left ventricular cardiac organoids (cardioids) from 24 h post aggregation to endpoint (Fig. 5A); scale bars: 1 mm. (**B**) Size of cardioids differentiated from two pools of iPS2-seq genome-edited hiPSCs in control conditions or tet-treated; data from 3 differentiations each with $N = 8$ cardioids/ condition (color-coded; no significant changes for CTR *vs.* TET). Error bars represent mean ± SD. (**C**) Differentiation efficiency of cardioids from the same experiments as (**B**) (fraction of cTnT+ cells and cTnT median fluorescence intensity, MFI, in cTnT+ cells, both calculated relative to the average values for control conditions in each differentiation, $N = 3$); * = adj. *P* of 0.0284 by RM two-way ANOVA with Holm-Šídák's multiple comparisons. (**D**) cTnT flow cytometry for cardioids from one of the experiments described in (**B**, **C**) that was analyzed by iPS2-10X-seq. (**E**) Expression patterns of selected cluster markers: *EOMES*, primitive streak; *ESRRG*, cardiac progenitors; *TTN*, pan-CMs; *IRX4*, ventricular CMs; *MYL4*, late CMs; *RYR2*, early CMs; *TOP2A*, proliferating cells; *TBX18*, epicardium; *HAND2*, mesoderm; *COL6A3*, cardiac fibroblasts; *CDH5*, endothelial cells; *APOB*, endoderm derivatives. (**F**) Aggregated list of top 5 gene markers per cardioid cluster (Fig. 5B). (**G**) Gene knockdown-associated cell clustering changes; * = adj. *P* < 0.05 by Fisher test for tet vs. control with B–H correction. (**H**) Benchmarking of cell clustering differences by control type: tet-treated cells compared to clone-matched controls or to shRNA controls (SCR or *B2M*). Dots represent individual cluster–shRNA comparisons ranked by Fisher test *P* values; dotted lines mark adj. *P* = 0.05 (B–H corrected). Color coding denotes whether each result was confirmed, missed, or falsely called in the alternative comparison. (**I**) Replot of (**H**) showing significance of tet vs. matched control, with color indicating reproducibility across comparisons using alternative control shRNA (SCR or *B2M*). This design yields the highest proportion of true positives (significant in >2 comparisons) and lowest rate of false positives (significant in only one). Adj. *P* by Fisher test with B–H correction. (**J**) Aggregated expression of a subset of most variant gene modules (Fig. 5H–J). (**K**) Gene knockdown-associated gene module alterations; adj. *P* by two-sided KS test of module expression for tet *vs.* control with B–H correction and significance threshold of 0.05 (*x* axis based on one-sided KS tests). Source data are available online for this figure.

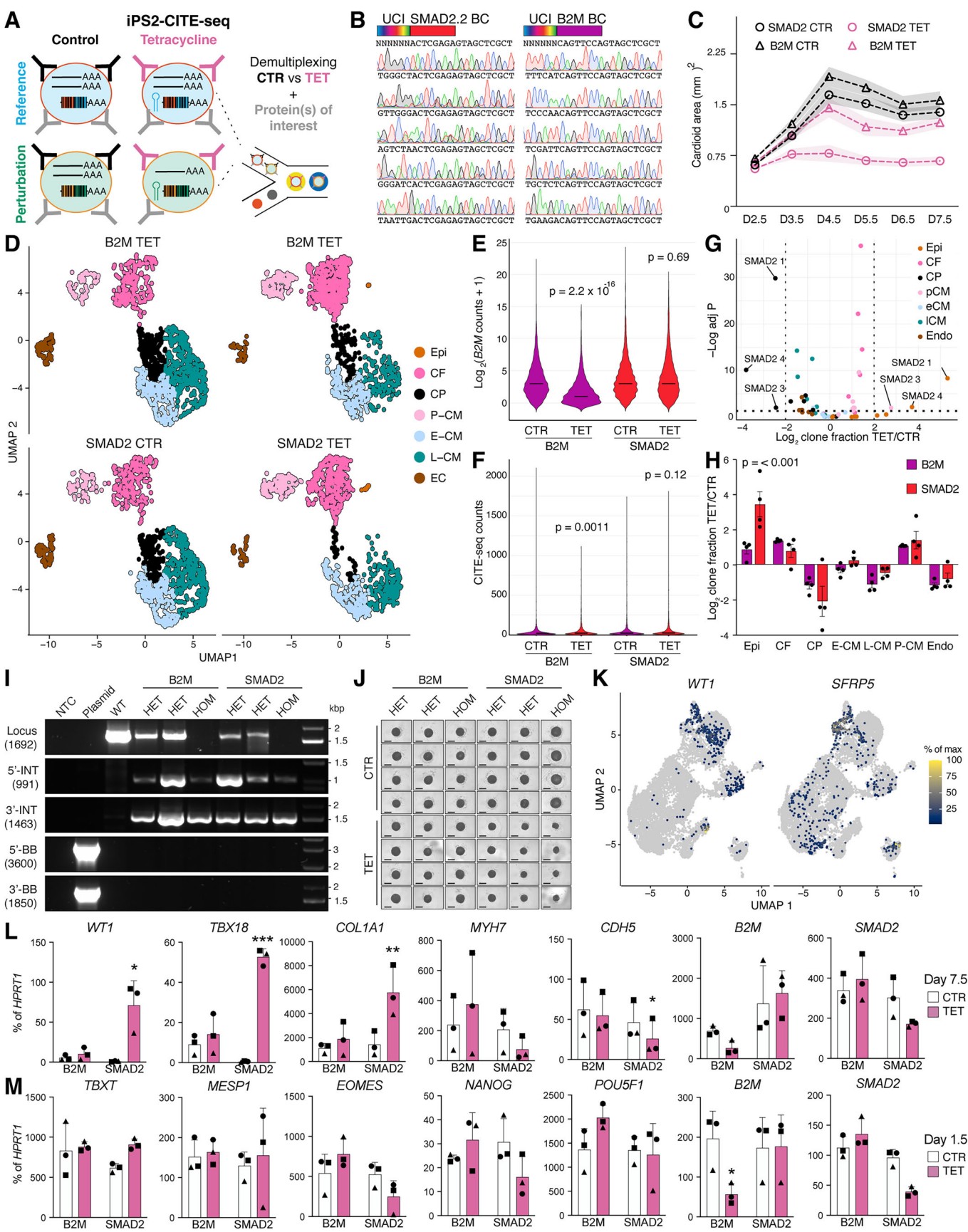

◀ **Figure EV6.** *SMAD2* **knockdown impairs cardiac organoid morphodifferentiation.**

**(A)** iPS2-CITE-seq relies on antibodies against a housekeeping cell surface protein (e.g., anti-ATP1B3) barcoded with two different sequences (black and pink), to distinguish control and tet-treated conditions. A reference condition (e.g., *B2M* shRNA) and one or more perturbations of interest (e.g., *SMAD2* shRNA) can be analyzed altogether and distinguished based on UCI-BCs (Fig. 1A). Additional CITE-seq antibodies (e.g., anti-B2M, gray) can be added for multi-omic assessment of protein levels in parallel to perturbation-resolved single-cell transcriptomes. All conditions can be pooled on a single microfluidic channel to provide a cost-saving solution that also eliminates batch effects. **(B)** Sanger sequencing validation of UCI-BCs in pool-cloned plasmids (Fig. EV1A), employed to generate *SMAD2* and *B2M* iPS2-seq genome-edited hiPSCs for the experiments described in **(C–H)**. **(C)** Time course analysis of cardioid size from clonal pools in control or tet-treated conditions. Representative data from one differentiation with $N = 24$ cardioids/condition. Day 7.5 cardioids were analyzed in **(D–H)**. **(D)** Dimensionality reduction and clustering of control and tet-treated cardioids from clonal pools, shown separately. Epi indicates epicardial cells identified by expression of *TBX18*, *PDPN* (Fig. 6F), *WT1*, and *SFRP5* **(K)**. Other cell type acronyms follow Fig. 5B and are annotated based on expression of markers listed in Fig. EV5E,F. **(E)** Violin plots of *B2M* mRNA expression quantified by scRNA-seq. adj. *P* vs. control by Wilcoxon signed-rank test, B–H corrected ($N = 1$ no TET control and $N = 1$ TET treated for each genotype). **(F)** As in **(E)**, but for B2M protein expression quantified by CITE-seq ($N = 1$ no TET control and $N = 1$ TET treated for each genotype). **(G)** Quantification of clustering changes in individual clones (1–4); adj. *P* by Fisher test comparing tet vs. control with B–H correction and significance threshold of 0.05. Unlike pooled screens (e.g., Fig. 5E), an additional threshold of effect magnitude was applied for this arrayed counter-screen, as it was sufficiently powered on a per-clone basis (absolute fold change >4). **(H)** Alternative visualization of the data plotted in **(G)**, with clones grouped by genotype to enable statistical assessment of intra-clonal reproducible clustering changes. Adj. $P < 0.001$ (***) by two-way ANOVA with Holm-Šídák's multiple comparisons. Error bars represent mean ± SEM. **(I)** Genotyping of *SMAD2* and *B2M* iPS2-seq clones with heterozygous (HET) or homozygous (HOM) integration of the inducible shRNA cassette, monitoring locus (loss-of-allele implies homozygous targeting), the expected targeting cassette junctions (5′/3′ integration, INT), and random shRNA plasmid integrations (5′/3′ backbone, BB). **(J)** Representative images of day 7.5 cardioids derived from the clones described in **(I)**. Scale bars: 800 µm. **(K)** Expression of additional epicardial markers in day 7.5 tet-treated cardioids from the homozygous *SMAD2* iPS2-seq clone, projected on the dimensionality reduction from Fig. 6E. **(L)** RT-qPCR quantification of markers of epicardial cells (*WT1* and *TBX18*), cardiac fibroblasts (*COL1A1*), cardiomyocytes (*MYH7*), and endothelial cells (*CDH5*), in day 7.5 cardioids derived from homozygous *SMAD2* and *B2M* iPS2-seq clones. shRNA target expression is also reported. $N = 3$ differentiations (symbols); *, *, **, *** = adj. *P* 0.0103 (*WT1*), 0.0425 (*CDH5*), 0.005, <0.001 vs. control by two-way RM ANOVA with Holm-Šídák's multiple comparisons. Error bars represent mean ± SD. **(M)** As in **(L)**, but for day 1.5 of the same differentiation runs (primitive streak stage), quantifying related markers (*TBXT/Brachyury*, *MESP1*, *EOMES*) and pluripotency genes (*NANOG*, *POU5F1/OCT4*), besides shRNA targets. * = adj. *P* 0.0446 vs. control by two-way RM ANOVA with Holm-Šídák's multiple comparisons. Error bars represent mean ± SD. Source data are available online for this figure.

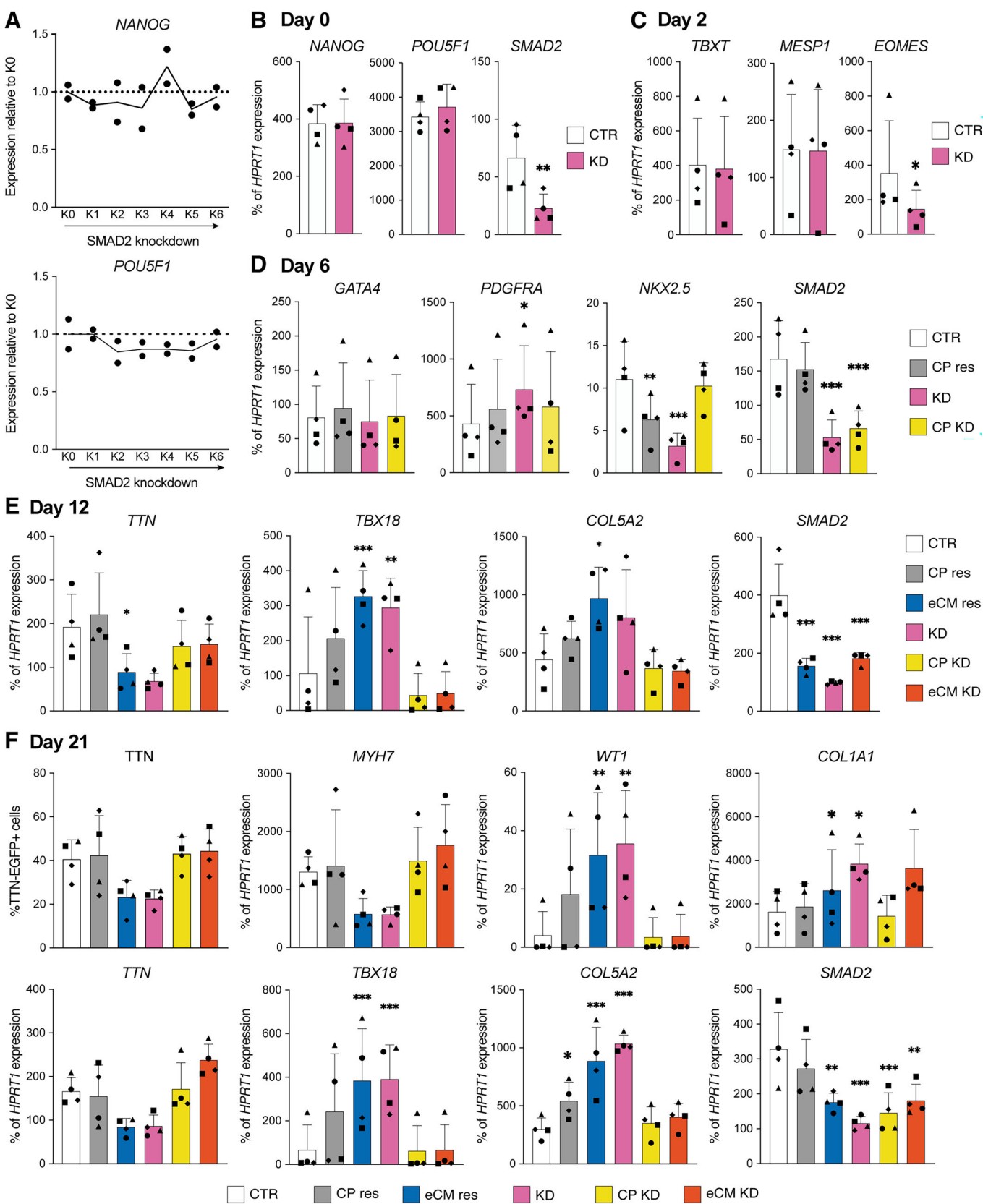

◄ **Figure EV7.** *SMAD2* **is required for cardiac progenitors patterning and specification.**

(A) RT-qPCR of pluripotency markers during the time course of *SMAD2* silencing and re-expression in an iPS2-seq homozygous clone examined according to the strategy of Fig. 7A. $N = 2$ cultures. This clone was used throughout this figure. (B) RT-qPCR of pluripotency markers and shRNA target at day 0 of hiPSC-CM differentiation in the experiment depicted in Fig. 7D. All other panels in this figure refer to other time points of this same experiment, with $N = 4$ differentiations (symbols); *, **, *** = adj. $P < 0.05$, 0.01, 0.001 vs. control by 1-way RM ANOVA with Dunnett's multiple comparisons. For *SMAD2* P = 0.0017. Error bars represent mean ± SD. (C) As in (B), for day 2 of differentiation, examining primitive streak markers (*TBXT/Brachyury*, *MESP1*, *EOMES*). P = 0.0435, error bars represent mean ± SD. (D) As in (B), for day 6 of differentiation, examining cardiac progenitors markers (*GATA4*, *PDGFRA*) and an early cardiac commitment gene (*NKX2-5*). $P = 0.0435$ (*), 0.002 (**) or < 0.001 (***). Error bars represent mean ± SD. (E) As in (B), for day 12 of differentiation, expanding on data of Fig. 7E for additional markers of cardiomyocytes (*TTN*), epicardial cells (*TBX18*), and cardiac fibroblasts (*COL5A2*). $P = 0.023$ (* *TTN*), 0.015 (* *COL5A2*), 0.001 (**) or < 0.001 (***). Error bars represent mean ± SD. (F) As in (B), for day 21 of differentiation, examining markers of cardiomyocytes, epicardial cells, and cardiac fibroblasts, plus TTN-mEGFP reporter expression by flow cytometry (top left; compare to Fig. 7E and (E)). In order of appearance, $P = 0.005$ or 0.002 (** *WT1*), 0.019 or 0.010 (* *COL1A1*), 0.035 (* *COL5A2*), 0.001 or 0.002 (** *SMAD2*) or < 0.001 (***). Error bars represent mean ± SD. Source data are available online for this figure.

