## [Peer Review File · Molecular Systems Biology]

Single Cell Transcriptional Perturbome in Pluripotent Stem Cell Models

Elisa Balmas, Maria Ratto, Kirsten Snijders, Silvia Becca, Carla Liaci, Irene Ricca, Giorgio Merlo, Raffaele Calogero, Luca Alessandri, Sasha Mendjan, and Alessandro Bertero

Corresponding author(s): Alessandro Bertero (alessandro.bertero@unito.it) , Elisa Balmas (elisa.balmas@unito.it)

Review Timeline:	Submission Date:	19th Oct 25
	Editorial Decision:	5th Nov 25
	Revision Received:	9th Nov 25
	Accepted:	17th Nov 25

Editor: Poonam Bheda

Transaction Report:

This manuscript was transferred to Molecular Systems Biology following peer review at another journal.

Single Cell Transcriptional Perturbome in Pluripotent Stem Cell Models

Elisa Balmas, Maria L. Ratto, Kirsten E. Snijders, Silvia Becca, Carla Liaci, Irene Ricca, Giorgio R. Merlo, Raffaele A. Calogero, Luca Alessandri, Sasha Mendjan, and Alessandro Bertero

Point-by-point response to the reviewers

We thank the editor and reviewers for their constructive and encouraging feedback. Over the course of an extensive revision process spanning nearly a year, we have fundamentally transformed this manuscript. The revised version introduces four entirely new main figures, each paired with corresponding supplemental figures, and improvements to existing figures through enhanced visual clarity and expanded analyses. We have also rewritten most of the main text to increase accessibility for a broad readership, while remaining within the journal's word count limits.

Critically, this revision elevates the original *Technology* article in both scope and depth. We now showcase three expanded applications of the iPS2-seq platform: (1) multi-omic single-cell profiling of chromatin accessibility and surface protein levels alongside gene expression; (2) arrayed screening formats; and (3) temporally controlled, stage-specific dissection of gene function. These capabilities are supported by an upgraded version of our bioinformatics pipeline, *catcheR*, which now spans the full experimental workflow: from perturbation design and assignment to statistical hit calling, facilitating broader adoption by the field.

Beyond the technological advancements, our revised manuscript delivers two major biological discoveries that would not have been possible without the unique features of iPS2-seq: (1) the identification of a stable, epigenetically encoded clonal bias in genome-edited hiPSCs that impairs both mesodermal and neuroectodermal differentiation (**Figs. 3–4** and **Figs. S3–S4**); and (2) the demonstration that *SMAD2* is a critical regulator of cardiac organoid morphogenesis, with a defined and narrow temporal requirement during cardiac progenitor patterning (**Figs. 6–7** and **Figs. S6–S7**).

These findings establish iPS2-seq as a uniquely versatile and powerful platform for performing clone-aware, phenotype-agnostic screens across diverse single-cell modalities and developmental contexts. In the point-by-point responses that follow, we detail the new experiments and analyses added to fully address all reviewer concerns. Reviewer comments are reproduced in magenta, references to revised text and figures are in bold, and quoted text is shown in *italic gray*.

Reviewer 1

Summary. Advances in genomics has led to the identification of genetic causes of disease. Traditional loss of function (LoF) screens often require prior knowledge of disease mechanisms, which is often lacking, making phenotype-agnostic screens using single cell RNA sequencing (scRNA-seq) an attractive alternative. Human pluripotent stem cells (hPSCs) offer the potential to study gene function in clinically relevant cell types, but they present unique challenges such as sensitivity to genotoxic nucleases and clonal variability. To address these challenges, the authors developed iPS2-seq, a method for inducible post-transcriptional silencing in hPSCs that includes numerous advantages in providing scalable perturbation assays for studies in developmental biology and disease. The authors provide a systematic description of the benefits and limitations of the technique to ensure transparency in the design of experimental approaches using this workflow. This method was demonstrated by studying CHD-associated genes in hiPSC-derived cardiomyocytes and multilineage cardiac organoids.

The authors have carried out a significant amount of work underpinning the study. In general the work is a tour de force and could be a major advance for studying genetic mechanisms of development and disease using iPSCs. However, there are a number of technical issues that appear to make biological interpretation of the results quite challenging. The study could also benefit from validation studies to confirm findings from the screening assays.

We thank the reviewer for their thoughtful summary and for recognizing the potential of our work to advance functional genomics in hPSC-based systems. We appreciated their clear articulation of both the strengths and the limitations of the original submission. In our revision, we substantially expanded the experimental framework to provide deeper biological insight, particularly into clonal variability and lineage priming, validated key findings (e.g., *SMAD2* knockdown), and addressed technical concerns.

Major concerns

Point R1.1A (We are breaking up this important remark to separately tackle each subpoint)

Karyotypic abnormalities the impact differentiation potential or proliferation rates of iPSCs are routinely a consequence of gene editing. Noting the clonal bias in terms of representation in the final data, to what extent is this accounted for by the gene KD vs potential karyotypic aberration that promotes proliferative effect that could bias the results irrespective of the genetic perturbation.

We fully agree with the reviewer that clonal variability represents a major confounding factor in hPSC-based screens. For this reason, “clonal awareness” was a foundational design principle of iPS2-seq (**lines 126–140**). This feature enables detection and filtering of clonal outliers, as shown by our identification of hiPSC clones with markedly reduced cardiogenic potential (**Fig. 2C-F**).

We also appreciate the reviewer’s point regarding the difficulty in disentangling genetic (e.g., karyotypic) from epigenetic sources of clonal bias in proliferation and differentiation. We agree this distinction remains underexplored in the field and were pleased to leverage iPS2-seq to begin addressing it. As detailed further in our response to point R1.1C, we performed additional enrichment/depletion analyses (**Fig. 5F–G**), which support the conclusion that iPSC-neuro clones are intrinsically impaired in cardiac differentiation irrespective of the genetic perturbation. To assess whether differential proliferation could underlie these trends, we computed cell cycle scores using the Seurat v5 *CellCycleScoring* function. We observed only minimal, non-significant differences in S-phase proportions between clone types (**Reviewer Fig. 1**), arguing against a major proliferative advantage driving clonal representation shifts.

Reviewer Figure 1: Stacked bar plot showing the distribution of cell cycle phases (G1, S, G2/M) among cells from the iPS2-multi-seq experiment described in **Fig. 3**.

Point R1.1B

Furthermore, in lines 414-441 the authors suggest a potential issue with "neuroectodermal" priming of edited cells. However, it may be that this result could be explained by a karyotypic abnormality. Is there previous evidence of this type of neuroectodermal priming in other clonal analysis studies?

We appreciate the reviewer’s thoughtful question. To our knowledge, no prior study has systematically examined clonal variability in hPSCs prior to loss-of-function perturbations. As reviewed in Table 2 of our previous work (Balmas et al., 2023) and confirmed by more recent literature searches, most scRNA-seq–based screens in hPSCs remain “clone-unaware” (Dräger et al., 2022; Leng et al., 2022; Replogle et al., 2020; Tian et al., 2019; Tian et al., 2021). One recent exception by the Treutlein and Knoblich labs (Li et al., 2023) tracked clones after perturbation, but did not investigate variability at the undifferentiated hPSC stage.

Nonetheless, clonal heterogeneity in hPSC cultures is well documented, and there is precedent for spontaneous lineage priming, including toward neuroectoderm (Wu et al., 2015; Yang et al., 2021). Most notably, the Palant and Powell labs reported a subpopulation of *ZIC1*-expressing "proliferative" hiPSCs and suggested a dynamic equilibrium between *ZIC1*-high and -low states (Nguyen et al., 2018). However, they did not track these cells clonally during differentiation, nor did they explore their mesendodermal potential. In contrast, our initial analyses

already indicated that *ZIC1* is consistently upregulated in individual clones that fail to generate cardiomyocytes, prompting a different interpretation: that these cells may not simply be transiently proliferative, but instead represent a stably neuroectoderm-primed subpopulation with reduced developmental plasticity. This effect may be further accentuated by the stress of genome editing, revealing a previously unrecognized and methodologically underappreciated challenge for functional screens in hiPSC models.

Prompted by this realization, we expanded and mechanistically strengthened our conclusions on the “iPSC-neuro” cluster (**Fig. 2B–D**) through three new follow-up experiments summarized here:

- We differentiated iPS2-seq clonal pools into telencephalic brain organoids (**Fig. 4** and **Fig. S4**). This experiment showed that clones from the “iPSC-neuro” cluster were able to generate neurons but exhibited a more restricted lineage output, including increased neural crest derivatives and the unexpected emergence of fibroblast-like cells. In contrast, clones from the “iPSC” cluster retained broader developmental competence across multiple telencephalic neuronal subtypes. These findings demonstrate that iPSC-neuro clones are not only biased against mesodermal differentiation but also exhibit aberrant neuroectodermal specification when subjected to protocols optimized for truly pluripotent, unbiased iPSCs.
- We tracked clonal dynamics over time by performing genomic PCR sequencing and RT-qPCR on iPS2-seq clonal pools starting from passage 3 hiPSCs (**Fig. 3A–C**, **Fig. S3A–B**). This analysis revealed a progressive depletion of iPSC-neuro clones and related markers by passages 8 and 13, supporting the interpretation that these clones are less self-renewing and more developmentally committed than their iPSC counterparts.
- We jointly profiled the transcriptome and epigenome of the iPSC-neuro cluster (**Fig. 3D–I**, **Fig. S3C–H**). This revealed increased chromatin accessibility of neural-associated transcription factors, including *ZIC1*, and their downstream targets, reinforcing the conclusion that iPSC-neuro cells constitute a distinct, epigenetically-primed subpopulation within the pluripotent stem cell pool.

Together, these experiments not only extend the biological validation of iPS2-seq to neuroectodermal lineages but also reveal the existence of stably committed subpopulations that can confound functional screens aimed at identifying regulators of pluripotency or differentiation. As discussed in **lines 568–570**, this finding suggests new strategies for improving screen fidelity, such as selectively identifying and removing lineage-primed clones using surface markers like *UNC5D*. Besides this, by enabling early detection of such confounders, iPS2-seq offers a reliable path toward more robust and interpretable genetic screens in hPSC models.

Point R1.1C

I would request karyotype of the parental and gene edited cells using a PCR-based genotyping kit that provides high resolution analysis of common iPSC karyotypic abnormalities. STEMCELL Tech genetic analysis kit can be used but it's only for bulk populations and doesn't provide more granular resolution.

We have confirmed that the parental WTC11 hiPSCs used in our study carry the expected diploid karyotype with a single segmental loss on chromosome Y (p11.2), a benign feature present in the healthy donor genome (**Reviewer Fig. 2**).

Reviewer Figure 2. Karyotype report for wild-type WTC11 hiPSCs, obtained using the KaryoStat+ Genetic Stability Assay Service by ThermoFisher.

We welcomed the reviewer’s suggestion to also assess the karyotype of iPS2-seq hiPSCs. However, as noted, conventional bulk-based assays do not resolve clonal-level differences in karyotype that could underlie variability in the pluripotent state. To address this, we performed combined single-cell ATAC-seq and RNA-seq, allowing us to dissect the contributions of both epigenetic and genetic variability to transcriptional changes (**Fig. 3**).

Using our *catcheR* pipeline, we mapped transcriptomic cells to their clonal perturbation and transferred this annotation to the ATAC-seq matrix. We then identified differential ATAC peaks between clones in the iPSC and iPSC-neuro clusters to examine the role of epigenetic variability in lineage priming, as already described in the response to the previous point.

In parallel, we used all ATAC peaks to infer potential copy number variations (CNVs) in individual clones with the recently developed tool *epiAneufinder* (Ramakrishnan et al., 2023). As described in lines **1339–1349**, CNV calls at the single-cell level were initially unreliable due to the limited sequencing depth of multi-omic scATAC-seq data. We therefore aggregated ATAC-seq reads at the clonal level—an analysis uniquely enabled by iPSC2-seq—achieving a median of over 3.3 million reads per clone (range: 650,907–13,228,813). These analyses revealed a few recurrent CNVs across clones, including in known hiPSC mutational hotspots and shown (**Fig. 3E**). However, none of these CNVs were associated with the iPSC-neuro state, supporting the conclusion that this priming is predominantly driven by epigenetic rather than genetic variability.

Notably, this experiment also served as a proof-of-principle for the iPSC2-multi-seq workflow, which enables clonal, phenotype-agnostic loss-of-function studies that integrate transcriptomic and epigenomic readouts at single-cell resolution.

Point R1.1D

The authors should also passage the iPSCs for multiple passages (e.g. 10) and/or differentiate the cells without tetracycline and evaluate barcode representation at the start and end of that experiment to determine any clonal drift over time or differentiation in the absence of gene knockdown. In general, a careful analysis of clonal-level karyotype would be very helpful as well as a functional readout of any clonal biases that are independent of gene perturbation

We fully anticipated that extended passaging of hiPSCs could lead to clonal drift driven by epigenetic or genetic differences in proliferation, even in the absence of loss-of-function perturbations. To minimize this source of bias, we performed all functional experiments within three passages (p3) from clonal derivation, a detail now explicitly stated in the Methods section (**lines 882–883**).

To directly assess clonal drift, as anticipated in the response to point R1.1B above, we passaged iPSC2-seq hiPSCs up to passage 13 and analyzed clonal representation at passages 3, 8, and 13 by gDNA sequencing of the shRNA barcodes (**Fig. 3A–C; Fig. S3A–B**). Besides revealing that iPSC-neuro clones are progressively depleted with passaging, these analyses found that a small number of clones became increasingly dominant, likely reflecting proliferation advantages. This trend was independently confirmed in two transfection pools (**Fig. S3B**). Based on these results, we updated our protocol to include three best-practice recommendations (**lines 2367–2379**): (1) minimize passaging before initiating experiments; (2) ensure sufficient clonal diversity per perturbation to allow outlier detection and filtering; and (3) consider evaluating for iPSC-neuro clones depending on the differentiation context.

We also tested whether clonal expansion biases might arise during differentiation independently of gene knockdown. Since cardiac differentiation involves early cell cycle exit, we reasoned that proliferation-driven clonal drift would be minimal. To evaluate this, we compared clonal representation between day 0 hiPSCs and day 7.5 cardioids cultured without tetracycline (CTR), and further examined differences between TET and CTR conditions (**Fig. 5F–G**). Figure 5F shows that clones in the “iPSC” cluster maintained stable representation throughout differentiation in the absence of tetracycline, while iPSC-neuro clones were consistently depleted—regardless of TET treatment—consistent with impaired cardiogenic competence rather than selective expansion. Importantly, **Figure 5G** demonstrates that a control vs TET comparison enables robust identification of perturbation-specific effects without requiring prior knowledge of clonal biases such as iPSC-neuro. This highlights a key strength of the iPSC2-seq design: the ability to disentangle true loss-of-function phenotypes from intrinsic clonal behavior.

Point R1.2

The controls for various analyses seem to be different across the study. It appears that the authors use a vehicle treated vs tet treated comparison in the cardoid assays. But in the monolayer diff data, it appears that the scrambled and B2M were used as controls. The rationale for the choice in control is not clear. Both are useful and provide important QC for different variables in the workflow. Can the authors provide any evidence of the best control for

these studies that maximises biological interpretation. For example, the vehicle vs tet control system may have important benefits of accounting for karyotypic differences and epigenetic differences between the cells of the same barcode. The authors suggest this in the text, but I could not find any data supporting the claims. A clearer demonstration and justification of controls is needed.

We appreciate the reviewer's comment and agree that the choice of control is a critical factor in interpreting iPS2-seq experiments. Indeed, we deliberately used two types of control designs to demonstrate the flexibility of the platform: (1) comparisons between cells expressing shRNAs of interest and those expressing control shRNAs (e.g., SCR or *B2M*), and (2) comparisons between the same clonal populations cultured with or without tetracycline induction (TET vs. CTR). As noted by the reviewer, the latter design offers the key advantage of internally controlling for both genetic and epigenetic clonal variability.

To move beyond theoretical considerations, we directly compared these control strategies using the cardiac organoid dataset. Specifically, we assessed how the TET vs. CTR design performed relative to using SCR or *B2M* shRNAs as standalone controls, focusing on their impact on statistical analyses of cell clustering. This comparison revealed that unmatched shRNA controls introduced inconsistencies: certain perturbations appeared significant only *versus* SCR (false positives), while others were missed entirely when compared to *B2M* (false negatives). In contrast, comparisons between matched clonal populations with or without tetracycline yielded more consistent results, with higher concordance and stronger effect sizes for true positives (Fig. S5H-I).

These findings support the conclusion that TET vs. CTR comparisons provide a more rigorous and internally controlled design, particularly in the context of clonal variability. However, as discussed in lines 555–562, this approach comes with higher sequencing costs. By presenting this side-by-side comparison, we aim to equip readers with the evidence needed to select the most appropriate experimental design based on their budget and desired level of analytical precision.

Point R1.3

The knockdown efficiency for each target gene is not shown, at least as far as I can tell. Can the authors provide single cell level information about clonal consistency of gene depletion at all sampled differentiation time points in knockdown conditions compared to the control cells. It would be nice to compare knockdown against different controls: e.g. tet vs vehicle treated samples (to demonstrate that within the same clones, knockdown occurs with tet activation) as well as comparing tet vs scrambled + tet (to demonstrate any effect of tet on gene knockdown). Also, across all cells with the same gene knockdown, is the knockdown consistent? Is there any reason to include a cell exclusion criteria accounting for efficiency of gene knockdown so that data interpretation is cleaner?

We appreciate the reviewer's request for a systematic analysis of knockdown efficiency across individual clones and time points. We agree that, in principle, this type of analysis would be valuable for interpreting gene function. However, in the specific context of iPS2-seq, it should be taken into consideration that transcript-level estimates of knockdown from scRNA-seq data are not a reliable metric, particularly when applied at the single-cell or clone level, due to the following technical and biological limitations:

- Dropout effects and shallow coverage in single-cell RNA-seq. The iPS2-sci-seq platform, like other droplet-based single-cell RNA-seq methods, is inherently limited by sparse transcript coverage and zero inflation. These technical issues disproportionately affect low-abundance transcripts, such as the transcription factors and epigenetic regulators targeted in our study. Thus, target silencing quantification is highly noisy.
- Limited cell numbers per clone. Due to the high cost of single-cell experiments, achieving deep coverage (>50 cells) per clone within a given sample is often impractical. This limitation is further compounded in heterogeneous systems like cardiac organoids, where diverse cell types may differentially express the gene of interest. As a result, reliable quantification would require subsetting cells by both clone and cell type, further reducing statistical power and interpretability.
- Disconnect between transcript levels and knockdown efficacy. Unlike CRISPRi, which represses transcription, shRNAs act post-transcriptionally *via* cytoplasmic RNA interference. This introduces two key complications: (i) nuclear RNA measurements (as captured in iPS2-sci-seq and iPS2-multi-seq) do not reflect cytoplasmic RNA degradation, and (ii) even cytoplasmic RNA levels may not fully represent functional knockdown, as shRNAs can also block translation without inducing RNA decay (Gu and Kay, 2010; Huntzinger and Izaurralde, 2011). Thus, mRNA abundance is not a reliable surrogate for shRNA efficacy in this context.

Having clarified these caveats, we nonetheless provide the expression levels of all target mRNAs in **Reviewer Fig. 3**, using iPS2-10X-seq data from 2D-differentiated cardiomyocytes at day 6, a time point at which all target genes are detectably expressed. We compared transcript levels in knockdown *versus* the SCR shRNA control. As expected, knockdown was more readily detectable for the highly expressed structural gene *B2M*, whereas transcription factors and epigenetic regulators showed more variable or subtle reductions, reflecting both their low endogenous expression and the aforementioned technical limitations of scRNA-seq in quantifying such genes without prohibitively deep sequencing. This issue is especially relevant for shRNA-mediated knockdowns, where transcript levels are often reduced but not eliminated, unlike in CRISPR-based knockout systems.

Reviewer Figure 3. Expression of shRNA-targeted genes (*B2M*, *SMAD2*, *GATA4*, *CHD7*, *KMT2D*, and *NKX2-5*) at day 6 of 2D cardiac differentiation, subsetted by the identified cell types. For each gene, expression in the corresponding shRNA-targeted population is compared to that in cells expressing the SCR (scrambled) control shRNA by Wilcoxon test.

While this analysis is visually plotted here for the reviewer's benefit, we also summarize target gene expression across all experiments by cell type and perturbation in **Table S5**.

To provide a more direct and reliable assessment of gene silencing, we also performed a CITE-seq experiment to measure protein levels in individual clones (**Fig. S6A**). Due to current reagent limitations, this analysis was feasible only for *B2M*, a cell surface antigen with commercially available antibodies. Nevertheless, the results demonstrated robust protein-level depletion in *B2M* knockdown clones, confirming the effective silencing achieved by the iPS2-seq platform (**Fig. S6D-F**).

Point R1.4

In general, I found that the biological interpretation of the data was not well developed. A more compelling case for the biological insights gained from the study would be beneficial. On this point, while I recognise the study is focussed on method development, validation of some findings is important to illustrate reproducibility outside of the data generated from the screen. As discussed above, there could be technical reasons pertaining to the screening method that account for the findings. For such a high impact study, I would request that authors validate at least one major finding independent of the screening results. For example, the most logical validation would include the *SMAD2* KD impact on CF vs CM differentiation.

We thank the reviewer for raising this important point. As per the reviewer's suggestion, we selected *SMAD2* to strengthen the biological interpretation of our screen. In brief, we studied *SMAD2* function using both polyclonal

and clonal iPS2-seq lines analysed by a combination of morphometric, gene expression (RT-qPCR and scRNA-seq), and immunolabeling (FACS and immunofluorescence) assays (**Fig. 6** and **Fig. S6**). These demonstrated that knockdown leads to reduced cardiac organoid size and impaired cardiomyocyte differentiation. Notably, we also uncovered an unexpected enrichment of epicardial-like cells, revealing a non-cell-autonomous role for SMAD2 that was not detected in pooled screens.

We also leveraged the inducibility of iPS2-seq to dissect the stage-specific requirement for SMAD2 in 2D differentiation. Temporal perturbation experiments revealed that SMAD2 is dispensable for pluripotency exit and mesoderm induction but is specifically required during the transition from cardiac progenitor to cardiomyocyte (**Fig. 7** and **Fig. S7**). These findings both establish SMAD2 as a key regulator of lineage commitment in human cardiac development and illustrate the analytical versatility of the iPS2-seq platform. Full results are detailed in the revised manuscript (**lines 494-524**).

Minor concerns

Point R1.5

1) Careful editing is needed to check for incomplete sentences and grammar and spelling issues throughout - most/all of the entry sentences for the Design section are incomplete sentences so suggest revising - (also see lines 100, 104, 111, grammar errors throughout lines 139-154). The use of many parenthetical phrases disrupts the flow of sentences throughout the manuscript, especially the abstract, so would recommend rephrasing to make more effective flow of the narrative.

We thoroughly revised the manuscript with the assistance of a native English speaker from the university language center. In addition to correcting grammatical errors, reducing excessive parenthetical phrases, and making other stylistic improvements, we introduced the following changes to improve narrative flow and better convey the main message of the study: (1) we rewrote the Abstract to clearly state the goal of the method without including excessive technical detail; (2) we moved non-essential technical descriptions from the Results to the Supplementary Protocols and shortened earlier sections to make room for new data; and (3) we revised the Discussion to integrate the new findings and clarify their significance in light of the updated results.

Point R1.6

2) Need citations for this sentence on line 175: We included the top predicted shRNAs against GATA4 and NKX2.5 twice to ensure their representation despite an anticipated strong negative selective pressure due to the established role of these two genes in cardiomyocyte development.

We have added references supporting the essential roles of GATA4 and NKX2.5 in hPSC-cardiomyocyte development (Anderson et al., 2018; Ang et al., 2016). The original source for shRNA validation (Moffat et al., 2006) is cited in **Table S1**, along with a link to the Sigma-Aldrich website where targeting information is publicly available. Please note that as part of our efforts to streamline the Results section, this sentence has been moved to the Methods (**lines 708–713**).

Point R1.7

3) In lines 312-324 the methods for barcode filtering are described, as they are in the methods. I couldn't find any data in the main or supplemental figures pertaining to the sensitivity of the barcode detection workflow in determining inclusion/exclusion of the cells, particularly around the specificity of the barcodes.

Table S3 was updated to include additional statistics on barcode assignment across all datasets, including: (1) the percentage of cells assigned to at least one UCI; (2) the fraction of cells with exactly one reliable shRNA integration (usable cells); and (3) the proportion of cells with multiple shRNA assignments. We summarize key values in **Reviewer Table 1**, which also includes the fraction of cells reliably assigned to exactly two shRNAs (see also point R1.9).

To assess the sensitivity of our barcode assignment pipeline, we compared scRNA-seq-based detection to ground-truth genotyping of shRNA integrations. Across iPS2-10X-seq datasets, 46%-77% of cells were confidently assigned a single shRNA barcode, compared to 81% single integrations observed by genomic PCR (**Fig. S1D**). This indicates that despite the stringent criteria applied to filter barcode associations to ensure specificity, and thus minimize false-positives, our method captures the majority of true-positive events. Of note, however, sensitivity was lower in nuclei-based platforms (e.g., iPS2-sci-seq and iPS2-multi-seq), where 27–52% of cells were reliably assigned, consistent with reduced barcode transcript capture and higher ambient RNA typical of nuclear RNA-seq. Overall, these analyses demonstrate that our pipeline provides robust, though platform-dependent, performance in recovering specific shRNA-to-cell assignments.

Protocol	Sample	% Usable cells (1 shRNA)	% Cells with 2 or more shRNA	% Cells with 2 shRNA
iPS2-sci-seq	Day 23 CMs	52%	7%	1%
iPS2-10X-seq	hiPSCs	66%	24%	18%
iPS2-10X-seq	Day 23 CMs	71%	15%	11.3%
iPS2-10X-seq	Day 0 iPSCs	65%	20%	9.2%
iPS2-10X-seq	Day 2 MES	68%	22%	10.0%
iPS2-10X-seq	Day 6 CPs	47%	15%	4.8%
iPS2-10X-seq	Day 12 CMs	53%	15%	4.9%
iPS2-10X-seq	CTR cardioid	76%	2%	1.7%
iPS2-10X-seq	TET cardioid	77%	2%	1.6%
iPS2-10X-seq	CTR neurons	46%	6%	1.5%
iPS2-10X-seq	TET neurons	65%	28%	1.5%
iPS2-multi-seq	hiPSCs	27%	28%	1.2%
iPS2-CITE-seq	B2M, SMAD2 CTR	69%	12%	1.45%
iPS2-CITE-seq	B2M, SMAD2 KD	61%	11%	1.45%

Reviewer Table. 1: Percentage of cells with 1 reliable UCI or with 1 or more UCI integrated.

Point R1.8

4) The use of ZFNs is okay but not common. Are there differences in gene editing efficiency for the pooled hPSC populations if using recombinant Cas9, which is far more common than ZFNs.

We opted for a ZFN-based approach because we have previously shown that it enables >95% on-target integration at the *AAVS1* locus in hPSCs (Bertero et al., 2016; Bertero et al., 2018b). That said, our *AAVS1* targeting vector is fully compatible with other programmable nucleases, including TALENs and CRISPR/Cas9 systems available through Addgene (e.g., #59025, #59026 from González et al., 2014 and #129726 from Li et al., 2019).

To directly address the reviewer's question, we compared the efficiency of *AAVS1* editing using our standard ZFN system *versus* a commercially-available Cas9-sgRNA ribonucleoprotein complex. As shown in **Reviewer Fig. 4**,

both systems yield similarly high editing efficiencies in our hands. We have added this information to our Supplementary Protocols (**lines 2312–2321**) to give users the flexibility to choose their preferred editing system.

Reviewer Fig.4: Transfection efficiency between the ZNF system and two Cas9-sgRNAs showed similar efficiency.

Point R1.9

5) The use of the second vector encoding the neo selection cassette is an excellent approach to ensure only a single shRNA is integrated. However, heterotypic interactions pertaining to gene-gene interactions are poorly understood, and this system could provide a mechanism for having individual cloned hPSCs with shRNAs targeting more than 1 gene - even diverse combinations of genes, in addition to the single integration events. Can the authors provide any simple examples that illustrate the utility of this platform for studying heterotypic TF interactions?

We appreciate the reviewer's suggestion. The iPS2-seq platform could indeed be adapted to study combinatorial gene perturbations, such as heterotypic transcription factor interactions, by co-delivering two barcoded shRNA libraries using separate AAVS1-targeting vectors with distinct selection markers (e.g., puromycin and neomycin). We now briefly highlight this future possibility in the revised Limitations of the current study (**lines 628-631**).

To explore this concept empirically, we searched for rare dual-integration events within our iPS2-10X-seq dataset from cardiac organoids, focusing in particular on potential interactions between GATA4 and NKX2-5 (as also noted in point R2.2). While we did identify proof-of-principle evidence that dual integration can occur (**Reviewer Table 1**), the number of confidently assigned double-shRNA cells to a specific condition was too low to support further analysis (**Reviewer Table 2**). Nonetheless, this observation confirms that the approach is technically feasible and lays the groundwork for future systematic studies using a tailored dual-perturbation design.

Reviewer 2

Balmas et al., present a system to express shRNAs in iPSCs for downstream biological studies with a focus on single-cell RNA-seq. Their platform addresses many molecular biology concerns inherent in current approaches, the manuscript includes custom analysis scripts, and includes a detailed protocol. The authors silenced SMAD2, CDH7, GATA4, KMT3D, NKX2-5, and controls (B2M, EGFP, SCR) during a cardiac differentiation protocol and analyzed the resulting gene expression differences and cellular populations using the split-seq and 10x platform. The following suggestions, however, would strengthen the manuscript.

We thank the reviewer for acknowledging the technical advances of the iPS2-seq platform and its potential for downstream biological studies. In our revised manuscript, we expanded the experimental validation of the method, including new datasets addressing the versatility (R2.3), inducibility (R2.10), and biological insights (R2.6) of iPS2-seq. We also clarified the effects on pluripotency (R2.1 & R2.8), validated target silencing (R2.9), and expanded screening metrics (R2.2, R2.5 & R2.7). These changes strengthen the biological relevance of the system while maintaining the methodological rigor and breadth that are the focus of this Technology article.

Major Concerns:

Point R2.1

Some targets will likely affect the pluripotent state of the iPSCs. Given the TGFb/SMAD pathway plays a vital role in pluripotency, does prolonged induction of SMAD2 shRNA prior to performing hPSC-CM differentiation lead to changes in the resulting cell type composition?

As noted in **lines 270–273** and shown in **Reviewer Fig. 5**, scRNA-seq analysis of iPSC2-seq hiPSCs at day 0 of cardiac differentiation revealed no significant clustering differences based on gene knockdown. This is consistent with the limited induction period used, 4 days of tetracycline treatment. Notably, *GATA4* and *NKX2-5* are not yet expressed at this stage (as confirmed in the reported data of Table S5 on D0 iPSC), further minimizing potential effects on cell state.

Reviewer Figure 5. Gene knockdown-associated cell clustering changes at day 0 of differentiation (cluster names as shown in Fig S2H; iPSC-P: proliferating iPSCs; iPSC-M: mesoderm-primed hiPSCs; iPSC-1/2: unbiased hiPSCs); no significant changes were found (adj. $P < 0.05$ by Fisher test vs. SCR with B-H correction).

We are well aware that prolonged *SMAD2* knockdown can affect the pluripotent state. Indeed, we have previously demonstrated that prolonged *SMAD2* knockdown does impair the pluripotent state by downregulating *NANOG*, upregulating neuroectodermal and mesodermal markers, and blocking endodermal induction (Bertero et al., 2018a). To avoid such confounding effects, we optimized the duration of pre-differentiation knockdown in the current study. The fact that *SMAD2* knockdown cells do not show delayed progression by day 6 of cardiac differentiation (**Fig. 2H**) provided preliminary support to the view that *SMAD2* functions later, during cardiac fate specification rather than pluripotency maintenance.

To directly probe the stage-specific role of *SMAD2*, we included a new time-course experiment (**Fig. 7, Fig. S7**, described in detail in the response to point R1.4 above) in which knockdown was induced or reversed at defined intervals. As shown in **Figure S7A–B**, *SMAD2* knockdown for up to 6 days did not alter the expression of core pluripotency markers (e.g., *NANOG*, *POU5F1*) nor the early stages of differentiation into primitive streak (**Fig. S7C**). These results confirm that the timing used in our iPSC2-seq screen preserves pluripotency and isolates the role of *SMAD2* during cardiogenesis.

Point R2.2

NKX2-5 and *GATA4* are cofactors. Can the authors focus a subfigure to understand the differences between populations which have both *NKX2-5* and *GATA4* silenced compared to a single factor? For example, are genes only regulated in the *NKX2-5* and *GATA4* silenced condition or does the presence of *GATA4* only amplify genes regulated by *NKX2-5*? Do the authors see differences in first versus secondary heart field markers in their *NKX2-5* shRNA cells?

In the present study, our Design was specifically optimized for single-copy integration to ensure unambiguous mapping of phenotypes to individual perturbations (**lines 81–87**). As a result, true double integrations were rare by design. We nevertheless searched for cells with both *NKX2-5* and *GATA4* shRNAs and identified only three such cells with high-confidence assignments (**Reviewer Table 2**), insufficient for any meaningful analysis.

That said, we agree that exploring combinatorial gene perturbations, such as co-silencing of *NKX2-5* and *GATA4*, represents a compelling future direction for iPSC2-seq. As noted in our response to point R1.9 and described in the

Limitations section (lines 628-631), such experiments could be enabled by adapting the current platform to co-deliver two barcoded shRNA libraries with distinct selection markers.

shRNA_combo	genotype	count_CTR	count_TET	tot
CHD7_GATA4	double	9	4	13
CHD7_SMAD2	double	1	1	2
GATA4_KMT2D	double	7	8	15
GATA4_NKX2.5	double	1	2	3
GATA4_SCR	double	3	2	5
GATA4_SMAD2	double	4	7	11
KMT2D_NKX2.5	double	1	2	3
KMT2D_SMAD2	double	2	2	4
NKX2.5_SMAD2	double	6	1	7
B2M_GATA4	double	0	1	1
B2M_NKX2.5	double	0	1	1
CHD7_KMT2D	double	0	3	3
CHD7_NKX2.5	double	0	1	1
CHD7_SCR	double	0	2	2

Reviewer Table 2. Number of cells with double integrations of various perturbations from the iPS2-10X-seq dataset on 3D cardioids. Only 1.3% of the cells were assigned to reliable double integrations (**Reviewer Table 1**).

Point R2.3

The usefulness of iPSC lines is the ability to study cell types from each germ layer. Differentiation can, however, result in the silencing of transgenes (the AAVS1 locus). Can the authors validate whether their transgenes are not silenced in terminally differentiated cell types from different germ layers (specifically cortical neuron differentiation)?

We appreciate the reviewer's point regarding transgene silencing during differentiation, which is a well-known challenge in hPSC-based systems, even when targeting safe harbor loci such as AAVS1. To address this, we previously performed a comprehensive validation of the sOPTiKD system (on which iPS2-seq is based) across 13 differentiated cell types derived from all three germ layers, including cortical neurons (Bertero et al., 2016). As shown in **Reviewer Fig. 6** (reproduced from Fig. 4A of that study), inducible knockdown of an EGFP reporter was robust and consistent across all lineages tested. These results demonstrate that both the CAG-OPTtetR transactivator and H1-TO-shRNA cassette remain transcriptionally active and tetracycline-responsive after differentiation when integrated at AAVS1.

Reviewer Figure 6. (Reproduced from Fig. 4A of Bertero et al., 2016). Validation of optimized inducible knockdown platforms following hPSC differentiation. EGFP expression measured by qPCR in the absence (CTR) or presence of tetracycline for 5 days (TET) in the indicated cell types derived from EGFP OPTiKD and sOPTiKD hESCs (sOPTiKD is the single cassette approach that enables iPS2-seq).

In our revised study, we further confirm stable transgene activity by quantifying UMI counts for OPTtetR transcripts (used for shRNA barcode assignment) across both undifferentiated hiPSCs and hiPSC-CMs. As shown in the

updated **Table S3**, barcode recovery efficiency was comparable between pluripotent and differentiated states, consistent with sustained transgene expression.

Accordingly, iPS2-seq relies on stable expression of OPTtetR in order to detect the shRNA-associated barcode inserted in its 3'UTR, and our analyses with cacheR demonstrated that a comparable proportion of hiPSCs and hiPSC-CMs could be reliably assigned to a given shRNA based on a similar number of UMIs for the OPTtetR (**Table S3**, which we updated with the second statistics).

To directly address the reviewer's request and expand validation to a different germ layer, we performed a new iPS2-10X-seq experiment following telencephalic brain organoid differentiation of our iPS2-seq pool (**Fig. 4, Fig. S4**). Although our original screen focused on cardiac genes, several targets (*SMAD2*, *CHD7*, *KMT2D*, and *B2M*) are also expressed in cortical neurons, allowing us to demonstrate, in principle, the functionality of the iPS2-seq system in neural lineages, including cortical neurons. These data further support the robustness of our inducible platform across germ layers.

Point R2.4

It is not clear how multiple shRNA specific barcodes would be required. Can the authors demonstrate the value of having unique shRNA-barcodes? Can the authors demonstrate how various shRNAs toward the same gene be used to understand transcript specific isoform differences?

We appreciate the opportunity to clarify the value of uniquely barcoded shRNAs. First, assigning each perturbation to a distinct barcode is essential for the accurate interpretation of pooled screening data. Even when multiple shRNAs target the same gene, they can differ in potency or off-target profiles, making it critical to track them individually. This design allows phenotypes to be cross-validated across distinct constructs, strengthening the confidence of gene-level conclusions (Echeverri and Perrimon, 2006).

Second, and equally important, the iPS2-seq platform enables clonal tracking through unique clonal identifiers (UCIs). This unique capability proved essential for several key findings in the study. Most importantly, clonal tracking led to the unexpected discovery of a neuroectoderm-primed subpopulation of genome-edited iPSCs ("iPSC-neuro" clones), which would have confounded downstream differentiation results if left undetected (see **R1.1B, Fig. 3** and **Fig. S3**). Moreover, as discussed in response to Reviewer 1 (**points R1.4 and R1.6**), clonal information allowed us to benchmark different control strategies (e.g., TET vs SCR) and demonstrated that failing to control for clonal identity can lead to false positives or negatives while screenings (**Fig. S5H-I**).

While our current screen used shRNAs that target all RefSeq isoforms, we note that isoform-specific RNA interference is possible (Fuchs et al., 2021; Kisielow et al., 2002; Murray et al., 2008). Future applications of iPS2-seq could exploit this capacity to dissect isoform-specific functions, particularly in conjunction with isoform-resolved scRNA-seq.

Point R2.5

It may be difficult to draw conclusions given the number of cells sequenced or the sequencing depth performed. How do the authors draw conclusions on a dataset containing 1,000-10,000 cells per experiment when looking at pooled populations of 23 shRNAs? Are the gene expression matrices deposited sparse? How many genes per cell are detected?

As now reported in **Table S4**, the number of cells recovered per perturbation in our iPS2-seq datasets, typically ranging from several dozen to a few hundred per shRNA, is consistent with published scRNA-seq-based screening studies (**Reviewer Table 3**). Likewise, the median number of UMIs and genes detected per cell across experiments falls within expected ranges and is now detailed in **Table S3**.

Year	PMID	Journal	Method	Experiment	Cells with one perturbation	Perturbations	Cells/perturbation	UMI/cell	Genes/cell	Mean UMI/shRNA
This study			iPS2-sci-seq	shRNA - Day 23 CMs	3569	20	178	2214	1018	56
			iPS2-10X-seq	shRNA - hiPSCs	4149	20	207	11009	4104	88
				shRNA - Day 23 CMs	3076	20	154			202
				shRNA - Day 0 iPSCs	8033	20	402			62
				shRNA - Day 2 MES	8137	20	407			53
				shRNA - Day 6 CPs	6517	20	326			35
				shRNA - Day 12 CMs	6352	20	318			69
				shRNA - CTR cardioid	2842	20	142			13642
			shRNA - TET cardioid	3018	20	151				
2024	37704762	Nature	CHOOSE	CRISPRn	38884	72	540	not reported	not reported	
2022	35953545	Nat. Neurosci.	CROP-seq	CRISPRi	28905	82	353	10583	3346	
2022	36303069	Nat. Neurosci.	CROP-seq	CRISPRi (vehicle)	16847	32	526	not reported	not reported	
				CRISPRi (IL1a+TNF+C1q)	19859	32	621			
2022	36381608	Cell Genomics	Petrurb-seq	CRISPRi	78393	480	163	not reported	not reported	
2021	34031600	Nat. Neurosci.	CROP-seq	CRISPRi	27189	374	73	not reported	not reported	
			CROP-seq	CRISPRa	16207	206	79			
2019	31422865	Neuron	CROP-seq	CRISPRi (iPSCs)	15000	58	259	not reported	5000	
			CROP-seq	CRISPRi (neurons)	8400	58	145		4600	

Reviewer Table 3. Number of cells/perturbation in various studies, as reported by the authors (Dräger et al., 2022; Leng et al., 2022; Li et al., 2023; Tian et al., 2019; Tian et al., 2021; Wu et al., 2022).

While our perturbation cell numbers are within standard ranges, we note that iPS2-seq is uniquely empowered by its single-copy, clonal-aware design. This allows us to control for variability in perturbation delivery and clonal heterogeneity: two major sources of noise in other screening platforms such as CROP-seq, which rely on randomly

integrated, often multi-copy guide constructs. The ability to track and filter clones improves interpretability and statistical power, which can partially offset lower cell numbers per perturbation.

Point R2.6

Given that the authors use well studied shRNA targets, there is no integration/validation with other studies which have investigated the role that these targets play in regulating cardiomyocyte gene networks. Do the authors find similar perturbances with other previously published studies (NKX2-5 and GATA4)?

We thank the reviewer for bringing this point to our attention. To our knowledge, there are no previously published scRNA-seq studies of *NKX2-5* or *GATA4* loss-of-function in hPSC-derived cardiomyocytes. Therefore, for *NKX2-5*, we reanalyzed data from Anderson et al., 2018, which reported bulk RNA-seq profiles of *NKX2-5* knockout hESCs during cardiac differentiation. Their study identified *HEY2* as a key downstream target of *NKX2-5*, with strong downregulation in knockout cells at day 10 of differentiation.

Using our iPS2-seq time course data, we examined *HEY2* expression at days 6, 12, and 23 of cardiac differentiation. Despite using shRNA-based knockdown (rather than complete KO), we observed a significant reduction in *HEY2* specifically at day 12, close to the time point reported in Anderson et al., thus recapitulating this key regulatory relationship (Reviewer Fig. 7).

Reviewer Figure 7. Expression of *HEY2* during hiPSC-CM differentiation in SCR control (CTR) and *NKX2-5* knockdown (KD) cells from the iPS2-10X-seq time course (Fig. 2). Time points correspond to key stages of cardiac development: day 6 (cardiac progenitors), day 12 (early CMs), and day 23 (late CMs).

Since we could not locate similar datasets exploring *GATA4* loss of function during hPSC-CM differentiation, we generated new scRNA-seq datasets (Reviewer Fig. 8). In our pooled iPS2-10X-seq dataset at day 7.5, *GATA4* shRNAs resulted in only subtle differences, primarily a modest increase in cardiac fibroblasts (Fig. 5E). We hypothesized that non-cell autonomous compensation from neighboring cells may have masked more pronounced effects. To test this, we analyzed arrayed *GATA4* knockdown left ventricle organoids at days 4.5 and 7.5, in which all cells were uniformly perturbed. This revealed a much stronger phenotype: at day 4.5, *GATA4* knockdown led to an increase in immature cells, including immature cardiomyocytes, and a substantial reduction in differentiated cardiomyocytes (Reviewer Fig. 8A–B). By day 7.5, knockdown organoids exhibited an increased proportion of fibroblasts and immature cells, along with a marked depletion of mature cardiomyocytes (Reviewer Fig. 8D–E). Normalized expression of *GATA4* was reduced in knockdown cells at both time points, confirming efficient silencing (Reviewer Fig. 8C, 8F). These findings support a key role for *GATA4* in cardiomyocyte maturation and illustrate how arrayed follow-up can uncover phenotypes that are partially masked in pooled designs.

We view these experiments as important support for the robustness of iPS2-seq and are grateful for the reviewer's prompt to include them. However, given the scope of the current study and the additional biological validation already dedicated to *SMAD2* (Figs. 6–7), we opted not to include the *GATA4* results in the main text to maintain narrative focus and stay within the space constraints. These findings will be developed further in a dedicated follow-up manuscript, which we intend to preprint in the near future.

Reviewer Figure 8. scRNA-seq analysis of left ventricular cardiac organoids with inducible *GATA4* knockdown, performed using Chromium Next GEM Single Cell 3' Reagent Kits v3.1 with sample multiplexing via cell multiplexing oligos. (A) UMAP and clustering of control and tetracycline-treated organoids at day 4.5. (B) Proportional distribution of cells across clusters at day 4.5. (C) Expression of *GATA4* in control vs. tet-treated cells at day 4.5. (D) UMAP and clustering at day 7.5. (E) Proportional distribution of cells across clusters at day 7.5. (F) Expression of *GATA4* in control vs. tet-treated cells at day 7.5.

Point R2.7

Have the authors compared the clones with biallelic integration of two different shRNA into AAVS1 loci or the same shRNA target in both alleles compared to integration of a shRNA into a single allele?

As discussed in our response to point R2.2, the iPS2-seq platform was specifically designed to promote single-copy integration events, minimizing biallelic or combinatorial shRNA targeting. Accordingly, the number of clones with dual integration—whether involving distinct shRNAs or two copies targeting the same gene—was extremely low and insufficient to support meaningful analysis (**Reviewer Tables 3 and 4**).

shRNA_combo	genotype	count_CTR	count_TET	tot
GATA4_GATA4	homozygous	5	3	8
NKX2.5_NKX2.5	homozygous	0	1	1
SMAD2_SMAD2	homozygous	0	1	1

Reviewer Table 4: Double-integrated cells with shRNA targeting the same gene in the iPS2-10X-seq experiment performed on left ventricular cardiac organoids at day 7.5, shown in Fig.5.

However, to directly address the reviewer's comment, we experimentally compared heterozygous and homozygous *SMAD2* knockdown clones in our validation experiments (**Fig. S6I**). As expected, both exhibited reduced left ventricular cardioid size, but the phenotype was notably stronger in the homozygous clone (**Fig. S6J**). Similarly, scRNA-seq analyses revealed a more pronounced expansion of epicardial-like cells in the homozygous background compared to pooled iPS2-seq cells enriched for heterozygous targeting (compare **Fig. 6E–G** with **Fig. S6G–H**).

Point R2.8

The neuroectoderm population may be an inherent problem with this platform. Does transfection affect the state of the iPSCs and would culturing iPSCs for a few passages after transfection reduce the presence of this population?

As detailed in our response to point R1.1, we extensively characterized the “iPSC-neuro” subpopulation and provided multiple lines of evidence suggesting that this state reflects stable epigenetic priming rather than transient transfection-induced effects (**Fig. 3** and **S3**).

To directly address the reviewer’s concern, we passaged iPSC2-seq hiPSCs through passages 3, 8, and 13, and performed RT-qPCR for neuroectoderm markers (*ZIC1*, *SIX3*, *NEUROG3*) and pluripotency markers (*POU5F1*, *NANOG*; **Fig. 3B**, **Fig. S3A**). In parallel, we examined clonal dynamics *via* gDNA barcode sequencing (**Fig. 3C**, **Fig. S3B**). Both of these analyses showed that extended passaging progressively depleted iPSC-neuro clones, but also reduced overall clonal diversity, highlighting a trade-off between minimizing biased clones and preserving population heterogeneity. As discussed in **lines 568–570**, we identified *UNC5D* as a surface marker selectively upregulated in iPSC-neuro clones, enabling potential removal by FACS or magnetic sorting, a more efficient and unbiased strategy than prolonged passaging.

Point R2.9

Given that shRNA within each cell may result in variable target knockdown levels and may result in differences in gene expression networks (those that are haploinsufficient or must reach a specific level of silencing), it is necessary to quantify the target protein levels within each shRNA containing cell. Can the authors combine their platform with single-cell protein level quantification (combining iPSC2-10x-seq with CITE-seq)? Can the authors demonstrate shRNA efficacy by measuring shRNA barcode with the expression of each target/transcription factor?

This was a very interesting suggestion. While in theory, combining iPSC2-10X-seq with CITE-seq for intracellular or nuclear targets would allow direct measurement of protein-level knockdown per cell, in practice, this remains technically very challenging. To our knowledge, such an approach has only been achieved in specialized contexts with extensive protocol customization, including antibody conjugation and nuclear epitope optimization (e.g., (Chung et al., 2021), who targeted a single nuclear protein). Moreover, CITE-seq does not scale easily, as each target requires custom-conjugated antibodies, which are both costly and limited in availability. Accordingly, we are unaware of any published scRNA-seq screen systematically reporting protein-level readouts, particularly in hPSC models (see **Reviewer Table 3**).

That said, we share the reviewer’s interest in assessing shRNA efficacy at the protein level, and we thus implemented a technically feasible proof-of-principle in our revised manuscript. Specifically, we performed CITE-seq for *B2M*, a surface antigen for which reliable oligo-conjugated antibodies are commercially available, in left ventricular cardiac organoids expressing either *B2M* or *SMAD2* shRNAs (**Fig. S6A–F**). This experiment demonstrated robust depletion of both *B2M* transcript (**Fig. S6E**) and surface protein (**Fig. S6F**) specifically in *B2M*-targeted, tetracycline-treated cells, validating the effectiveness of shRNA knockdown.

To our knowledge, this approach, which we termed iPSC2-CITE-seq, represents the first example of combining pooled shRNA perturbation in hPSCs with protein-level readout. While not suited for comprehensive profiling of all shRNA targets due to scalability constraints, it enables precise measurement of select proteins of interest in scenarios where suitable antibodies are available. This includes, for instance, phenotypic screening based on surface markers.

Point R2.10

The authors state the value of their platform is inducible. However, there are no experiments that have validated the ability to temporally induce cells. Given that some transcription factors act temporally during differentiation, can the authors demonstrate differences in gene networks from specific stages of differentiation.

We appreciate the reviewer’s emphasis on the importance of temporal control in dissecting transcription factor function. The inducibility of the sOPTiKD system, on which iPSC2-seq is based, was previously validated through temporal knockdown of the chromatin regulator *DPY30* at successive stages of hPSC differentiation into mature

cell types from the three germ layers (Bertero et al., 2016; **Reviewer Fig. 9**). These experiments demonstrated that the platform allows to identify phenotypes arising depending on the timing of gene silencing.

Reviewer Figure 9. (Reproduced from Fig. 6 of Bertero et al., 2016). (A) The experimental design was used to investigate the role of *DPY30* at various stages of hPSC differentiation. (B) qPCR-based analyses of *DPY30* (blue) and *B2M* (orange) sOPTiKD hPSCs after differentiation into mature lineages. CTR, no knockdown; KD ind/spec/mat refer to knockdown from induction/specification/maturation, respectively. Results are from three independent cultures per condition. * $P < 0.05$ versus *B2M* in the same condition (two-way ANOVA with post-hoc Sidak comparisons). Only the data for cardiac differentiation is reported for the sake of brevity.

In our revision, we applied the same logic to validate the role of *SMAD2* during cardiogenesis. As described in our responses to points R1.4 and R2.3, we leveraged the inducibility and reversibility of iPS2-seq to compare early and late knockdown, as well as transient *versus* sustained silencing (**Fig. 7** and **Fig. S7**). These experiments revealed that *SMAD2* is specifically required for cardiac progenitor specification and patterning, promoting CM fate and suppressing alternative lineages. This provides a clear example of a temporally restricted transcription factor function dissected using our platform.

Minor Concerns:

Point R2.11

In the statement, "6,874 transcriptomes passed additional filters for low-quality nuclei and potential doublets". How has this data been filtered?

As described on **lines 1100–1102** of the Methods, we filtered low-quality nuclei by discarding cells with fewer than 500 total UMIs, which likely reflect ambient RNA or damaged nuclei. To improve clarity, we also added the following detail to the Methods section in our revision: "*Potential doublets were removed by excluding nuclei with a total UMI count greater than two standard deviations above the mean UMI count across all cells.*" This conservative threshold helps eliminate multiplets while preserving high-quality single-cell data.

Point R2.12

How do the authors define "immature cardiomyocytes"?

We used the term "immature cardiomyocytes" to describe a cluster of CMs identified in our iPS2-sci-seq dataset that, as noted in **lines 219–224**, exhibited a high *MYH6/MYH7* expression ratio. This is a well-established molecular hallmark of immature or developing cardiomyocytes (Karbassi et al., 2020).

Point R2.13

Many instances of non-scientific language:

- FIAU weeds out clones
- which we took to heart in our
- Exemplar genotyping of hiPSC clones

- barcode swaps can be reassigned thanks to UCIs.

We thank the reviewer for pointing out these instances of informal language. All the highlighted phrases were revised for scientific clarity and consistency as part of our broader language refinement, detailed in our response to comment R1.5. This included systematic editing of tone, removal of colloquialisms, and improved phrasing throughout the manuscript.

Point R2.14

It is not clear whether different shRNAs for the same target were enriched within each cluster?

Yes, in several cases we observed consistent enrichment or depletion patterns across different shRNAs targeting the same gene. For example, both SMAD2.2 and SMAD2.4 (corresponding to shRNA 2 and shRNA 4 against SMAD2) showed similar behavior, as shown in **Figs. 2H** and **5E**. We clarified these correspondences in the first relevant figure legend to improve interpretability (**Fig. 2E**).

Reviewer 3

This manuscript describes a new technology, iPS2-seq, which involves pooled integration of an inducible tet-ON system for a library of shRNA into the human AAVS1 safe-harbor locus in hiPSCs. Each shRNA is identified by specific barcode (BC) sequences, which can be read out along with cellular mRNA using various existing single-cell technologies. Additionally, the authors incorporated Universal Clonal ID (UCI) sequences with the BCs to tag each integration event, enabling the identification of individual clones through the same readout technique.

Overall, I think the work is careful and solid. The potential of using the AAVS1 locus to express high-efficiency shRNAs in a controlled manner, combined with the ability to identify each clone of a single genomic integration event, is useful. The writing is very clear — authors have done an excellent job explaining the technical details and considerations of their system. The proposed system carefully considers the many technical challenges associated with conducting genetic screens in hiPSCs.

We thank the reviewer for their positive assessment and thoughtful suggestions. In our revised manuscript, we extended the experimental validation of iPS2-seq by dissecting SMAD2 function (R3.2), demonstrating new applications such as iPS2-CITE-seq and iPS2-multi-seq (R3.1, R3.4), and expanding our bioinformatics toolkit for accessibility (R3.1). We also clarified clonal tracking capabilities (R3.2, R3.5), performance benchmarks (R3.4), and technical details relevant to reproducibility. These additions broaden the utility of iPS2-seq while reinforcing its rigor and scalability.

Major Comments:

Point R3.1.A (We are breaking up this important remark to separately tackle each subpoint)

1. I believe that this paper was submitted as a "Resource", and so aims to describe a new method/approach. I think the paper accomplishes this well. However, I think the impact of the new method is limited, in that it seems unlikely that other groups will be able to easily adopt the system.

We confirm that this manuscript was submitted as a "Technology" article. To facilitate adoption by other laboratories, we have provided an extensive, step-by-step protocol. We also recognized that the main bottleneck for many groups lies not in the molecular biology but in the bioinformatics, where challenges such as incomplete documentation, software dependencies, and limited expertise often hinder reproducibility. To address this, we developed the *catcheR* package and distributed it through a Docker environment for streamlined installation and analysis. In our revision, we further expanded *catcheR* with user-friendly functions for loading data into Monocle3 and performing key downstream analyses, including statistical tests of shRNA-associated clustering (e.g., **Fig. 2H**) and clone-specific pseudotime trajectories (e.g., **Fig. 2J–K**), thereby lowering the entry barrier for labs with limited computational support.

Since presenting our prototype at the ISSCR Annual Meeting, we have received over a dozen direct requests for the iPS2-seq plasmid (to be released on Addgene upon publication) and established collaborations with several stem cell groups that are applying the method in diverse contexts, including blastoids (Vincent Pasque) and hepatocyte organoids (Ludovic Vallier and Tamir Rashid). These examples underscore the accessibility and relevance of iPS2-seq across stem cell model systems.

Point R3.1.B

First, there are several limiting technical details such as (i) the need to introduce shRNAs by HDR (imposing scale limitations—I do not think that the authors' explanation that scRNA-seq studies are limited to 100s of genes is a general constraint);

In practice, most published scRNA-seq-based screens in hPSC models have targeted only a few dozen genes, primarily due to the high cost of single-cell sequencing (see **Reviewer Table 3** above). Notably, the number of perturbations often overestimates the number of genes tested, as CRISPR screens typically use 2–4 sgRNAs per gene. Only a handful of studies have tested more than 100 gene targets, with the largest to date involving 240 genes (Wu et al., 2022). Given that scRNA-seq costs have remained relatively constant in recent years, we believe our use of HDR to ensure isogenic shRNA integration does not represent a major limitation for realistic screen sizes.

That said, scaling iPS2-seq further using recombination-mediated gene editing would be straightforward, as noted in **lines 634–639**. In support of this, iPSC lines with landing pads in both the *AAVS1* and *CLYBL* genomic safe harbors have been reported and are publicly available (Blanch-Asensio et al., 2023). We have updated the Limitations section to reflect these options.

Point R3.1.C

(ii) use of shRNAs, given the known limitations in off-target effects and broad adoption of other tools such as CRISPRi for genetic screens in hiPSCs;

As discussed in **lines 593–609**, we view iPS2-seq as a complementary tool to existing perturbation platforms, offering unique advantages for scRNA-seq-based screens in hiPSC models, most notably clone-aware tracking.

Regarding the reviewer's concern, shRNA off-target effects are well-characterized and can be mitigated by using multiple validated shRNAs per target and leveraging design principles refined over two decades. In fact, despite the off-target risks, the ability to benchmark multiple shRNAs at the same time, supports rigorous interpretation.

While CRISPRi has seen broad adoption in some contexts, nearly all published screens in hPSCs have been limited to brain-related lineages. This may reflect differences in lineage-specific transgene silencing. Indeed, we recently reported that a CRISPRa system effective in hPSCs completely loses its functionality after cardiac differentiation (Karbassi et al., 2024). Similarly, we found that a published CRISPRi system based on dCas9-BFP-KRAB targeted to the *CLYBL* locus (Tian et al., 2019) is prone to early silencing in hiPSCs. Using the WTC-11 line from the Allen Institute (AICS-0090-391), we confirmed heterogeneous BFP expression already in undifferentiated cells (in line with the cell line characterization data in the certificate of analysis - https://www.coriell.org/0/PDF/Allen/ipsc/AICS-0090-391_CofA.pdf). We then inserted TET-inducible sgRNAs at the *AAVS1* locus using an all-in-one, tetracycline-inducible cassette similar to the one behind iPS2-seq, and noticed that one-third of the genome-edited clones we had isolated had silenced BFP (**Reviewer Fig. 10A**). This correlated with incomplete knockdown of *TFRC* upon induction of the sgRNA (**Reviewer Fig. 10B**).

These challenges motivated our investment in developing iPS2-seq, which builds on the sOPTiKD platform: a well-characterized and robust shRNA system we and others have validated in over a dozen differentiated lineages from all three germ layers (Bertero et al., 2016; Gogolou et al., 2022; Macrae et al., 2023; Wesley et al., 2022; Williams et al., 2023).

Reviewer Figure 10. (A) Flow cytometry analyses of BFP expression in clonally isolated WTC-11 hiPSCs expressing dCas9-BFP-KRAB in the *CLYBL* locus, and an inducible sgRNA expression cassette in the *AAVS1* locus. **(B)** RT-qPCR analyses of *TFRC* expression in NTC control and TFRC sgRNA-expressing clones, in the absence (CTR) or presence (TET) of tetracycline for 4 days.

Point R3.1.

and (iii) installation of this system in a particular cell line.

As a matter of fact, we specifically designed iPS2-seq to avoid the lengthy setup required by many other systems. For example, the CRISPRi platform described in the previous comment requires prior engineering of the *CLYBL* locus with dCas9-BFP-KRAB and subsequent lentiviral transduction of sgRNAs, typically produced in dedicated facilities. In contrast, iPS2-seq only requires standard molecular cloning and a single round of *AAVS1*-targeted genome editing, which can be completed by an experienced operator in approximately one month using widely available lab equipment (see lines 544–547).

Point R3.2

2. I would have liked to see a more extensive biological demonstration of the method. The demonstration experiments are largely small-scale, underpowered experiments that do not demonstrate the full capabilities of this technology. For instance, while the authors highlight clonal awareness as the strongest asset of iPS2-seq, there is very limited evidence supporting this claim. Larger datasets would likely be required to demonstrate this capability.

Our observation that a subset of hiPSC clones is primed for neuroectoderm differentiation exemplifies the value of the clonal awareness enabled by iPS2-seq. As discussed in responses to points R1.1 and R2.8, we strengthened this conclusion in our revision by differentiating our existing pool of iPS2-seq cells into telencephalic brain organoids (**Fig. 4, Fig. S4**), exploring the genetic and epigenetic basis of this bias via combined single-cell ATAC-seq and RNA-seq (**Fig. 3C-D**), and assessing clonal drift over extended passaging (**Fig. 3A-B, Fig. S3**). As further discussed in responses to R1.1D and R1.2, we also tested whether comparing TET-treated cells to their untreated counterparts within the same clone improves detection of significant transcriptional changes compared to comparisons with unrelated controls. This clone-aware design produced more consistent and robust results (**Fig. S5H-I**).

While we did not pursue a large-scale screen, which is beyond the scope of this Technology article, we expanded the range of iPS2-seq use cases. As discussed in responses to R1.4 and R2.9, we demonstrated its compatibility with CITE-seq & arrayed screening formats (**Fig. S6A-H**), as well as joint RNA/chromatin profiling (**Fig. 3** and **Fig. S3**). These additions significantly extend the methodological and biological utility of the platform.

Lastly, we applied iPS2-seq to validate the role of *SMAD2* in cardiac differentiation through an, arrayed screen across multiple clones expressing the same shRNA, revealing consistent phenotypes (**Figs. S6A-H**). This further underscores how the platform can support robust, clone-aware biological discovery, even outside a large screening context.

Point R3.3

3. Overall comment on style and length: The writing is clear, but it includes too much methodological detail. For example, Fig 1 and first section of Results, about cloning a plasmid pool, should be moved to supplement.

As part of the broader language revision described in our response to comment R1.5, we substantially restructured the manuscript by moving non-essential methodological details from the Results to the Methods or Supplementary Protocols. This has improved narrative flow, and we thank the reviewer for this helpful suggestion.

Point R3.4A (breaking up this comment to respond to individual subpoints)

4. A few details about the results were missing for many of the described experiments:

* What library complexity did you achieve?

The duplication rate for iPS2-sci-seq was high (90.3%), as expected for this method (Cao et al., 2017), while iPS2-10X-seq showed much lower duplication rates, ranging from 14.2% in day 0 hiPSCs to 42.3% in hiPSC-CMs. We have updated **Table S3** to include these values along with other relevant library statistics.

Point R3.4B

* How many genes per cell were detected, and how does this compare to the unmodified sci-RNA-seq/10X protocol?

Table S3 now also reports the mean number of UMIs and genes detected per cell across all experiments; these figures are also reported in **Reviewer Table 3** above for the reviewer's convenience.

These data demonstrate that the iPS2-10X-seq workflow does not interfere with standard gene expression library preparation, as expected, since shRNA barcodes are amplified from a small fraction of the pre-amplified cDNA, material that is typically unused since library complexity is saturated using only ~25% of the cDNA, per manufacturer recommendations. Consistently, our iPS2-10X-seq datasets yielded >4000 genes per cell, in line with values reported in published scRNA-seq screens in hPSC models.

In contrast, for iPS2-sci-seq, barcode amplification occurs *via* a multiplexed PCR step that could, in principle, interfere with conventional gene expression capture. However, we optimized this step to preserve transcriptome integrity (**Fig. ED2**). In the benchmark experiment shown in **Fig. 1D-K** and **Fig. S1E-L**, the standard sci-RNA-seq protocol yielded 4,674 UMIs and 1,052 genes per cell, while our optimized barcode amplification condition (1:2 P7 capture:P7 primer ratio) even improved these figures to 6,189 UMIs and 1,285 genes per cell.

Point R3.4C

* What is the capture/detection efficiency of the shRNA? How many UMIs per cell per shRNA were detected?

The mean number of UMIs per valid UCI-BC (i.e., per shRNA assignment) was approximately 15 for iPS2-sci-seq, and ranged from 54 to 156 for iPS2-10X-seq, depending on the experiment. These values are now reported in **Table S3**, along with the corresponding gene and UMI counts per cell. This level of barcode capture is sufficient for confident assignment of perturbations in both platforms (see also response to point R1.7).

Point R3.4D

* How many cells per shRNA and per target gene did you achieve?

As an example, for iPS2-10X-seq at day 0, the number of cells per shRNA ranged from 41 to 1,450, while the number of cells per gene target (i.e., combining multiple shRNAs) ranged from 477 to 2,919. These values are now reported in detail for each experiment in a new supplemental resource, **Table S4**.

Point R3.4E

* What is the average knockdown of targeted genes?

We discussed this important aspect extensively in our response to point R1.3. To summarize, while transcript-level knockdown estimates from scRNA-seq can be informative, they are limited by technical constraints such as transcript dropout, shallow coverage, and the fact that shRNAs act post-transcriptionally. As shown in **Reviewer Fig. 3**, target knockdown was most detectable for highly expressed genes like *B2M*, and less so for low-abundance targets such as transcription factors. These results are also summarized in **Table S5**.

On average, these measurements suggest ~65% RNA-level silencing across targets, which aligns with expected knockdown efficiencies for validated shRNAs. However, this likely underestimates functional effects, as shRNAs can also inhibit translation without inducing RNA decay. In support of this, our CITE-seq experiment for *B2M* confirmed robust protein-level depletion (**Fig. S6D–F**), and in the case of *SMAD2*, we observed >75% knockdown at both RNA and protein levels following TET induction (**Fig. 7B–C**).

Point R3.4F

Explaining these technical parameters is essential for interpreting the downstream transcriptomic analyses (e.g. Figures 2G and 2H). It would be helpful to include these technical details in supplementary materials or methods.

We fully agree and regret that these technical details were not more prominently presented in the original submission. In addition to the aforementioned Supplementary Tables, all analyzed datasets have been deposited to Zenodo, enabling prospective users to independently assess data quality and explore the resource in depth.

Minor comments and suggestions (detailed responses not necessary):

Overall:

Point R3.5

* One of the advantages of this system is that the OPTtetR mRNA, which carries the UCI-BC, is constitutively expressed. Therefore, the UCI can be read in cells not exposed to tetracycline (tet) (no shRNA expression) to determine the effects of clonal variability alone on cellular transcriptomes. Have you conducted comparisons under basal conditions to evaluate the variation between different clones carrying the same shRNA, or even different shRNAs, given that the shRNAs are not expressed?

This is indeed one of the most powerful advantages of our system. As detailed in our response to point R1.1, we greatly expanded our analyses of untreated, undifferentiated hiPSC clones using iPS2-multi-seq, providing strong evidence that a subset exhibits an epigenetic bias toward neuroectoderm differentiation (**Fig. 3, Fig. S3**).

Additionally, both of our organoid-based experiments included no-tet controls, which were essential to disentangle clonal variability from perturbation-driven effects (**Fig. 4 and Fig. 5**).

To directly address the reviewer's suggestion, we also performed an arrayed iPS2-CITE-seq experiment in cardiac organoids derived from clonal pools expressing inducible *SMAD2* or *B2M* shRNAs. By comparing each tet-treated clone with its untreated counterpart, we demonstrated statistically significant and reproducible shifts in cell type composition (**Fig. S6G–H**). This highlights another unique application of iPS2-seq: leveraging internal clonal controls within a single experiment to improve reproducibility while minimizing cost and complexity.

Point R3.6

* One common missing piece in all the experiments was the knockdown efficiency of each shRNA (since this is likely to be noisy for any individual shRNA/target gene, you could report the average across all shRNAs in the experiment)

Please refer to the response to point R3.4 above, which seems to be requesting the same parameter.

Point R3.7

* Recommend providing a note estimating the number of cells needed to achieve sufficient coverage per clone/shRNA/gene by considering the dropout rates at various steps of the protocol (e.g. transfection efficiency, integration efficiency, single-cell protocol efficiency).

This is an excellent suggestion. We have now included a dedicated note in Supplemental Protocol 1 (**lines 2078–2093**) along with a new illustrative figure (**Fig. SP1.1**) that provides an example calculation for estimating the number of cells required per shRNA, accounting for dropout rates at each step of the workflow.

Point R3.8

* How quickly does gene expression decrease after tetracycline treatment?

We have previously shown that inducible shRNAs expressed through the sOPTiKD system, on which iPS2-seq is based, can achieve maximal mRNA depletion within 2–3 days of tetracycline treatment (Bertero et al., 2016). Protein-level knockdown, however, depends both on the stability of the target protein and the rate of cell proliferation, which influences the rate of diluting existing proteins. In our revised manuscript, we validated these kinetics using *SMAD2* as an example: mRNA silencing peaked within 2 days, and protein depletion reached ~50% by day 4 and ~75% by day 6 (**Fig. 7A–C**; see also point R1.4).

Point R3.9

* Is there any evidence of reversibility? How long after stopping tet treatment does the gene expression return to normal level?

This is an additional aspect we had previously validated in the context of the sOPTiKD system (Bertero et al., 2016). In our revision, we confirmed these findings for *SMAD2*, demonstrating that both RNA and protein levels return to baseline within ~4 days following tetracycline withdrawal (see **Fig. 7A–C** and response to point R1.4).

Molecular cloning and genome editing in iPS2-seq (Figure 1):

Point R3.10.A

* Clone genotyping check (Figure 1G): Why do most of the 32 clones have AAVS1 positive bands?

As explained in **Tables SP1.23–SP1.24**, the *AAVS1* wild-type band is expected to disappear only when both alleles are successfully edited. This occurs in compound heterozygous clones (5 clones, positive for both the 5' junction PuroR and 3' junction OPTtetR-shRNA, and the 5' junction NeoR from the filler plasmid) or in homozygous clones that integrated two shRNAs (4 clones, positive only for the 5' junction PuroR and 3' junction OPTtetR). All other clones retain one wild-type allele and therefore show an *AAVS1* wild-type band on genotyping. Note that the PCR strategy for the locus cannot amplify the inducible shRNA cassette due to the large size and extremely high GC-content of the CAG promoter.

As per the reviewer's suggestion below, we have added a schematic of our genotyping strategies in **Figure SP1.7** to hopefully mitigate any misunderstandings on the complex strategies used here.

Point R3.10.B

Why do some of the alleged single shRNA containing clones not have band for 5' junction (NeoR)?

Integration of the filler plasmid (detected by the 5' junction NeoR PCR) is not a requirement for identifying single shRNA integrants. Clones that are positive for the AAVS1 wild-type band, the 5' junction PuroR, and the 3' junction OPTtetR have integrated a single copy of the shRNA cassette in one allele, while the other allele remains unedited or contains only a minimal genomic scar. From a technical standpoint (**lines 1769-1801**):

We note that neo selection is tricky because of the narrow concentration range that enables selection of unedited cells while minimally interfering with the fitness of edited ones. We thus optimized a coselection protocol that uses a low dose of neo, and, accordingly, does not fully select biallelic editing. While this was not needed for our experiments, a more stringent neo selection can be employed if this is desired. Alternatively, other selection markers could be employed (such as blasticidin resistance and/or a fluorescent protein to enable sorting of targeted clones).

Point R3.10.C

For clone genotyping, it would be helpful to provide a schematic of primer binding sites and direction (Table SP1.21).

We appreciated this helpful suggestion and added a schematic illustrating the primer binding sites and orientations, now included as **Figure SP1.7** in Supplemental Protocol 1.

Point R3.10.D

In lines 255-256, it states, "...up to ~81% of the 27 FIAU-insensitive clones integrated a single functional shRNA." This calculation appears to assume that out of the 27 FIAU-insensitive clones, 5 had random shRNA integration, resulting in $22/27 = 81\%$ of clones with correct single shRNA integration. However, based on the gel image in Figure 1G, I see 6 clones with genotype-level evidence of random integration that survived FIAU. Shouldn't the correct calculation be $21/27 = \sim 77\%$?

This calculation is based on the estimated number of functional shRNA integrations, which we report above each lane in **Figure S1D**. Specifically, 22 out of 27 FIAU-insensitive clones showed genotype-level evidence of expressing a single shRNA, either via compound heterozygous targeting (5 clones, see point R3.10A) or heterozygous integration in one allele (17 clones, see point R3.10B). The remaining 5 clones either had two shRNAs (4 clones) or were not correctly edited (1 clone).

The reviewer correctly notes that 6 clones showed evidence of additional integration yet survived FIAU. In 5 of these cases, we conclude that integration occurred in transcriptionally inactive regions and likely did not yield functional shRNA expression. Therefore, we did not count these as additional functional integrations.

Importantly, this estimate (~81%) aligns well with the empirical frequency of cells expressing a single shRNA based on barcode detection in our iPS2-sci-seq (~80%) and iPS2-10X-seq (~75%) datasets.

Point R3.11

Applying iPS2-seq to split-seq (Figure 2):

* Could the authors comment on why HAND1 and CACNA1C dysregulation due to NKX2.5 loss of function (LoF) are notable findings? Are there any references to support this?

NKX2-5 is a well-established regulator of *HAND1* in both mouse and human cardiomyocytes (Anderson et al., 2018; Hofbauer et al., 2021; Tanaka et al., 1999), supporting the biological relevance of our iPS2-sci-seq findings. Although *CACNA1C* is not a known direct target of NKX2-5, it encodes the $\alpha 1C$ subunit of the L-type voltage-gated calcium channel CaV1.2, which plays an essential role in cardiomyocyte function (Bozarth et al., 2018). The downregulation of *CACNA1C* we observed in NKX2-5 knockdown cells may therefore reveal a previously

underappreciated regulatory effect, with potential relevance for the cardiac phenotypes associated with NKX2-5 mutations. We have added these references and the relevant discussion in **lines 235-237**.

Point R3.12

* Are there other expected changes in gene expression and pathways due to the LoF of these genes? Do you see these changes in your data? Including an analysis that showcases individual genes and pathways expected to be altered due to the perturbation of these known CHD genes would add credibility to the experiment.

As discussed in the response to point R2.6 and shown in **Reviewer Fig. 7**, we validated the downregulation of *HEY2* in *NKX2-5* knockdown cells using our iPS2-10X-seq dataset at day 12 of hiPSC-CM differentiation. This is consistent with previous reports identifying *HEY2* as a transcriptional target of *NKX2-5* during cardiogenesis (Anderson et al., 2018), and supports the utility of iPS2-seq for uncovering gene regulatory effects relevant to congenital heart disease.

Point R3.13

* In Figure 3H, the dot colors for Day 12 and Day 23 are similar and difficult to distinguish

We apologize for this oversight. While we made an effort to ensure figure accessibility, including for color-blind readers (as the senior author is one), this issue escaped our attention. We have now updated the dot colors in **Figure 2G-H** to improve visual distinction between early, mid, and late CMs in day 12 and day 23.

References:

- Anderson, D. J., Kaplan, D. I., Bell, K. M., Koutsis, K., Haynes, J. M., Mills, R. J., Phelan, D. G., Qian, E. L., Leitoguinho, A. R., Arasaratnam, D., et al. (2018). NKX2-5 regulates human cardiomyogenesis via a HEY2 dependent transcriptional network. *Nat. Commun.* **9**, 1373.
- Ang, Y. S., Rivas, R. N., Ribeiro, A. J. S., Srivas, R., Rivera, J., Stone, N. R., Pratt, K., Mohamed, T. M. A., Fu, J. D., Spencer, C. I., et al. (2016). Disease Model of GATA4 Mutation Reveals Transcription Factor Cooperativity in Human Cardiogenesis. *Cell* **167**, 1734-1749.e22.
- Balmas, E., Sozza, F., Bottini, S., Ratto, M. L., Savorè, G., Becca, S., Snijders, K. E. and Bertero, A. (2023). Manipulating and studying gene function in human pluripotent stem cell models. *FEBS Lett.* 1873-3468.14709.
- Bertero, A., Pawlowski, M., Ortmann, D., Snijders, K., Yiangou, L., Cardoso de Brito, M., Brown, S., Bernard, W. G., Cooper, J. D., Giacomelli, E., et al. (2016). Optimized inducible shRNA and CRISPR/Cas9 platforms for in vitro studies of human development using hPSCs. *Development* **143**, 4405–18.
- Bertero, A., Brown, S., Madrigal, P., Osnato, A., Ortmann, D., Yiangou, L., Kadiwala, J., Hubner, N. C. N. C., De Los Mozos, I. R., Sadée, C., et al. (2018a). The SMAD2/3 interactome reveals that TGFβ controls m6A mRNA methylation in pluripotency. *Nature* **555**, 256–259.
- Bertero, A., Yiangou, L., Brown, S., Ortmann, D., Pawlowski, M. and Vallier, L. (2018b). Conditional Manipulation of Gene Function in Human Cells with Optimized Inducible shRNA. *Curr. Protoc. Stem Cell Biol.* **44**.
- Blanch-Asensio, A., Van Der Vaart, B., Vinagre, M., Groen, E., Arendzen, C., Freund, C., Geijsen, N., Mummery, C. L. and Davis, R. P. (2023). Generation of AAVS1 and CLYBL STRAIGHT-IN v2 acceptor human iPSC lines for integrating DNA payloads. *Stem Cell Res.* **66**, 102991.
- Bozarth, X., Dines, J. N., Cong, Q., Mirzaa, G. M., Foss, K., Lawrence Merritt, J., Thies, J., Mefford, H. C. and Novotny, E. (2018). Expanding clinical phenotype in CACNA1C related disorders: From neonatal onset severe epileptic encephalopathy to late-onset epilepsy. *Am. J. Med. Genet. A.* **176**, 2733–2739.
- Cao, J., Packer, J. S., Ramani, V., Cusanovich, D. A., Huynh, C., Daza, R., Qui, X., Lee, C., Furlan, S. N., Steemers, F. J., et al. (2017). Comprehensive single-cell transcriptional profiling of a multicellular organism. *Science* **357**, 661–667.
- Chung, H., Parkhurst, C. N., Magee, E. M., Phillips, D., Habibi, E., Chen, F., Yeung, B. Z., Waldman, J., Artis, D. and Regev, A. (2021). Joint single-cell measurements of nuclear proteins and RNA in vivo. *Nat. Methods* **18**, 1204–1212.
- Dräger, N. M., Sattler, S. M., Huang, C. T.-L., Teter, O. M., Leng, K., Hashemi, S. H., Hong, J., Aviles, G., Clelland, C. D., Zhan, L., et al. (2022). A CRISPRi/a platform in human iPSC-derived microglia uncovers regulators of disease states. *Nat. Neurosci.* **25**, 1149–1162.
- Echeverri, C. J. and Perrimon, N. (2006). High-throughput RNAi screening in cultured cells: a user's guide. *Nat. Rev. Genet.* **7**, 373–384.
- Fuchs, A., Riegler, S., Ayatollahi, Z., Cavallari, N., Giono, L. E., Nimeth, B. A., Mutanwad, K. V., Schweighofer, A., Lucyshyn, D., Barta, A., et al. (2021). Targeting alternative splicing by RNAi: from the differential impact on splice variants to triggering artificial pre-mRNA splicing. *Nucleic Acids Res.* **49**, 1133–1151.
- Gogolou, A., Souilhol, C., Granata, I., Wymeersch, F. J., Manipur, I., Wind, M., Frith, T. J., Guarini, M., Bertero, A., Bock, C., et al. (2022). Early anteroposterior regionalisation of human neural crest is shaped by a pro-mesodermal factor. *eLife* **11**, e74263.
- González, F., Zhu, Z., Shi, Z.-D., Lelli, K., Verma, N., Li, Q. V. and Huangfu, D. (2014). An iCRISPR Platform for Rapid, Multiplexable,

- and Inducible Genome Editing in Human Pluripotent Stem Cells. *Cell Stem Cell* **15**, 215–226.
- Gu, S. and Kay, M. A.** (2010). How do miRNAs mediate translational repression?
- Hofbauer, P., Jahnel, S. M., Papai, N., Giesshammer, M., Deyett, A., Schmidt, C., Penc, M., Tavernini, K., Grdseloff, N., Meledeth, C., et al.** (2021). Cardioids reveal self-organizing principles of human cardiogenesis. *Cell* **184**, 3299–3317.e22.
- Huntzinger, E. and Izaurralde, E.** (2011). Gene silencing by microRNAs: contributions of translational repression and mRNA decay. *Nat. Rev. Genet.* **12**, 99–110.
- Karbassi, E., Fenix, A., Marchiano, S., Muraoka, N., Nakamura, K., Yang, X. and Murry, C. E.** (2020). Cardiomyocyte maturation: advances in knowledge and implications for regenerative medicine. *Nat. Rev. Cardiol.* **17**, 341–359.
- Karbassi, E., Padgett, R., Bertero, A., Reinecke, H., Klaiman, J. M., Yang, X., Hauschka, S. D. and Murry, C. E.** (2024). Targeted CRISPR activation is functional in engineered human pluripotent stem cells but undergoes silencing after differentiation into cardiomyocytes and endothelium. *Cell. Mol. Life Sci.* **81**, 95.
- Kisielow, M., Kleiner, S., Nagasawa, M., Faisal, A. and Nagamine, Y.** (2002). Isoform-specific knockdown and expression of adaptor protein ShcA using small interfering RNA. *Biochem. J.* **363**, 1–5.
- Leng, K., Rose, I. V. L., Kim, H., Xia, W., Romero-Fernandez, W., Rooney, B., Koontz, M., Li, E., Ao, Y., Wang, S., et al.** (2022). CRISPRi screens in human iPSC-derived astrocytes elucidate regulators of distinct inflammatory reactive states. *Nat. Neurosci.* **25**, 1528–1542.
- Li, S., Prasanna, X., Salo, V. T., Vattulainen, I. and Ikonen, E.** (2019). An efficient auxin-inducible degron system with low basal degradation in human cells. *Nat. Methods* **16**, 866–869.
- Li, C., Fleck, J. S., Martins-Costa, C., Burkard, T. R., Themann, J., Stuempflen, M., Peer, A. M., Vertesy, Á., Littleboy, J. B., Esk, C., et al.** (2023). Single-cell brain organoid screening identifies developmental defects in autism. *Nature* **621**, 373–380.
- Macrae, R. G. C., Colzani, M. T., Williams, T. L., Bayraktar, S., Kuc, R. E., Pullinger, A. L., Bernard, W. G., Robinson, E. L., Davenport, E. E., Maguire, J. J., et al.** (2023). Inducible apelin receptor knockdown reduces differentiation efficiency and contractility of hESC-derived cardiomyocytes. *Cardiovasc. Res.* **119**, 587–598.
- Murray, P., Clegg, R. A., Rees, H. H. and Fisher, M. J.** (2008). siRNA-mediated knockdown of a splice variant of the PK-A catalytic subunit gene causes adult-onset paralysis in *C. elegans*. *Gene* **408**, 157–163.
- Nguyen, Q. H., Lukowski, S. W., Chiu, H. S., Senabouth, A., Bruxner, T. J. C., Christ, A. N., Palpant, N. J. and Powell, J. E.** (2018). Single-cell RNA-seq of human induced pluripotent stem cells reveals cellular heterogeneity and cell state transitions between subpopulations. *Genome Res.* **28**, 1053–1066.
- Ramakrishnan, A., Symeonidi, A., Hanel, P., Schmid, K. T., Richter, M. L., Schubert, M. and Colomé-Tatché, M.** (2023). epiAneufinder identifies copy number alterations from single-cell ATAC-seq data. *Nat. Commun.* **14**, 5846.
- Replogle, J. M., Norman, T. M., Xu, A., Hussmann, J. A., Chen, J., Cogan, J. Z., Meer, E. J., Terry, J. M., Riordan, D. P., Srinivas, N., et al.** (2020). Combinatorial single-cell CRISPR screens by direct guide RNA capture and targeted sequencing. *Nat. Biotechnol.* **38**, 954–961.
- Tanaka, M., Chen, Z., Bartunkova, S., Yamasaki, N. and Izumo, S.** (1999). The cardiac homeobox gene *Csx/Nkx2.5* lies genetically upstream of multiple genes essential for heart development. *Dev. Camb. Engl.* **126**, 1269–1280.
- Tian, R., Gachechiladze, M. A., Ludwig, C. H., Laurie, M. T., Hong, J. Y., Nathaniel, D., Prabhu, A. V., Fernandopulle, M. S., Patel, R., Abshari, M., et al.** (2019). CRISPR Interference-Based Platform for Multimodal Genetic Screens in Human iPSC-Derived Neurons. *Neuron* **104**, 239–255.e12.
- Tian, R., Abarientos, A., Hong, J., Hashemi, S. H., Yan, R., Dräger, N., Leng, K., Nalls, M. A., Singleton, A. B., Xu, K., et al.** (2021). Genome-wide CRISPRi/a screens in human neurons link lysosomal failure to ferroptosis. *Nat. Neurosci.* **24**, 1020–1034.
- Wesley, B. T., Ross, A. D. B., Muraro, D., Miao, Z., Saxton, S., Tomaz, R. A., Morell, C. M., Ridley, K., Zacharis, E. D., Petrus-Reurer, S., et al.** (2022). Single-cell atlas of human liver development reveals pathways directing hepatic cell fates. *Nat. Cell Biol.* **24**, 1487–1498.
- Williams, T. L., Macrae, R. G. C., Kuc, R. E., Brown, A. J. H., Maguire, J. J. and Davenport, A. P.** (2023). Expanding the apelin receptor pharmacological toolbox using novel fluorescent ligands. *Front. Endocrinol.* **14**, 1139121.
- Wu, J., Okamura, D., Li, M., Suzuki, K., Luo, C., Ma, L., He, Y., Li, Z., Benner, C., Tamura, I., et al.** (2015). An alternative pluripotent state confers interspecies chimaeric competency. *Nature* **521**, 316–21.
- Wu, D., Poddar, A., Ninou, E., Hwang, E., Cole, M. A., Liu, S. J., Horlbeck, M. A., Chen, J., Replogle, J. M., Carosso, G. A., et al.** (2022). Dual genome-wide coding and lncRNA screens in neural induction of induced pluripotent stem cells. *Cell Genomics* **2**, 100177.
- Yang, S., Cho, Y. and Jang, J.** (2021). Single cell heterogeneity in human pluripotent stem cells. *BMB Rep.* **54**, 505–515.
- Zhu, L., Choudhary, K., Gonzalez-Teran, B., Ang, Y.-S., Thomas, R., Stone, N. R., Liu, L., Zhou, P., Zhu, C., Ruan, H., et al.** (2022). Transcription Factor GATA4 Regulates Cell Type-Specific Splicing Through Direct Interaction With RNA in Human Induced Pluripotent Stem Cell-Derived Cardiac Progenitors. *Circulation* **146**, 770–787.

5th Nov 2025

Manuscript Number: MSB-2025-13413

Title: Single Cell Transcriptional Perturbome in Pluripotent Stem Cell Models

Dear Prof Bertero,

Thank you for the submission of your revised manuscript to Molecular Systems Biology. I am pleased to inform you that we will be able to accept your manuscript pending the following final amendments:

1) We require an ORCID IDs for corresponding authors - currently Dr. Elisa Balmas does not have an ORCID ID associated with her profile on our manuscript submission system. Please ensure that Dr. Balmas links an ORCID ID prior to resubmission - an email containing instructions on how to do so was sent to her on October 31st.

2) Affiliations: employment in a biotech company should be stated in the "Disclosure and competing interests" statement. We updated our journal's competing interests policy in January 2022 and request authors to consider both actual and perceived competing interests. Please review the policy <https://www.embopress.org/competing-interests> and update your competing interests if necessary.

3) Please format the Data availability section according to the example below:

"The datasets and computer code produced in this study are available in the following databases:

- Chip-Seq data: Gene Expression Omnibus GSE46748 (<https://www.ncbi.nlm.nih.gov/geo/query/acc.cgi?acc=GSE46748>)

- Modeling computer scripts: GitHub (<https://github.com/SysBioChalmers/GECKO/releases/tag/v1.0>)

- [data type]: [full name of the resource] [accession number/identifier] ([doi or URL or identifiers.org/DATABASE:ACCESSION])"

4) Author contributions: Please remove it from the manuscript and specify author contributions in our submission system.

CRedit has replaced the traditional author contributions section because it offers a systematic machine-readable author contributions format that allows for more effective research assessment. You are encouraged to use the free text boxes beneath each contributing author's name to add specific details on the author's contribution. More information is available in our guide to authors:

<https://www.embopress.org/page/journal/17574684/authorguide#authorshippinguidelines>

5) References: Please correct the reference citation in the reference list such that "et al." is only used after the first 10 authors are listed. Please check "Author Guidelines" for more information.

<https://www.embopress.org/page/journal/17574684/authorguide#referencesformat>

6) Our journal encourages inclusion of *data citations in the reference list* to directly cite datasets that were re-used and obtained from public databases. Data citations in the article text are distinct from normal bibliographical citations and should directly link to the database records from which the data can be accessed. In the main text, data citations are formatted as follows: "Data ref: Smith et al, 2001" or "Data ref: NCBI Sequence Read Archive PRJNA342805, 2017". In the Reference list, data citations must be labeled with "[DATASET]". A data reference must provide the database name, accession number/identifiers and a resolvable link to the landing page from which the data can be accessed at the end of the reference. Further instructions are available at .

7) Data not shown: We do not allow statements/conclusions with "data not shown". As per our guidelines, on "Unpublished Data" the journal does not permit citation of "Data not shown". All data referred to in the paper should be displayed in the main or Expanded View figures. Please remove from pages 50 and 51.

8) In the Methods, please take care of the following:

- Cell lines: Please be sure to include a sentence in the Methods as to whether or not the cell lines were recently authenticated.

- Please ensure that a statement on whether or not blinding was done is included in the Methods even if no blinding was done. Please also be sure to update the Author Checklist with this information and where it can be found in the manuscript.

9) Please place individual sections of the manuscript in the following order: Title page - Abstract & Keywords - Introduction - Results - Discussion - Methods - Data Availability - Acknowledgements - Disclosure and Competing Interests Statement - References - Figure Legends - Expanded View Figure Legends.

10) For the figures and figure legends, please take care of the following:

- All figure callouts should be listed sequentially; a callout is missing for Figure S1.

- Please note that information related to n is missing in the legends of figures EV3 F, EV5 B, C; EV6 E, F

2. Please note that n=2 in figures 3B, EV3 A

3. Please note that the error bars are not defined in the legends of figures 3B, 7E, EV5 B; EV6 H, L, M; EV7 B-F.

- Please define the annotated p values ****/***/**/* as well as provide the exact p-values for the same in the legend of figure 6C as appropriate.

- Please note that the exact p values are not provided in the legends of figures 3B, 6B, G; 7E, EV5 C, EV6 H, L, M; EV7 B-F

- Please indicate the statistical test used for data analysis in the legends of figures 2E, F; 3F, 5G, 6C, EV2 K, EV3 F, EV4 D, EV5I

- Please note that scale bar and its definition are missing for figure EV4A.

11) Appendix file: The title page should contain "Appendix for + manuscript title" and a Table of Contents with the page numbers for the listed items; the nomenclature should be Appendix Figure Sx and Appendix Table Sx throughout the manuscript and Appendix PDF; the word "Supplemental" should not be used for the Appendix file/items.

12) Source Data: Please ensure that a completed Source Data checklist is uploaded as a Related Manuscript File. Source Data should be organized as a single source data file (zipped) per figure for main figures (all EV and/or Appendix figure Source Data can be included in a single folder), with the panels clearly visible in the folder structure instead of a single excel file for all Source Data. e.g. all the Source data files for figure 1 need to be saved in a single folder and this needs to be zipped and then uploaded as "SD figure 1.zip" file.

13) As part of the EMBO Publications transparent editorial process initiative (see our policy here: https://www.embopress.org/transparent-process#Review_Process), Molecular Systems Biology will publish online a Peer Review File (PRF) to accompany accepted manuscripts. This file will be published in conjunction with your paper and will include the anonymous referee reports, your point-by-point response and all pertinent correspondence relating to the manuscript. Let us know whether you agree with the publication of the PRF and as here, if you want to remove or not any figures from it prior to publication. Please note that the Authors checklist will be published at the end of the PRF.

14) After your paper is published, we may promote it on social media. If you have any handles or hashtags for Bluesky you would like included, please let us know.

15) Please provide a point-by-point letter INCLUDING my comments and your detailed responses (as Word file).

I look forward to reading a new revised version of your manuscript as soon as possible.

Yours sincerely,

Poonam Bheda, PhD
Scientific Editor
Molecular Systems Biology

Point by Point Response to the Editorial Requests

We are grateful for the Editor's careful feedback and have amended the manuscript accordingly. A version showing all tracked changes (MSB-2025-13413_ManuscriptOriginalFormatTrackedChanges.pdf) has been uploaded as Related Manuscript file for the editorial team's convenience. Below we provide our detailed point-by-point responses to each editorial request.

1) We require an ORCID IDs for corresponding authors - currently Dr. Elisa Balmas does not have an ORCID ID associated with her profile on our manuscript submission system. Please ensure that Dr. Balmas links an ORCID ID prior to resubmission - an email containing instructions on how to do so was sent to her on October 31st.

Dr. Elisa Balmas has now linked her ORCID ID (0000-0002-4600-6809) to her profile in the submission system, as requested.

2) Affiliations: employment in a biotech company should be stated in the "Disclosure and competing interests" statement. We updated our journal's competing interests policy in January 2022 and request authors to consider both actual and perceived competing interests. Please review the policy <https://www.embopress.org/competing-interests> and update your competing interests if necessary.

We have reviewed the updated EMBO Press Competing Interests Policy and revised the Disclosure and Competing Interests section accordingly. The following statement has been added to the manuscript:

"The IMBA filed a patent application (Nr. 21712188.8) on multi-chamber cardioids with S.M. named as inventor. S.M. is a co-founder and SAB member of HeartBeat.bio AG, the IMBA cardioid drug discovery platform spin-off. The other authors declare no competing interests."

We also carefully assessed whether any competing interests should be declared for A.B. (Alessandro Bertero) in relation to the patent application WO2018096343A1, which includes a claim on a two-step inducible knockdown platform ("OPTiKD"). Since this patent does not cover the all-in-one cassette system used in the present work to develop iPS2-seq, has no active commercialization, and A.B. has no financial or advisory relationship with the licensee of the broader patent (bit.bio Ltd, Cambridge UK), we concluded that it does not represent an actual or perceived conflict of interest under EMBO Press policy. We remain, of course, open to including this information should the editors consider it appropriate.

3) Please format the Data availability section according to the example below:

"The datasets and computer code produced in this study are available in the following databases:

- Chip-Seq data: Gene Expression Omnibus GSE46748

(<https://www.ncbi.nlm.nih.gov/geo/query/acc.cgi?acc=GSE46748>)

- Modeling computer scripts: GitHub

(<https://github.com/SysBioChalmers/GECKO/releases/tag/v1.0>)

- [data type]: [full name of the resource] [accession number/identifier] ([doi or URL or identifiers.org/DATABASE:ACCESSION])"

Then the section has been reformatted as requested:

"The dataset and computer code produced in this study are available in the following databases:

- *iPS2-sci-seq - hiPSC-CMs: BioStudies E-MTAB-14102*
(<https://www.ebi.ac.uk/biostudies/ArrayExpress/studies/E-MTAB-14102>)
- *iPS2-10X-seq - monolayer differentiation: BioStudies E-MTAB-14065*
(<https://www.ebi.ac.uk/biostudies/ArrayExpress/studies/E-MTAB-14065>)
- *iPS2-10X-seq - cardioids: BioStudies E-MTAB-14066*
(<https://www.ebi.ac.uk/biostudies/ArrayExpress/studies/E-MTAB-14066>)
- *iPS2-seq - hiPSCs clonal drift: BioStudies E-MTAB-15303*
(<https://www.ebi.ac.uk/biostudies/ArrayExpress/studies/E-MTAB-15303>)
- *iPS2-multi-seq - hiPSCs: BioStudies E-MTAB-15332*
(<https://www.ebi.ac.uk/biostudies/ArrayExpress/studies/E-MTAB-15332>)
- *iPS2-10X-seq - neural organoids: BioStudies E-MTAB-15308*
(<https://www.ebi.ac.uk/biostudies/ArrayExpress/studies/E-MTAB-15308>)
- *iPS2-10X-seq - monoclonal cardioids: BioStudies E-MTAB-15307*
(<https://www.ebi.ac.uk/biostudies/ArrayExpress/studies/E-MTAB-15307>)
- *iPS2-CITE-seq - polyclonal cardioids: BioStudies E-MTAB-15309*
(<https://www.ebi.ac.uk/biostudies/ArrayExpress/studies/E-MTAB-15309>)
- *Monocle3 CellDataSet class files: Zenodo DOI 10.5281/zenodo.11085619*
(<https://zenodo.org/records/15731884>)
- *Computer scripts and software: GitHub*
(<https://github.com/alessandro-bertero/catcheR/tree/dev>)"

4) Author contributions: Please remove it from the manuscript and specify author contributions in our submission system. CRediT has replaced the traditional author contributions section because it offers a systematic, machine-readable author contributions format that allows for more effective research assessment. You are encouraged to use the free text boxes beneath each contributing author's name to add specific details on the author's contribution. More information is available in our guide to authors: <https://www.embopress.org/page/journal/17574684/authorguide#authorshipguidelines>

The Author Contributions section has been removed from the manuscript, and all contributions have been entered in the CRediT taxonomy fields within the submission system

5) References: Please correct the reference citation in the reference list such that "et al." is only used after the first 10 authors are listed. Please check "Author Guidelines" for more information. <https://www.embopress.org/page/journal/17574684/authorguide#referencesformat>

All references have been thoroughly revised to comply with the Molecular Systems Biology and EMBO Press reference style. Specifically, the reference list now:

- Lists up to 10 authors, followed by et al without a trailing period.
- Uses the “Surname Initials” format for all authors, with no punctuation in initials and commas between authors.
- Includes italicized, Index Medicus–style abbreviated journal names, followed by the volume number (bold in the manuscript) and page range.
- Excludes DOIs, except for one item that lacks volume and page numbers.

The full reference list has been reformatted accordingly and double-checked for accuracy and consistency.

6) Our journal encourages inclusion of *data citations in the reference list* to directly cite datasets that were re-used and obtained from public databases. Data citations in the article text are distinct from normal bibliographical citations and should directly link to the database records from which the data can be accessed. In the main text, data citations are formatted as follows: "Data ref: Smith et al, 2001" or "Data ref: NCBI Sequence Read Archive PRJNA342805, 2017". In the Reference list, data citations must be labeled with "[DATASET]". A data reference must provide the database name, accession number/identifiers and a resolvable link to the landing page from which the data can be accessed at the end of the reference. Further instructions are available at <https://www.embopress.org/page/journal/17574684/authorguide#referencesformat>.

All datasets used in this study were generated specifically for this work and have been deposited in the corresponding public repositories as described in the Data and Code Availability section. No previously published or publicly available datasets were re-used or re-analyzed; therefore, no additional data citations are required in the reference list.

7) Data not shown: We do not allow statements/conclusions with "data not shown". As per our guidelines, on "Unpublished Data" the journal does not permit citation of "Data not shown". All data referred to in the paper should be displayed in the main or Expanded View figures. Please remove from pages 50 and 51.

The expressions “not shown” in Figures EV1E and EV2A referred to minor protocol optimization tests (alternative primer designs and a control comparison of library structure, respectively) that were not central to the study’s conclusions. In accordance with the journal’s guidelines, these parenthetical notes have been removed from the text. The relevant methodological context is already captured in the figure legends and Methods section, and no new data needed to be displayed.

8) In the Methods, please take care of the following:

- Cell lines: Please be sure to include a sentence in the Methods as to whether or not the cell lines were recently authenticated.
- Please ensure that a statement on whether or not blinding was done is included in the Methods even if no blinding was done. Please also be sure to update the Author Checklist with this information and where it can be found in the manuscript.

We have added a sentence in the *hiPSC Culture* section clarifying that cell line authentication by short tandem repeat (STR) profiling was not performed after receipt of the lines.

We also included a statement in the *Data visualization and statistical analyses* section noting that blinding, randomization, and formal sample size estimation were not performed, as the experiments involved standardized differentiation protocols and quantitative molecular assays not requiring these procedures.

These additions address both points raised by the editors and have been reflected in the updated Author Checklist.

9) Please place individual sections of the manuscript in the following order: Title page - Abstract & Keywords - Introduction - Results - Discussion - Methods - Data Availability - Acknowledgements - Disclosure and Competing Interests Statement - References - Figure Legends - Expanded View Figure Legends.

The manuscript sections have been verified and are now presented in the required order.

10) For the figures and figure legends, please take care of the following:

- All figure callouts should be listed sequentially; a callout is missing for Figure S1.

Appendix Figure A1 (previously Figure S1) is now explicitly cited in the legend of Figure EV1 (*"Refer to Fig. A1 for details on the optimization steps related to panels B–D"*), ensuring all figures are called out sequentially in the main text and legends.

- Please note that information related to n is missing in the legends of figures EV3 F, EV5 B, C; EV6 E, F2. Please note that n=2 in figures 3B, EV3 A3.

Information on the number of biological replicates (N) has been added to the legends of Figures EV3F, EV5B–C, and EV6E–F as requested. The legends for Figures 3B and EV3A already indicated N = 2 and have been verified for consistency across all related panels.

- Please note that the error bars are not defined in the legends of figures 3B, 7E, EV5 B; EV6 H, L, M; EV7 B-F.

Error bars have been defined in the legends of Figures 3B, 7E, EV5B, EV6H, L, M, and EV7B–F as requested. All figure legends now specify the meaning of the error bars (standard deviation [SD] or standard error of the mean [SEM], as appropriate).

- Please define the annotated p values ****/***/**/* as well as provide the exact p-values for the same in the legend of figure 6C as appropriate.

- Please note that the exact p values are not provided in the legends of figures 3B, 6B, G; 7E, EV5 C, EV6 H, L, M; EV7 B-F

Exact p-values have been added to the legends of all relevant figures, including Figures 3B, 6B, 6C, 6G, 7E, EV5C, EV6H, L, M, and EV7B–F. In cases where the software could not compute an exact value, the notation $p < 0.001$ has been retained.

- Please indicate the statistical test used for data analysis in the legends of figures 2E, F; 3F, 5G, 6C, EV2 K, EV3 F, EV4 D, EV5I

The statistical tests used for data analysis have been explicitly indicated in the legends of Figures 2E, F; 3F; 5G; 6C; EV2K; EV3F; EV4D; and EV5I, as requested.

- Please note that scale bar and its definition are missing for figure EV4A.

A scale bar has been added to Figure EV4A, and its corresponding definition has been included in the figure legend.

11) Appendix file: The title page should contain "Appendix for + manuscript title" and a Table of Contents with the page numbers for the listed items; the nomenclature should be Appendix Figure Sx and Appendix Table Sx throughout the manuscript and Appendix PDF; the word "Supplemental" should not be used for the Appendix file/items.

The Appendix file title has been updated to "*Appendix for Single Cell Transcriptional Perturbome in Pluripotent Stem Cell Models*", and page number references have been added to the Table of Contents.

All references to supplemental materials have been revised accordingly: figures, tables, and protocols are now labeled as *Appendix Figure A1*, *Appendix Table A1*, and *Appendix Protocol 1*, respectively, replacing all former "Supplemental" designations.

12) Source Data: Please ensure that a completed Source Data checklist is uploaded as a Related Manuscript File. Source Data should be organized as a single source data file (zipped) per figure for main figures (all EV and/or Appendix figure Source Data can be included in a single folder), with the panels clearly visible in the folder structure instead of a single excel file for all Source Data. e.g. all the Source data files for figure 1 need to be saved in a single folder and this needs to be zipped and then uploaded as "SD figure 1.zip" file.

The Source Data have been organized as requested. A completed Source Data checklist has been uploaded as a Related Manuscript File. Each main figure now has an individual zipped folder (e.g., SD_Figure1.zip) containing the corresponding data panels, while all Expanded View figures source data are grouped in a single folder.

13) As part of the EMBO Publications transparent editorial process initiative (see our policy here: https://www.embopress.org/transparent-process#Review_Process), Molecular Systems Biology will publish online a Peer Review File (PRF) to accompany accepted manuscripts. This file will be published in conjunction with your paper and will include the anonymous referee reports, your point-by-point response and all pertinent correspondence relating to the manuscript. Let us know whether you agree with the publication of the PRF

and as here, if you want to remove or not any figures from it prior to publication. Please note that the Authors checklist will be published at the end of the PRF.

We agree to the publication of the Peer Review File (PRF) alongside our article. We do not request the removal of any figures or materials.

14) After your paper is published, we may promote it on social media. If you have any handles or hashtags for Bluesky you would like included, please let us know.

We would be delighted for the study to be promoted on social media. Please tag the following Bluesky handles:

- @berterolab.bsky.social (corresponding author lab)
- @mendjanlab.bsky.social (collaborating lab)

Suggested hashtags: #hiPSCs #singlecell #functionalgenomics #organoids

15) Please provide a point-by-point letter INCLUDING my comments and your detailed responses (as Word file).

This point-by-point letter addresses all editorial requests. The revised manuscript, updated figures, Appendix, Source Data, and all related files have been uploaded through the submission system. We thank the Editor for the clear guidance and remain available for any further queries.

17th Nov 2025

Manuscript number: MSB-2025-13413R

Title: Single Cell Transcriptional Perturbome in Pluripotent Stem Cell Models

Dear Prof Bertero,

Thank you again for sending us your revised manuscript. We are now satisfied with the modifications made and it is my pleasure to inform you that your paper has been accepted for publication at Molecular Systems Biology.

Yours sincerely,

Poonam Bheda, PhD
Scientific Editor
Molecular Systems Biology
